# Neural representation and modulation of volitional motivation in response to escalating efforts

Liping Zhang[1] , Chengwei Liu[1], Xiaopeng Zhou[1], Hui Zhou[1], Shengtao Luo[1], Qin Wang[1], Zhimo Yao[1] and Jiang-Fan Chen[1,2]

[1]*The Molecular Neuropharmacology Laboratory and the Eye-Brain Research Center, The State Key Laboratory of Ophthalmology, Optometry and Vision Science, School of Ophthalmology & Optometry and Eye HospitalWenzhou Medical University, Wenzhou, China*
[2]*Oujiang Laboratory (Zhejiang Laboratory for Regenerative Medicine, Vision and Brain Health), School of Ophthalmology & Optometry and Eye Hospital, Wenzhou Medical University, Wenzhou, China*

Handling Editors: Richard Carson & Jing-Ning Zhu

The peer review history is available in the Supporting Information section of this article (https://10.1113/JP283915#support-information-section).

**Abstract**  Task-dependent volitional control of the selected neural activity in the cortex is critical to neuroprosthetic learning to achieve reliable and robust control of the external device. The volitional control of neural activity is driven by a motivational factor (volitional motivation), which directly reinforces the target neurons via real-time biofeedback. However, in the absence of motor

**Liping Zhang** received her PhD degree from Wuhan University in 2013. She worked at the school of Optometry and Ophthalmology, Wenzhou Medical University, in 2013 and joined the molecular neuropharmacology lab in 2015. Currently, her research focuses on neural circuit mechanisms of volitional control of neural activity and cognitive disorders in neurodegenerative diseases. **Jiang-Fan Chen** has worked at Massachusetts General Hospital/Harvard Medical School as Instructor (1997) and Assistant Professor (1999) and at Boston University School of Medicine as Associate Professor (2003) and Professor (2010) before returning to Wenzhou Medical University as Professor (2016 to present). Professor Chen's work has contributed to our understanding of adenosine receptor neurobiology as an integrated neuromodulator of cognition and other basal ganglia functions as well as important therapeutic strategies for neurodegenerative and neuropsychiatric disorders. Currently, he is the director of the Molecular Neuropharmacology Lab, School of Optometry and Ophthalmology, Wenzhou Medical University.

L. Zhang and C. Liu contributed equally to this work.

**The Journal of Physiology**

behaviour, how do we evaluate volitional motivation? Here, we defined the criterion ($\Delta F/F$) of the calcium fluorescence signal in a volitionally controlled neural task, then escalated the efforts by progressively increasing the number of reaching the criterion or holding time after reaching the criterion. We devised calcium-based progressive threshold-crossing events (termed 'Calcium PTE') and calcium-based progressive threshold-crossing holding-time (termed 'Calcium PTH') for quantitative assessment of volitional motivation in response to progressively escalating efforts. Furthermore, we used this novel neural representation of volitional motivation to explore the neural circuit and neuromodulator bases for volitional motivation. As with behavioural motivation, chemogenetic activation and pharmacological blockade of the striatopallidal pathway decreased and increased, respectively, the breakpoints of the 'Calcium PTE' and 'Calcium PTH' in response to escalating efforts. Furthermore, volitional and behavioural motivation shared similar dopamine dynamics in the nucleus accumbens in response to trial-by-trial escalating efforts. In general, the development of a neural representation of volitional motivation may open a new avenue for smooth and effective control of brain–machine interface tasks.

(Received 18 October 2022; accepted after revision 7 December 2022; first published online 17 December 2022)

**Corresponding authors** Liping Zhang and Jiang-Fan Chen: The Molecular Neuropharmacology Laboratory and the Eye-Brain Research Centre, The State Key Laboratory of Ophthalmology, Optometry and Vision Science, School of Ophthalmology & Optometry and Eye Hospital, Wenzhou Medical University, Wenzhou, 325027, China. Email: lip_zhang@eye.ac.cn and chenjf555@gmail.com

**Abstract figure legend** Evaluation scheme of volitional motivation and behavioural motivation.

## Key points

- Volitional motivation is quantitatively evaluated by M1 neural activity in response to progressively escalating volitional efforts.
- The striatopallidal pathway and adenosine $A_{2A}$ receptor modulate volitional motivation in response to escalating efforts.
- Dopamine dynamics encode prediction signal for reward in response to repeated escalating efforts during motor and volitional conditioning.
- Mice learn to modulate neural activity to compensate for repeated escalating efforts in volitional control.

## Introduction

The operation of brain–computer interfaces (BCIs) and brain–machine interfaces (BMIs) usually depends on the degree of volitional control of neural activity (Fetz, 2007).The volitional drive on cortical neurons can be demonstrated directly by operant training subjects to control neural activity via biofeedback (Eaton et al., 2017; Fetz, 1969; Ishikawa et al., 2014; Moritz & Fetz, 2011; Schmidt et al., 1977; Wyler & Prim, 1976). Volitional control of single or multiple neurons using biofeedback bypasses the normal biological pathways mediating volitional movements (Moritz & Fetz, 2011). Since there is no direct relationship between volitional control of neurons and their physiological functions, we can set up different criteria to reinforce neural activity via biofeedback. Just as in animal behaviour training, animals are rewarded by setting a criterion to reinforce

their behaviours. The volitional control of neural activity provides a defined link between neural activity and the criteria set by the experimenter, allowing a detailed study of the neural adaptive responses for the changed criteria (Chase et al., 2012).

Motivation, defined as the energizing of behaviour in pursuit of a goal, requires the subject to weigh the costs of an action against its potential benefits (Berridge, 2004; Cook & Artino, 2016; Salamone & Correa, 2012). Motivation is represented by the rewards of maximal efforts against the costs of an action for its potential benefits (Salamone & Correa, 2012). In animal models, this is mainly evaluated by an animal's behavioural response to progressively escalating efforts, with the breakpoints representing the size of the motivation, that is when the animal stops (motor) responding to the efforts (behavioural motivation). Volitional control of neural activity is also driven by the motivational factor

(volitional motivation), which is critical for improving the volitional modulation of neural activity and neuro-prosthetic learning (Kleih, Riccio, Mattia, Kaiser et al., 2011; Kleih, Riccio, Mattia, Schreuder et al., 2011). How do we evaluate volitional motivation in the absence of a motor response? The volitional control of neurons directly reinforces neural activity by biofeedback. The escalating effort for volitional control can be specifically increased by predefined criteria (schedule) to progressively increase the required holding time for neural activity above a defined threshold. Finally, volitional motivation was evaluated here by the response of neuroplasticity to escalating effort, with the breakpoint (maximum plasticity of neurons) representing the size of the volitional motivation.

The imaging of neural activity using calcium indicators (Gcamp6f) has been widely used to observe neural activity based on the fluorescence intensity of the calcium indicator (Chen et al., 2013). In this study, mice volitionally controlled the neural activity of the M1 population under operational conditioning by real-time monitoring of calcium fluorescence signals using a fibre photometry system. We first set a criterion of calcium fluorescence signal (defined threshold) in the volitionally controlled neural activity task. We then progressively increased the efforts by increasing the defined threshold-crossing event (TCE) or holding time after a defined threshold-crossing. We also developed a representation and quantitative analysis of volitional motivation by coupling a volitionally controlled neural task with the scheme of a progressive-ratio task (PRT) (Bradshaw & Killeen, 2012) and a progressive hold-down (PHD) task (Bailey et al., 2015). Specifically, we devised the calcium-based progressive threshold-crossing events (termed 'Calcium PTE') and calcium-based progressive threshold-crossing holding-time (termed 'Calcium PTH') for quantitative assessment of volitional motivation responding to progressively escalating efforts. Using this novel representation of volitional motivation, we demonstrated that volitional motivation was similarly modulated by chemicogenetic and pharmacological manipulation of the striatopallidal pathway and shared similar dopamine dynamics in nucleus accumbens (NAc) in response to escalating efforts as with behavioural motivation. Totally, our findings established the first neural representation of volitional motivation and provided novel insights into circuit and neuromodulator control of volitional motivation that may help overcome bottlenecks in smooth and effective control of BMI tasks.

## Methods

### Ethical approval

Animals were handled in accordance with national and institutional guidelines. All experimental protocols were approved by the Institutional Ethics Committee for Animal Use in Research and Education at Wenzhou Medical University, China (ID Number: WYDW2020-0299). All surgical procedures were performed under aseptic conditions. Following the completion of the protocols, all mice were killed by anaesthetic overdose and cervical dislocation. The investigators understand the ethical principles under which *The Journal of Physiology* operates and the work within this study fully complies with the journal's animal ethics checklist. All efforts were made to reduce the number of animals used.

### Animals

Adult (8–10 weeks old) male C57B6/J mice were purchased from SPF Biotechnology Co., Ltd (Beijing, China), and $A_{2A}$-rM3Ds mice were obtained from the Jackson's labs (JAX Stock No. 017863) as described previously (Farrell et al., 2013). All mice were maintained under a 12/12 h photoperiod (lights on at 08.00 h). After surgery, the mice were individually housed under a 12 h light-dark cycle for at least 14 days before conducting any further experiments. After completing Calcium PTE, the mice rested for half a month and then trained on Calcium PTH. rM3Ds was selectively and stably expressed in striatopallidal neurons in $A_{2A}$-rM3Ds mice and activation of the striatopallidal pathway in $A_{2A}$-rM3Ds mice was achieved by systemic injection of clozapine N-oxide (CNO), which specifically activates rM3Ds in the striatopallidal neurons (Farrell et al., 2013). Blockade of $A_{2A}$Rs by KW6002 and monitoring of dopamine dynamics in NAc were performed with male C57B6/J mice.

### Surgery, virus injection and optic fibre implantation

Mice were anaesthetized with pentobarbital (I.P. 60 mg/kg) and mounted on a stereotaxic apparatus. A homeothermic pad was placed below each mouse to maintain body temperature at ∼36°C. Ophthalmic gel was applied to the eyes to prevent dryness. Each animal was unilaterally injected with 200 nl of rAAV-hsyn-DA4.4-WPRE-hGH (catalogue no. PT-1340; BrainVTA, Wuhan, China) into NAc (AP: 1.0 mm, ML: 1.2 mm, DV: −3.9 mm) and/or injected with 300 nl of AAV9-Syn-GCaMP6f-WPRE-SV40 into the left M1 cortex (AP: 1.50 mm, ML: 1.54 mm, DV: −1 mm) using a Nanojet II injector (Drummond Scientific, Broomall, PA, USA) at a rate of 60 nl/min. The mice were then implanted with an optical fibre (230 $\mu$m O.D., 0.37 NA; Shanghai Fiblaser, Shanghai, China) within a ceramic ferrule at the same virus injection sites of the NAc and M1. The ceramic ferrule was supported with a skull-penetrating M1 and/or NAc screw and dental acrylic resin.

## The volitionally controlled neural task

We used an operant volitionally controlled neural task with closed-loop feedback system by volitional conditioning of population neurons in the M1 cortex by real-time monitoring of calcium fluorescence signal using a fibre photometry system (the low baseline procedure) (Zhang et al., 2020). In the low baseline procedure, the baseline was defined as the lowest $F_0$ value within a 1 min time window and recalculated for every minute using the lowest $F_0$ value (Zhang et al., 2020). Briefly, mice were transfected with AAV9-syn-GCaMP6f-WPRE-SV40 to express the genetically encoded $Ca^{2+}$ indicator GCaMP6f in M1 neurons and implanted with optical fibres into the same area. The mice were then conditioned to increase calcium fluorescence signal in M1 neurons above the defined threshold value within a specific time interval (30 s) to acquire a sucrose drop reward (Fig. 1A). The defined threshold was referenced averaging M1 neural activities over 1 day of instrumental conditioning (pressure lever). This operant volitionally controlled neural task is the basis for all the training in the following task.

In a previous study, we attempted to eliminate the overt movement in an operant volitionally controlled neural task. For example, we examined the temporal disassociation of the volitional control of M1 neural activity from movements of the right forelimb as monitored with EMG recordings (Zhang et al., 2020). Furthermore, the mice did not cross the defined threshold during free movement and foraging. Lastly, the M1 population calcium fluorescence signal in one-lever instrumental behaviour (i.e. by pressing the lever once to get a reward in a trial) displayed different patterns compared to volitional control of neural activity.

## Analysis of volitional motivation by Calcium PTE and Calcium PTH

Development of the representation and quantitative analysis of motivation involved three main steps: (1) establishing an operant volitionally controlled neural task; (2) formation of stable mapping of M1 activity responding to increasing efforts by a fixed ratio schedule; and (3) assessing motivation by Calcium PTE and Calcium PTH. The timeline of the training and testing procedures is illustrated in Figs 1B and 2A. After completing Calcium PTE, six male C57B6/J mice rested for half a month and then trained on Calcium PTH.

**Fixed-ratio 1 (FR1) and fixed-ratio 5 (FR5).** Mice were conditioned to exceed the defined threshold (calcium fluorescence signal, $\Delta F/F$ once (FR1) or five times (FR5) to earn a drop of sucrose (50 rewards per session), and they earned 50 rewards in 30 min. A red light indicated

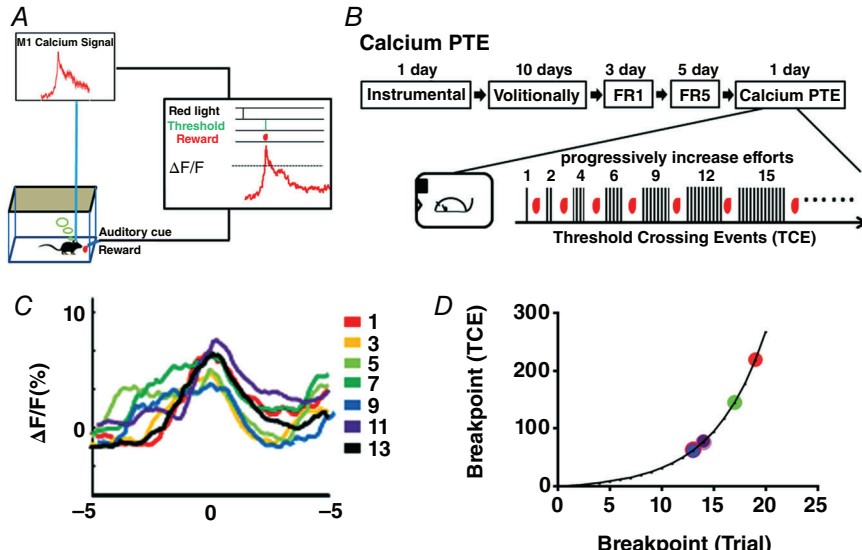

**Figure 1. Development of Calcium PTE for detecting volitional motivation**
*A*, closed-loop volitional control system. The calcium fluorescence signal ($\Delta F/F$) of M1 neurons was monitored in real time by a fibre photometry system. Calcium fluorescence signals ($\Delta F/F$) exceeding the defined threshold value triggered the operant box to deliver a drop of sucrose solution reward. *B*, scheme of the training procedure for Calcium PTE. The upper panel indicates the scheme of the training procedure for Calcium PTE. The lower panel indicates the number of TCEs of the sequential trial for the Calcium PTE test. *C*, the calcium fluorescence signal change in M1 neurons before/after the reward delivery ($\pm$ 5 s) for escalating efforts (trials 1, 3, 5, 7, 9, 11 and 13) in Calcium PTE testing (*n* = 6). *D*, the breakpoint (the maximal TCEs) distribution of six mice by the Calcium PTE test (*n* = 6). [Colour figure can be viewed at wileyonlinelibrary.com]

the beginning of a trial, and an auditory cue would appear when the defined threshold was exceeded, after which there was a 10 s interval. FR1 and FR5 in the instrumental behaviour (PRT) were one or five presses of the pressure lever, respectively, by mice to earn a drop of sucrose

**Calcium PTE.** Mice underwent the volitionally controlled neural task for 10 days, and the proportion of correct trials was 85–100%. The mice then underwent FR1 training for 3 days and FR5 training for 5 days. The mice were then subjected to the Calcium PTE test, where they were required to make progressively increasing numbers of TCEs to obtain a reward. The criterion was set at one TCE for the first time, and the following TCE was calculated by the formula ($TCE = 5 \times e \times (0.2t) - 5$, $t =$ trial number). Each session could last up to 2 h but ended early if the mouse did not cross the defined threshold for 10 min. Motivation was measured by recording the total TCEs in the session and the breakpoint (the total TCEs of the last trial).

**Calcium PTH.** Mice underwent the volitionally controlled neural task for 10 days, and the proportion of correct trials was 85–100%. The mice were trained to earn a reward by continuously holding the calcium fluorescence signal above the defined threshold of 200 ms for 3 days. The mice were then trained to earn a reward by continuously holding the calcium fluorescence signal above the defined threshold of 240 ms for 5 days. The mice were tested in the Calcium PTH task in which rewards could be earned by continuously holding the calcium fluorescence signal above the pre-defined threshold. Every trial's holding time was calculated by the formula (holding

time $= 0.1 \times 1.05^{(t-1)}$, $t =$ trial number). Each session could last up to 2 h but ended early if the mouse did not reach the defined holding time for 10 min. Motivation was measured by recording the total TCEs in the session and the breakpoint (i.e. the holding time of the last trial).

### Calcium and dopamine fluorescence signal analysis

Photometry data were exported to MATLAB Mat files from fibre photometry for further analysis (Li et al., 2016). As in our previous study, we performed data analysis in the MatLab platform (Math Works, Natick, MA, USA) with custom-written programs (Zhang et al., 2020). After smoothing the data with a moving average filter (20 ms span with a 10 ms moving step), we analysed the event-related calcium fluorescence signal and dopamine fluorescence signal in relationship to the reward (with the reward as time '0' point). We derived the values of fluorescence change ($\Delta F/F$) by calculating ($F - F_0)/F_0$, where $F_0$ is the baseline fluorescence signal averaged over a 1–2 s control time window, which was typically set 1–2 s preceding reward delivery. For the dopamine fluorescence signal analysis, the baseline was defined as the average fluorescence signal within $-5$ to $-6$ s prior to reward delivery('0'). 'Height' was analysed as the highest peak of dopamine dynamics of 0 to $-5$ or 0 to $+5$ s of reward delivery ('0'). No recording data were excluded from analysis.

### Fibre photometry

To record fluorescence signals for the GCaMP6f and GRAB$_{DA}$ sensors, a laser beam from a 488 nm laser (OBIS 488LS; Coherent) was reflected by a dichroic mirror

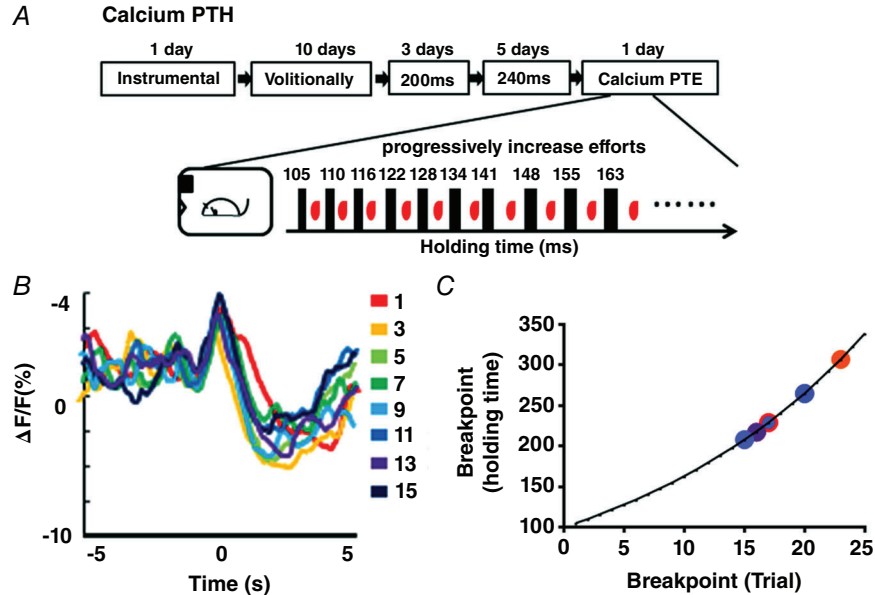

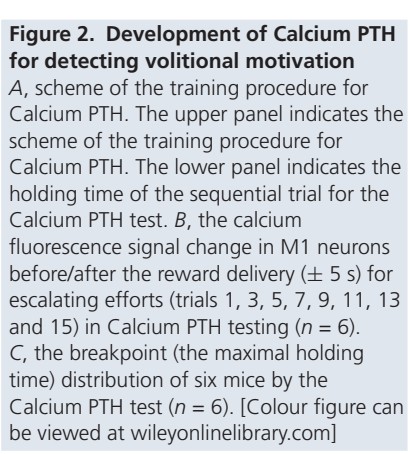

**Figure 2. Development of Calcium PTH for detecting volitional motivation**
*A*, scheme of the training procedure for Calcium PTH. The upper panel indicates the scheme of the training procedure for Calcium PTH. The lower panel indicates the holding time of the sequential trial for the Calcium PTH test. *B*, the calcium fluorescence signal change in M1 neurons before/after the reward delivery ($\pm$ 5 s) for escalating efforts (trials 1, 3, 5, 7, 9, 11, 13 and 15) in Calcium PTH testing (*n* = 6). *C*, the breakpoint (the maximal holding time) distribution of six mice by the Calcium PTH test (*n* = 6). [Colour figure can be viewed at wileyonlinelibrary.com]

(MD498; Thorlabs, Newton, NJ, USA) (Li et al., 2016) (Thinkertech Nanjing Bioscience Inc, Co., Ltd.).

### KW6002 or CNO treatment

The specific adenosine $A_{2A}R$ antagonist KW6002 (5 mg/kg, Sundia, USA) for male C57B6/J mice was suspended in vehicle [15% DMSO (Sigma, St Louis, MO, USA), 15% ethoxylated castor oil (Sigma) and 75% saline] and was administered by intraperitoneal injection. The KW6002 and vehicle groups had six male C57B6/J mice each. CNO (Sigma) for $A_{2A}$-rM3Ds mice was dissolved in DMSO and then administered by intraperitoneal injection (1 mg/kg). The CNO and vehicle groups had six male $A_{2A}$-rM3Ds mice each.

### Immunohistochemistry and imaging

Mice were deeply anaesthetized with an overdose of chloral hydrate (400 mg/kg). Transcardiac perfusion was conducted with saline, followed by 4% paraformaldehyde. Brains were removed and post-fixed in 4% paraformaldehyde for 4–6 h at 4°C, and then allowed to equilibrate using a graded sucrose solution (10%, 20%, 30%). Brain slices (30 $\mu$m) were sectioned on a freezing microtome (Leica CM 307 1850). For immunohistochemistry analysis, we used the following primary antibodies $A_{2A}R$ (frontier, 1:500), mCherry (Clontech, Palo Alto, CA, USA; 1:500), D1 (Clontech, 1:500), together with secondary antibodies goat anti-rabbit AlexaFluor-594 (1:250), goat anti-rat AlexaFluor-555 (1:250). Neurons in the mouse brain expressing Gcamp6f in M1 or/and $GRAB_{DA}$ sensors in NAc were post-fixed, equilibrated, and sectioned. Brain slices were imaged by a fluorescence microscope.

### Data analysis

Statistical analyses were performed using Graphpad Prism 5.01. Data are expressed as mean $\pm$ SD. Unpaired two-tailed Student's *t* tests and Mann–Whitney *U* tests were used to compare two-group data, as appropriate. The mean 'height' of the dopamine fluorescence signals analysed by one-way ANOVA and followed by *post hoc* comparison with Fisher's Least Significant Difference (LSD) test. A *P*-value of < 0.05 was considered statistically significant: $^*P < 0.05$, $^{**}P < 0.01$, $^{***}P < 0.001$, $^{****}P < 0.0001$.

## Results

### Establishment of Calcium PTE and Calcium PTH to assess volitional motivation

Mice were transfected with AAV9-syn-GCaMP6f-WPRE-SV40 to express the genetically encoded $Ca^{2+}$ indicator GCaMP6f in M1 neurons and the calcium fluorescence signal was monitored via a fibre photometry system (Zhang et al., 2020). Mice were trained to perform the volitionally controlled neural task at least 85% correct to obtain a reward (Fig. 1*A*). To quantitatively assess volitional motivation at the neural level, we used the volitionally controlled neural task to establish Calcium PTE and Calcium PTH methods. These methods combined the behavioural concept of motivation (PRT and PHD) to represent volitional motivation by neural activity in response to progressively escalating efforts with the breakpoints representing the size of the motivation (Figs 1 and 2). We set a criterion ($\Delta F/F$, defined threshold) for the calcium fluorescence signal in the volitionally controlled neural task, then escalated efforts by progressively increasing the number of the defined TCE or holding time after a defined threshold-crossing. For the calcium PTE analysis, six mice received 1 day of instrumental conditioning and then 10 days of training of the volitionally controlled neural task, followed by 3 days of FR1 training (one TCE per drop of sucrose; 50 trials/day), followed by 5 days of FR5 training (five TCEs per drop of sucrose; 50 trials/day), and finally calcium PTE was assessed on the last day (Fig. 1*B*, upper panel). In Calcium PTE, TCEs were progressively increased to escalate volitional efforts in the sequential trials (Fig. 1*B*, lower panel). The breakpoint was defined as the maximal number of TCEs at which the subject stops responding to progressive escalation of efforts (progressive increase in TCEs). We analysed the calcium fluorescence signal locked into the reward delivery ($\pm 5$ s) for trials 1, 3, 5, 7 9, 11 and 13, indicating these signals show a difference in response to the escalating efforts (trial by trial) in the calcium PTE test (Fig. 1*C*, $n = 6$). Moreover, the breakpoints of the six mice ranged from 13 to 19 (number of sessions) and 62 to 219 (total TCEs), indicating individual variation in volitional motivation (Fig. 1*D*, $n = 6$).

For Calcium PTH analysis, six mice received 1 day of instrumental conditioning and then training over 10 days for the volitionally controlled neural task. This was followed by 200 ms holding time above the defined threshold (criterion) to earn a drop of sucrose for 3 days, and then 240 ms holding time above the defined threshold to earn a drop of sucrose for 5 days, and finally a day of Calcium PTH test with progressive increasing holding time from 105 to 339 ms (Fig. 2*A*, upper panel). In Calcium PTH, holding time after crossing the defined threshold was progressively increased to escalate efforts in the sequential trials (Fig. 2*A*, lower panel). The breakpoint was defined as the maximal holding time at which the subject stopped responding to a progressive escalation of efforts. Similar to Calcium PTE, there was a difference in neural activity for the escalating efforts during the Calcium PTH test (Fig. 2*B*, $n = 6$). However, the difference between trials was relatively small for Calcium PTH.

Furthermore, Calcium PTH analysis revealed the breakpoint distribution ranged from 218 to 307 ms in holding time above the threshold from 15 to 23 trials (Fig. 2*C*, $n = 6$). Taken together, we concluded that Calcium PTE and Calcium PTH analyses provided a quantitative assessment of volitional motivation at the level of M1 neural activity.

### Striatopallidal pathway and adenosine A$_{2A}$ receptor modulate volitional motivation

We further used the Calcium PTE and Calcium PTH to evaluate the neural circuit modulation of volitional motivation by chemogenetic activation or pharmacological blockade of the striatopallidal pathway. The striatopallidal pathway has been confirmed to exert an inhibitory effect on behavioural motivation (Gallo et al., 2018a, 2018b; Soares-Cunha et al., 2018). For this, we first used a transgenic approach with the genetically mutant acetylcholine receptor hM3Dq, which is unresponsive to endogenous acetylcholine, but can be activated by the exogenous ligand CNO. In this model, the transgenic hM3Dq receptors are preferentially expressed in striatopallidal neurons under control of the adenosine A$_{2A}$ receptor (A$_{2A}$R) gene promoter, which promotes 20-fold higher expression in striatopallidal neurons compared to other brain regions (Farrell et al., 2013). As shown in Fig. 3*A–C*, the transgenic hM3Dq receptors were preferentially expressed in the striatopallidal neurons

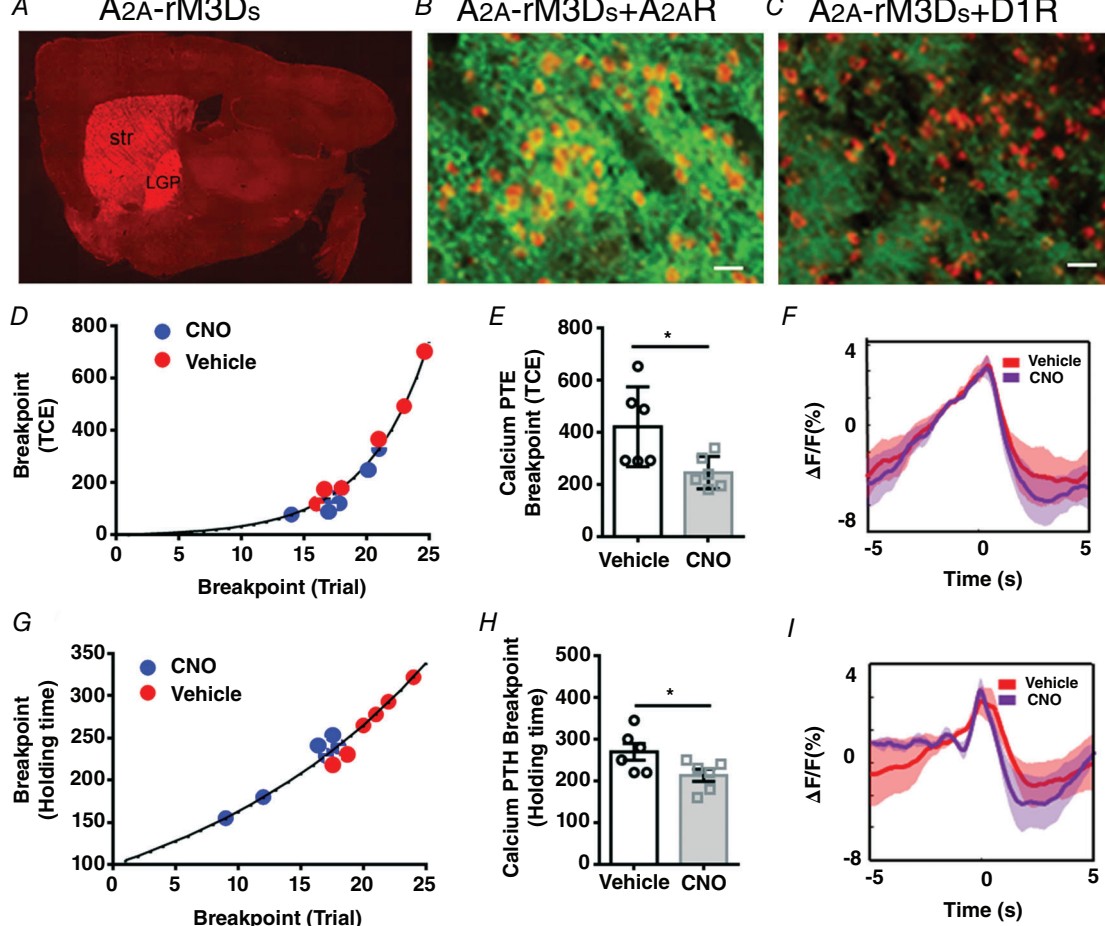

**Figure 3. Striatopallidal pathway regulates volitional motivation**
*A*, sagittal whole-brain expression pattern of A$_{2A}$-rM3Ds mice (Str: striatum; LGP: lateral globus pallidus). *B*, A$_{2A}$-rM3Ds (red) was co-localized with A$_{2A}$ receptor (green) in the striatonigral neurons. Scale bar: 50 $\mu$m. *C*, A$_{2A}$-rM3Ds (red) was not co-localized with D1 receptor (green) in the striatonigral neurons. Scale bar: 50 $\mu$m. *D*, the breakpoint distribution of the Calcium PTE test in individual mice for CNO (blue) and vehicle-treated groups (red) ($n = 12$: CNO group = 6 and vehicle group = 6). *E*, chemogenetic activation of the striatopallidal pathway impaired the breakpoint in the Calcium PTE task (*B*; unpaired *t* test, $P = 0.026$, $t = 2.609$, df = 10). *F*, the mean calcium fluorescence signal change in M1 neurons before (+5 s) and after (+5 s) the reward delivery (0) for Calcium PTE in CNO (purple) and vehicle-treated groups (red). *G–I*, calcium PTH under the same condition as *A–C* (*E*; unpaired *t* test, $P = 0.047$, $t = 2.257$, df = 10). [Colour figure can be viewed at wileyonlinelibrary.com]

and striatopallidal projections (Fig. 3*A*). The transgenic hM3Dq receptors (red) co-localized with A$_{2A}$R (green) in the striatonigral neurons (Fig. 3*B*), but not dopamine D1 receptor (D1R, green) (Fig. 3*C*).

After successfully establishing the stable mapping of the calcium fluorescence signal responding to the escalating efforts, the mice were tested for Calcium PTE (Fig. 3*D*–*F*) and Calcium PTH breakpoints (Fig. 3*G*–*I*) after intra-peritoneal injection of saline or CNO 30 min before the test (1 mg/kg). Compared to the vehicle group, the break-point distribution of Calcium PTE (Fig. 3*D*) and Calcium PTH (Fig. 3*G*) in the CNO-treated group was lower. Moreover, similar to previous reports (Gallo et al., 2018b; Soares-Cunha et al., 2018), chemogenetic activation of the striatopallidal pathway reduced the breakpoint for Calcium PTE (Fig. 3*E*; unpaired *t* test, $P = 0.026$, $t = 2.609$, df = 10) and Calcium PTH (Fig. 3*H*; unpaired *t* test, $P = 0.047$, $t = 2.257$, df = 10). We also analysed calcium fluorescence signal for 5 s before and after reward delivery during Calcium PTE (Fig. 3*F*, $n = 6$) and Calcium PTH (Fig. 3*I*, $n = 6$) between CNO and vehicle groups. The calcium fluorescence signal was not different between CNO and vehicle groups. Consistent with previous studies (Farrell et al., 2013), activation of the striatopallidal pathway inhibited motor function but had no influence on the calcium fluorescence signal in M1 cortical neurons. These results suggested that Calcium PTE and Calcium PTH were sensitive to manipulation of the neural circuit that was known to control behavioural motivation and that activation of the striatopallidal pathway similarly suppressed volitional motivation as with behavioural motivation.

Lastly, we determined the effect of striatal A$_{2A}$Rs on volitional motivation by intraperitoneal injection (5 mg/kg) of the specific A$_{2A}$R antagonist KW6002 30 min before the Calcium PTE or Calcium PTH test (Fig. 4*A*–*F*). The breakpoint distribution of Calcium PTE (Fig. 4*A*) and Calcium PTH (Fig. 4*D*) in the KW6002-treated group was higher than in the vehicle-treated group. The breakpoint for the maximal TCEs increased compared to the vehicle-treated group by Calcium PTE (Fig. 4*B*; Mann–Whitney *U* test, $P = 0.009$). Similarly, the break-point for Calcium PTH in the KW6002 group was higher than in the vehicle group (Fig. 4*E*; unpaired *t* test, $P = 0.004$, $t = 3.699$, df = 10). Moreover, this result also indicated that KW6002 acted indirectly at the striatal A$_{2A}$Rs with feedback onto the M1 neurons to regulate volitional control (Fig. 4*C* and *F*). Collectively, these data revealed that the striatopallidal pathway and A$_{2A}$R activity similarly modulate volitional motivation.

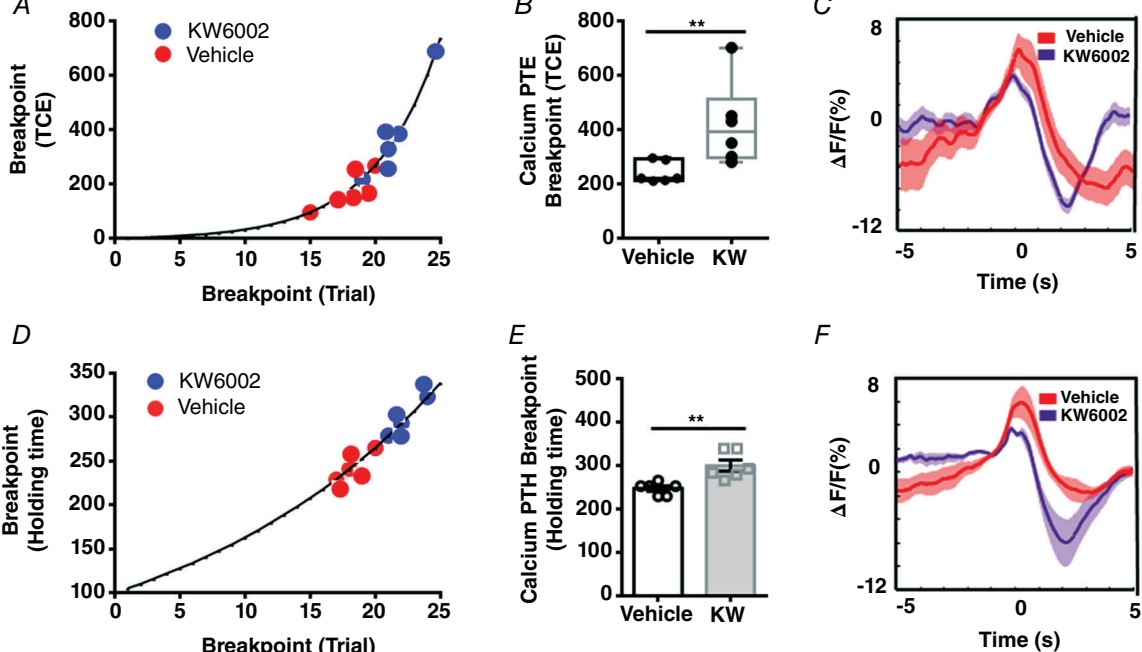

**Figure 4. Adenosine A$_{2A}$Rs regulate volitional motivation**
*A*, blockade of adenosine A$_{2A}$Rs enhanced motivation in volitional control ($n = 12$: KW6002 group = 6 and vehicle group = 6). The breakpoint distribution of Calcium PTE testing in individual mice for the KW6002 (blue) and vehicle-treated groups (red) (*A*). *B*, blockade of adenosine A$_{2A}$Rs improved the breakpoint in Calcium PTE (*B*; Mann-Whitney *U* test, $P = 0.009$). *C*, the mean calcium fluorescence signal change in M1 neurons before (+5 s) and after (+5 s) the reward delivery (0) for Calcium PTE in KW6002 (purple) and vehicle-treated groups (red). *D*–*F*, calcium PTH under the same conditions as *A*–*C* (*E*; unpaired *t* test, $P = 0.004$, $t = 3.699$, df = 10). [Colour figure can be viewed at wileyonlinelibrary.com]

## Escalating efforts produce diminishing dopamine signal in NAc during volitional and behavioural motivation

The dopamine projection from VTA (ventral tegmental area) to NAc is critical for reward motivation and reward-driven learning (Mohebi et al., 2019). To determine the effect of dopamine in these two different motivation assessment methods, we separately monitored the dopamine dynamics in the NAc using GRAB_DA sensors (Sun et al., 2018) for PRT (behaviour motivation test) and Calcium PTE (volitional motivation test). As

illustrated in Fig. 1*B*, the mice performed 3 days of FR1 training, followed by 5 days of FR5 training, and finally 1 day of Calcium PTE after the learning volitionally controlled neural task. Figure 5*A* indicates the loci for expression of GCamp6f in M1 and GRAB_DA sensors in NAc. We analysed the dopamine fluorescence signal ($\Delta F/F$) before (10 s) and after (5 s) the delivery of the reward during FR5 training and Calcium PTE testing. However, we detected two dopamine signal peaks in the NAc during volitional control of neural activity: the prediction signal for the future reward (the signal detected within 5 s prior to the reward delivery, indicated by a

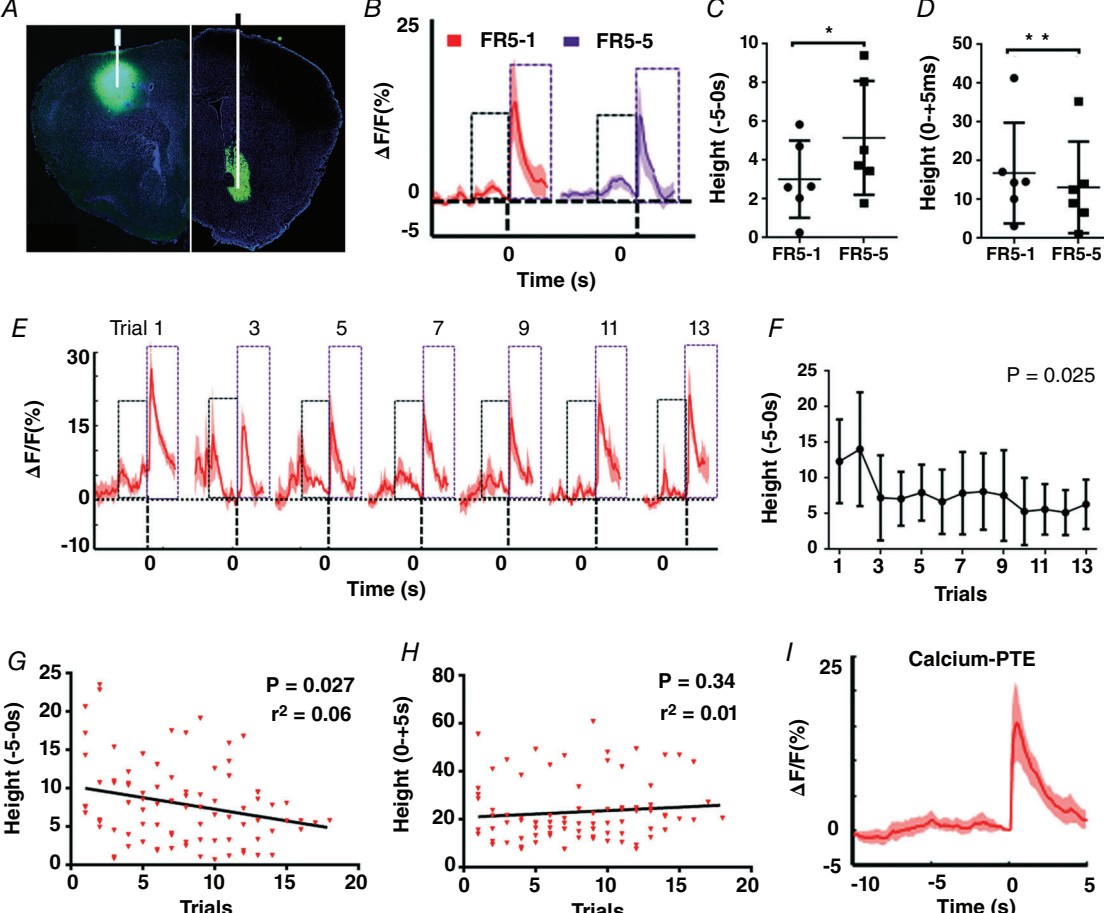

**Figure 5. Analysis of the dopamine dynamics in the NAc for calcium PTE**
*A*, illustration of GCaMP6f-expressing loci in M1 (left) and GRAB_DA sensor-expressing loci in NAc (right) (*n* = 6). The expression of GCaMP6f in M1 was only used for the volitionally controlled neural task. *B*, mean dopamine dynamics changes ($\Delta F/F$) before (10 s) and after (5 s) the reward presentation for FR5-1 and FR5-5 training. *C*, mean 'height' before (−5) the reward presentation is higher in FR5-5 training (*P* = 0.029). *D*, mean 'height' after the reward (+5) presentation is lower in FR5-5 (*P* = 0.009). *E*, mean dopamine dynamic changes before (10 s) and after (5 s) the reward presentation in the Calcium PTE test for the last 13 trials (aligned to the last trial). *F*, mean 'height' for each trial before (5 s) the reward delivery in the Calcium PTE test (RM one-way ANOVA, *P* = 0.025, $F_{2.960, 14.80} = 4.182$). *G–I*, correlation analysis of the dopamine dynamics for the reward prediction (*F*, *P* = 0.027, $r^2 = 0.06$) and reward value (*G*, *P* = 0.34, $r^2 = 0.01$) with the volitional efforts in Calcium PTE. *H*, mean dopamine dynamic changes before (10 s) and after (5 s) the reward delivery in the Calcium PTE test. '0' represents the reward delivery; 'height' represents the highest peak of dopamine dynamics of ±5 s of reward delivery. The black box indicates the prediction signal, and the purple box indicates the reward value signal. [Colour figure can be viewed at wileyonlinelibrary.com]

black box) and reward value (the signal detected within 5 s after the reward delivery, indicated by a purple box) (Fig. 5*B*). To verify dopamine dynamics for reward value, we programmed the time for the reward delivery with a delay of 10 s. Interestingly, the delayed reward delivery by 10 s was associated with a delayed phasic dopamine dynamics by about 10 s (Supplementary Information Fig. S1). These findings also demonstrated that the dopamine dynamics consisted of an early prediction component and a subsequent reward component in the NAc. We then analysed the 'Height' of the dopamine fluorescence signal before and after reward delivery for FR5, indicating there were significant changes in the prediction component [repeated-measures (RM) one-way ANOVA, $P = 0.043$, $F_{2.333, 11.67} = 4.007$] and in the reward component (RM one-way ANOVA, $P = 0.035$, $F_{2.263, 11.32} = 4.422$). Furthermore, compared with FR5-1 (first day of FR5 training) the reward prediction signal (height) of the dopamine fluorescence signal FR5-5 (day 5 of FR5 training) was increased (Fig. 5*C*; $P = 0.029$) and the reward value decreased in FR5-5 (Fig. 5*D*; $P = 0.009$,), indicating the mice increased their prediction of reward, but decreased their sensitivity to reward after 5 days of learning. During the Calcium PTE test, the required TCEs were progressively increased on each trial according to the formula given. However, the dopamine dynamics were analysed for the last 13 trials of the Calcium PTE test, indicating the dopamine dynamics for the prediction signal progressively disappeared (Fig. 5*E* and *F*, RM one-way ANOVA, $P = 0.025$, $F_{2.960, 14.80} = 4.182$). Similarly, the mean reward predictions signal for total trials in the Calcium PTE test also disappeared (Fig. 5*I*). When we analysed the correlation between the reward prediction signals and the escalating volitional efforts for each trial, we found that the reward prediction signal was negatively correlated with the escalating volitional efforts (Fig. 5*G*, $r^2 = 0.06$, $P = 0.027$). However, there was no correlation between the reward value and escalating volitional efforts (Fig. 5*H*, $r^2 = 0.01$, $P = 0.34$). In total, escalating efforts were negatively correlated with dopamine dynamics for reward prediction in NAc but not with the reward value in volitional control of neural activity.

Lastly, we also analysed dopamine dynamics in the NAc in response to escalating efforts (with PRT) during motor skills (Fig. 6). Figure 6*A* indicates the loci for expression of GRAB$_{DA}$ sensors in NAc. There were also two dopamine signal peaks in the NAc during motor skills: the prediction signal for the future reward (the black box) and reward value (the purple box) (Fig. 6*B*). We then analysed the 'Height' of the dopamine fluorescence signal before and after reward delivery for FR5, indicating there were significant changes in the prediction component (RM one-way ANOVA, $P = 0.044$, $F_{2.115, 14.80} = 3.839$) and in the reward component (RM

one-way ANOVA, $P = 0.027$, $F_{2.797, 19.58} = 3.874$). As with volitional control of neural activity, the reward prediction signal and reward value of the dopamine fluorescence signal changed significantly between FR5-1 and FR5-5 (Fig. 6*C*, $P = 0.049$; Fig. 6*D*, $P = 0.01$). The reward prediction signals of dopamine largely disappeared during trial-by-trial PRT testing (Fig. 6*E* and *F*, RM one-way ANOVA, $P = 0.046$, $F_{2.048, 10.24} = 4.184$). The mean reward predictions signal for total trials also disappeared in the Calcium PTH test (Fig. 6*I*). Similarly, correlation analyses revealed that dopamine dynamics for the reward prediction was negatively correlated with the escalating behavioural efforts during PRT testing (Fig. 6*G*, $r^2 = 0.06$, $P = 0.018$), but there was no correlation between the reward value and escalating behavioural efforts (Fig. 6*H*, $r^2 = 0.01$, $P = 0.56$). These results also indicated that escalating efforts were negatively correlated with dopamine dynamics for reward prediction in NAc but not with the reward value in motor skills.

## Discussion

### Development of the first neural representation and quantitative assessment of volitional motivation

In this study, by utilizing the causal link between neuron activity and criteria set by the experimenter, we have adopted the concept of behavioural motivation in PRT and PHD to develop Calcium PTE and Calcium PTH tests, which allow us to directly link the calcium signal (neural activity) to the escalating effort-related motivation (i.e. a subject is willing to expend to earn a reward) during volitional conditioning of neural activity. This quantitative analysis of volitional motivation by Calcium PTE and Calcium PTH allowed us to determine individual variations in volitional motivation at the neural level. The validity of these calcium-based PTE and PTH analyses for quantification of the volitional motivation is supported by the chemogenetic finding that activation of the striatopallidal pathway inhibited motivation in the neuroprosthetic control, in agreement with previous studies on behavioural motivation (Gallo et al., 2018b; Ruder et al., 2021).

The development of the first neural representation and quantitative method for volitional motivation provides opportunities to address the specific contribution of the neural circuit and neuromodulator for BMI improvement. From the perspective of human subjects, the assessment and training of cognitive impairments in advanced stages of paralysis represent a challenge as the standard assessment of motivation typically involves a motor response. However, some or all motor abilities are lost in stroke patients and in other cases of severe motor loss (Carelli et al., 2017). Therefore, quantitative analysis of motivation at the neuron level in disabled patients may

lead to a new therapeutic approach to enhance motivation during neurorehabilitation.

### Dopamine dynamics in the NAc reflect cue-triggered reward 'wanting' not escalating efforts

Motivation and reinforcement learning has been classically associated with dopamine neurons in the VTA that predominantly project to the NAc (Volkow et al., 2017). The critical role of the dopamine reward system in motivational control of behaviours is supported by the finding that disrupting dopamine transmission by pharmacological and neurotoxic approaches regulates response vigour (Hamid et al., 2016; Salamone & Correa, 2012) and work output (Salamone et al., 2018). Animals with impaired dopamine transmission reallocate their instrumental behaviour away from food-reinforced tasks with high response requirements, and instead select less effortful food-seeking behaviours (Nunes et al., 2013; Salamone et al., 2001). The instrumental output and effort-related choice impaired by dopamine D2 antagonism were reversed by $A_{2A}R$ blockade or genetic deletion (Collins et al., 2012; Farrar et al., 2007; Mott et al., 2009; Pardo et al., 2012; Salamone et al., 2009). Indeed, dopamine dynamics in the NAc encode the reward prediction error (Schultz, 2016a, 2016b, 2016c;

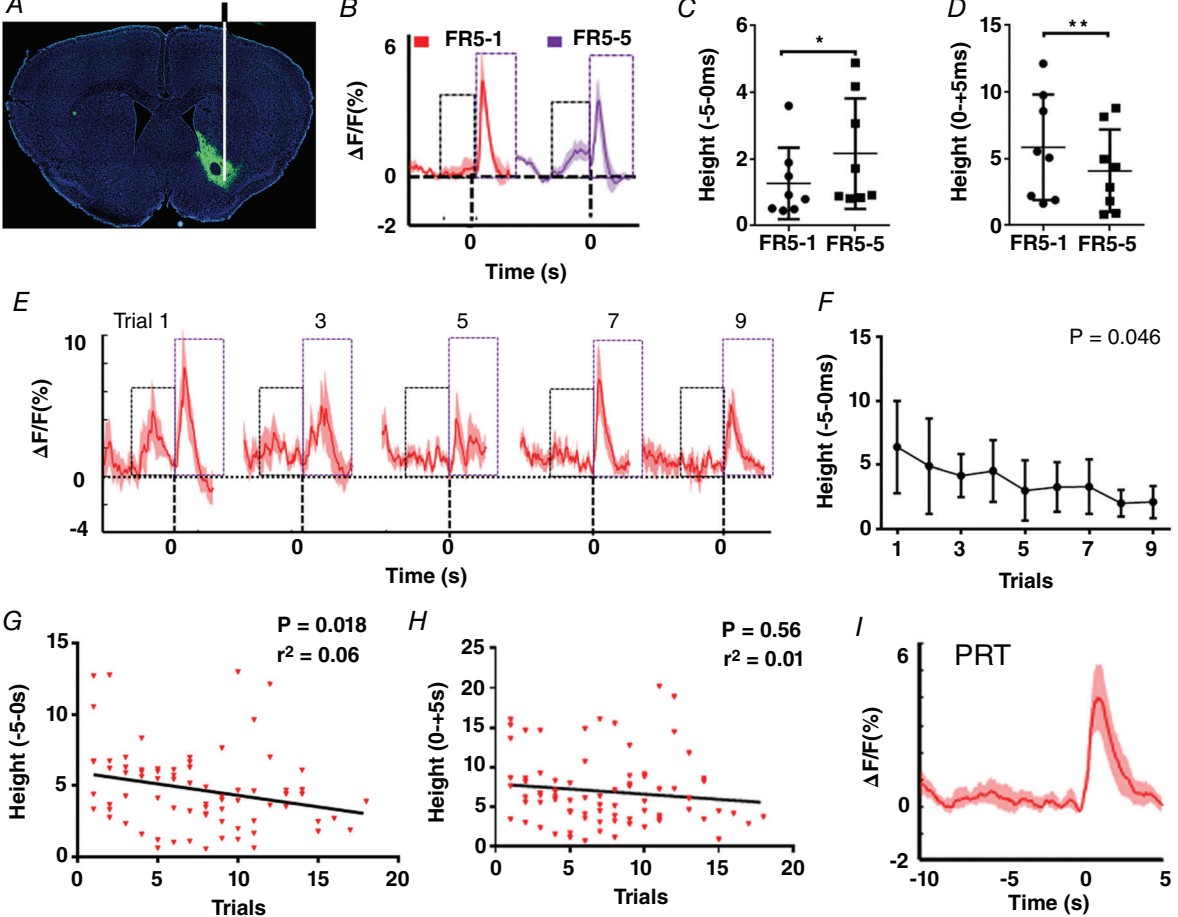

**Figure 6. Analysis of dopamine dynamics in the NAc for PRT**
*A*, illustration of GRAB$_{DA}$ sensor expression and the fluorescence signal observation loci in NAc ($n = 8$). *B*, mean dopamine dynamics changes ($\Delta F/F$) before (10 s) and after (5 s) the reward presentation for FR5-1 and FR5-5 training. *C*, mean 'height' before ($-5$) the reward presentation is higher in FR5-5 training ($P = 0.049$). *D*, mean 'height' after the reward ($+5$) presentation is lower in FR5-5 (pair *t* test, $P = 0.01$). *E*, mean dopamine dynamic changes before (10 s) and after (5 s) the reward presentation in PRT testing for the last nine trials (aligned to the last trial) ($n = 6$). *F*, mean 'height' for each trial before (5 s) the reward delivery in PRT test (RM one-way ANOVA, $P = 0.046$, $F_{2.048, 10.24} = 4.184$, $n = 6$). *F* and *G*, correlation analysis of the dopamine dynamics for the reward prediction (*F*, $P = 0.018$, $r^2 = 0.06$) and reward value (*G*, $P = 0.56$, $r^2 = 0.01$) with the behavioural efforts in PRT. *H*, the mean dopamine dynamic changes before (10 s) and after (5 s) the reward delivery in PRT test. '0' represents the reward delivery, and 'height' represents the highest peak of dopamine dynamics of 0 to $-5$ s or 0 to $+5$ s of reward delivery. The black box indicates the prediction signal, and the purple box indicates the reward value signal. [Colour figure can be viewed at wileyonlinelibrary.com]

du Hoffmann & Nicola, 2014), efforts and delay-related costs (Day et al., 2010) and modulate rewards through delays conferred by the escalating costs (Wanat et al., 2010). Consistent with the prediction error signal (An et al., 2019; Yao et al., 2021), we detected the development of prediction signal (i.e. calcium fluorescence signal associated with cue presentation, before the reward) in the repeated FR1→FR5 trials. Furthermore, according to the incentive salience hypothesis, a reward cue triggers 'wanting' and potentiates instrumental performance for that reward (Wyvell & Berridge, 2000). Applying this hypothesis, a behaviour can be designed to gradually enhance or decrease 'wanting' to test incentive motivation. Hamid et al. (2016) reported that the same dynamically fluctuating dopamine signal influences both current and future motivated behaviour by monitoring dopamine fluctuation in the NAc through the enhancement of 'wanting'. In our study, Calcium PTE and Calcium PTH assessment of volitional motivation revealed that the prediction signal was negatively correlated with the breakpoint, indicating that escalating efforts caused the gradual decrease in 'wanting' for the same reward. Thus, the dopamine (prediction) signal in the (trial-by-trial) progressively escalating effort scheme encoded the 'wanting' but not escalating efforts. Overall, dopamine dynamics in the NAc encodes the rewards cue-triggered 'wanting' and the subjective value of reward.

## Volitional control of neural activity shares brain structures and learning mechanisms, including motivational control

The direct control of neural activity in BMIs may be a consequence of the integration of the cortical system, sub-cortical motivational areas and neurotransmitter system information, indicating that neural activity may represent integrated signals (An et al., 2019; Marsh et al., 2015; Ramkumar et al., 2016; Ramakrishnan et al., 2017; Yao et al., 2021; Zhao et al., 2018). The acquisition of neuro-prosthetic learning also accompanies the creation of neural networks with distinct neural plasticity patterns (Orsborn & Carmena, 2013; Zhang et al., 2020). In contrast to natural motor skill control, BMIs involve only limited (but distinct) direct neurons that are decoded to control the neuroprosthetic device (Chapin et al., 1999; Hira et al., 2013). However, a simple task, such as pressing a lever, are known to involve bilateral control of motor programmes in different brain areas and the brainstem motor 'centres' (Lopez-Huerta et al., 2021). Then, Calcium PTE only reinforced the limited population neurons in the M1 cortex to acquire reward in operant conditioning. Interestingly, dopamine dynamics in NAc were similar in the volitional Calcium PTE test and behavioural PRT. Notably, we recently demonstrated that $A_{2A}R$ antagonists

can enhance volitional control using our current neuro-prosthetic learning paradigm (Zhang et al., 2020). Our follow-on analysis suggested that $A_{2A}Rs$ improve BMI performance by increasing motivational control given that antagonizing $A_{2A}Rs$ enhanced the breakpoint of Calcium PTE (Li et al., 2018; Zhang et al., 2020). Thus, dopamine dynamics and adenosine $A_{2A}R$ activity similarly contribute to volitional motivation control of neural activity in a similar manner to behavioural motivation. Furthermore, as learning and skilful volitional control of neural activity rely on the natural motor repertoire (Hwang et al., 2013), increasing evidence suggests that both motor and neuroprosthetic learning processes share a common circuit structure. For example, corticostriatal plasticity is also essential for learning intentional neuroprosthetic skills (Koralek et al., 2012, 2013) and the emergence of coordinated neural dynamics underlies neuroprosthetic learning. Moreover, reaching proficient control with cohesive neural firing patterns (Koralek et al., 2012, 2013; Marchesotti et al., 2017; Neely RM et al., 2018) similarly requires reinforcement learning with a lot of repetitive training to produce stable representation mapping (Athalye, 2018; Pohlmeyer et al., 2014). Our study further confirms that both behavioural and volitional conditionings are driven by motivational factors with similar modulation at the neural circuit and the neurotransmitter levels. Indeed, we found that chemogenetic activation of the striatopallidal neurons similarly suppresses volitional motivation (as evident by reduced breakpoint of Calcium PTE) and behavioural motivation (Farrell et al., 2013; Gallo et al., 2018a, 2018b; Soares-Cunha et al., 2018). Collectively, the above-mentioned studies together with ours suggest that the similarities between volitionally controlled neural activity and control of motor behaviours far outweigh their differences.

## Conclusions

We have developed novel methods for detecting volitional motivation based on representation of the M1 population neural activity in response to progressively escalating efforts. Meanwhile, we further verified volitional control of population neural activity in shared brain structures and learning mechanisms including motivational control with sensorimotor learning.

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

## Additional information

### Data availability statement

Data will be made available upon reasonable request to the corresponding author.

### Competing interests

The authors declare that they have no known competing financial interests or personal relationships that could have appeared to influence the work reported in this paper.

### Author contributions

L.Z. designed the experiments. J.-F.C. and L.Z. conceptualized the project. C.L. performed most of the experiments. X.Z., H.Z. and S.L. provided experimental facilities, administrative assistance, implant surgery and immunohistochemistry. L.Z., Q.W. and Z.Y. analysed the data and acquired the funding. J.-F.C. and L.Z. wrote and revised the paper. All authors read and approved the final version of the manuscript. All persons designated as authors qualify for authorship, and all those who qualify for authorship are listed.

### Funding

This work was supported by Zhejiang Provincial Natural Science Foundation Grant Nos. LY22C090005 (to L.P.Z.), LQ19C090005 (to Q.W.), LQ18C090002 (to Z.M.Y.).

### Compliance with Ethics Requirements

All Institutional and National Guidelines for the care and use of mice were followed.

### Acknowledgements

We thank Shaomin Zhang from Zhejiang University and Yi Zhang from Wenzhou Medical University for commenting on earlier versions of the manuscript.

### Keywords

$A_{2A}$ receptor, BMIs, dopamine, efforts, motivation, NAc, volitional control

### Supporting information

Additional supporting information can be found online in the Supporting Information section at the end of the HTML view of the article. Supporting information files available:

**Statistical Summary Document**
**Peer Review History**
**Figure S1**

