## [Peer Review History · The Journal of Physiology]

Neural Representation and Modulation of Volitional Motivation in Response to Escalating efforts

Liping Zhang, Chengwei Liu, Xiaopeng Zhou, Hui Zhou, Shengtao Luo, Qin Wang, Zhimo Yao, and Jiang-Fan Chen
DOI: 10.1113/JP283915

Corresponding author(s): Liping Zhang (lip_zhang@mail.eye.ac.cn)

Review Timeline:

Submission Date:	04-Aug-2022
Editorial Decision:	05-Sep-2022
Resubmission Received:	18-Oct-2022
Editorial Decision:	07-Nov-2022
Revision Received:	22-Nov-2022
Editorial Decision:	02-Dec-2022
Revision Received:	05-Dec-2022
Accepted:	07-Dec-2022

Senior Editor: Richard Carson

Reviewing Editor: Jing-Ning Zhu

Transaction Report:

Dear Dr Zhang,

Re: JP-RP-2022-283655 "Neural Representation and Modulation of Volitional Motivation in Response to Escalating efforts" by Liping Zhang, Chengwei Liu, Xiaopeng Zhou, Hui Zhou, Shengtao Luo, Qin Wang, Zhimo Yao, and Jiang-Fan Chen

Thank you for submitting your manuscript to The Journal of Physiology. It has been assessed by a Reviewing Editor and by 2 Referees and the reports are copied below.

Please let your co-authors know of the following editorial decision as quickly as possible.

As you will see, in its current form, the manuscript is not acceptable for publication in The Journal of Physiology. In comments to me, the Reviewing Editor expressed interest in the potential of this study, but much work still needs to be done (and this may include new experiments) in order to satisfactorily address the concerns raised in the reports.

In view of this interest, I would like to offer you the opportunity to carry out all of the changes requested in full, and to resubmit a new manuscript using the "Submit Special Case Resubmission for JP-RP-2022-283655..." on your homepage.

We cannot, of course, guarantee ultimate acceptance at this stage as the revisions required are substantial. However, we encourage you to consider the requested changes and resubmit your work to us if you are able to complete or address all changes.

A new manuscript would be renumbered and redated, but the original referees would be consulted wherever possible. An additional referee's opinion could be sought, if the Reviewing Editor felt it necessary. A full response to each of the reports should be uploaded with a new version.

I hope that the points raised in the reports will be helpful to you.

Yours sincerely,

Richard Carson
Senior Editor
The Journal of Physiology

EDITOR COMMENTS

Reviewing Editor:

The manuscript designs a novel behavioral paradigm to assess volitional motivation and reports a role of striatopallidal pathway and adenosine A2A receptor in motivation. Although both reviewers think the manuscript is interesting, it need a substantial revision according to the reviewers' comments, especially demonstrating the relationship between the calcium PTE/PTH in the M1 and behavioral action or volitional motivation, explaining the criterion for behavioral procedures, checking the statistics, and extensively revising sentences.

Besides, it would be better if the authors will clarify the following issues:

1. The activity of the M1 is generally considered to be related to behavior execution. The manuscript need to clarify whether the calcium signal change in the PTE and PTH test was related to movement or motivation.
2. Whether A2AR is specifically expressed in the striatopallidal neurons, which determines whether the chemogenetic and pharmacological manipulations are specific.
3. Figure 2B does not clearly reflect that the holding time of the calcium signal above the threshold gradually increases with the progressively increase efforts. The holding time above the calcium fluorescence threshold across trials should be counted in the PTH test.

Senior Editor:

In the event that you choose to resubmit a new version of the manuscript, I would ask that you first pay particular attention to

prediction signal (i.e., calcium fluorescence signal associated with cue presentation, before the reward) in the repeated FR1→FR5 trials.

a. 1. Marsh, B. T., Tarigoppula, V. S., Chen, C. & Francis, J. T. Toward an autonomous brain machine interface: integrating sensorimotor reward modulation and reinforcement learning. *J Neurosci* 35, 7374-87 (2015).

b. 2. An, J., Yadav, T., Hessburg, J. P. & Francis, J. T. Reward Expectation Modulates Local Field Potentials, Spiking Activity and Spike-Field Coherence in the Primary Motor Cortex. *eNeuro* June 6 2019, (2019).

c. 3. Yao, Z., Hessburg, J. P. & Francis, J. T. Normalization by valence and motivational intensity in the sensorimotor cortices (PMd, M1, and S1). *Sci Rep* 11, 24221 (2021).

d. 4. Ramkumar, P., Dekleva, B., Cooler, S., Miller, L. & Kording, K. Premotor and Motor Cortices Encode Reward. *PLoS One* 11, e0160851 (2016).

e. 5. Ramakrishnan, A. et al. Cortical neurons multiplex reward-related signals along with sensory and motor information. *Proc Natl Acad Sci U S A* 114, E4841-E4850 (2017).

11. For this text many of the above refs would be applicable. "indicating that neural activity may represent integrated signals." In addition, I've added a ref where BMI was performed while considering this integrated activity of reward expectation/motivation and movement related neural decoding. Note: Please do not feel that you must use any of the suggested citations, but if not these references then please do include any other pertinent refs that might take their place.

a. Zhao, Y., Hessburg, J. P., Kumar, J. N. A. & Francis, J. T. Paradigm Shift in Sensorimotor Control Research and Brain Machine Interface Control: The Influence of Context on Sensorimotor Representations. *Front Neuroscience* 12, (2018).

12. Baseline window concerns: "fluorescence change ($\Delta F/F$) by calculating $(F-F_0)/F_0$, where F_0 is the baseline fluorescence signal averaged over a 1-2s-long control time window, which was typically set 1-2 s preceding the trigger events." Again, not clear what the trigger even is in this sentence, please make this clear. Secondly, as there may be changes in the baseline with learning, or changes due to the increased effort with time, the ratio between baseline and post-trigger could be changing due to either movement of the baseline's height, the post-trigger height, or both. It would be helpful to see non-baseline corrected, more raw representations of the data.

13. Changes in variance:

14. This sentence is hard to follow please edit it as I'm not sure what you are saying. "The volitional control of neurons directly reinforce the neural activity and efforts of volitional control can be escalated by the changed criteria to continuously increase neural activity or by continuously increase holding time for neural activity."

15. Please explain how the below is not contradictory as it is stated that the threshold for the volitional neural task comes from the behavioral task, but that the neural activity between the two tasks is opposite. Perhaps I'm missing something that you can help me see. "mice were conditioned to increase calcium fluorescence signal in M1 neurons above the defined threshold value"... "The defined threshold was based on averaging M1 neural activities over 6 days of instrumental conditioning (pressure lever)."..."Lastly, the M1 population calcium fluorescence signal in the behavior learning is opposite to volitional control of neural activity learning."

16. The below statement seems more pessimistic than it may need to be as you can determine what muscles are activated by the brain region you are recording from and then you would only need to obtain EMG from those muscles I would think. Also, some BMI research uses animals or humans that can't move. "Nevertheless, this is a question common to all BMI studies that is ultimately unanswerable without recordings from every muscle in the body."

17. Author Contributions: Please get rid of all the "or" statements and simply put down what everyone did.

18. Do you really mean to say that you, the authors, volitionally controlled the M1 population? "In this study, we volitionally controlled the M1 population neural activity in the operational conditioning by real-time monitoring of calcium fluorescence signal using fiber photometry system" If not please correct this sentence to make it clear who is controlling the M1 population.

19. Some indication as to how the neurons are activated would be helpful to the reader here, as in what is the mechanism of activation. "Activation of the striatopallidal pathway in A2A-rM3Ds mice was performed by CNO."

20. There is no indication as to where the GRABDA sensors were obtained from. In some sections it is written as above and in others it is GRABDA , please be consistent and use one or the other of these.

21. I'm not sure I fully follow the logic behind the below two statements. Please explicitly state what you have in mind as to how these statements make sense as I seem to be missing something.

a. "Moreover, this result also indicated that the effect of KW6002 on M1 neural activity did not affect the volitional control of

neural activity, because M1 signals were lower in KW6002 group compared to the control group(Fig. 4C, 4F).However, the reduction of calcium fluorescence signal after the reward was more pronounced in the KW6002 group compared to the vehicle group (Fig.4C,4F, n=6), which maybe indicated KW6002 enhanced the sensitivity for the reward."

22. In the below statements you state (data not shown) and then mention a ref [23], but you never state what the actual outcome was. Please state explicitly what ref 23 and the (data not shown) indicate.

a. "To verify dopamine dynamics for reward value, we delayed the reward delivery for 10s, then phasic dopamine dynamics also delayed for about 10s (data no shown). These findings are consistent with the phasic dopamine dynamics in the NAc during motor skill tasks [23]."

23. indicated that the dopamine dynamics for reward predictions in the NAc was inversely correlated with motivation, but not with the reward value in motivation test.

24. Don't animals in general choose the path of least effort when obtaining food. I'm assuming some information is missing in the following statement. ..."but they select the path to food reinforcement that requires less effort [30, 31],"

25. As the previous sentence uses ref 9 perhaps you could use a more specific ref for this definition of motivation "Motivation is represented by the rewards of maximal efforts against the costs of an action for its potential benefits [9]."

(see below)

Figures:

Fig.1 It seems panel C shows -2 seconds to +5 seconds, not +-5 as stated in the legend. "C) The calcium fluorescence signal change before/after the reward delivery ({plus minus} 5s) for escalating efforts (trial by trial) in Calcium PTE testing (n = 6)." You may also want to include M1 in the title to make this more clear.

Fig.2 Panel A has PTE rather than PTH in the flow chart. This figure shows the +-5s, perhaps you should change Fig.1 to match Fig.2s format.

Fig.4. C) should read (-5) and after (+5) rather than what is currently written, which is both are +5.

Fig. 5. Is F significant and if so perhaps use the same convention of *, **, *** etc.

Fig. 6. Same as Fig.5.

Referee #2:

General comments: The manuscript "Neural Representation and Modulation of Volitional Motivation in Response to Escalating efforts" by Zhang et al. reported a criterion of calcium fluorescence signal in the progressive ratio task and progressive hold-down task, which may consider as a quantitative assessment of progressively escalating efforts and relate to striatopallidal pathway. It is an interesting design. The authors clearly described the procedure and detected calcium signals in the M1 and dopamine dynamics in the NAc during the behaviors. But the details of the experiments are less compelling. Below are some comments that the authors may need to consider to improve from the current version:

Major concerns:

1. In general, the authors tried to use the calcium PTE and PTH in the M1 to detect volitional motivation. However, it is still

-The Reference List must be in Journal format

-The Journal of Physiology funds authors of provisionally accepted papers to use the premium BioRender site to create high resolution schematic figures. Follow this link and enter your details and the manuscript number to create and download figures. Upload these as the figure files for your revised submission. If you choose not to take up this offer we require figures to be of similar quality and resolution. If you are opting out of this service to authors, state this in the Comments section on the Detailed Information page of the submission form. The link provided should only be used for the purposes of this submission. Authors will be charged for figures created on this premium BioRender account if they are not related to this manuscript submission.

-A Statistical Summary Document, summarising the statistics presented in the manuscript, is required upon revision. It must be on the Journal's template, which can be downloaded from the link in the Statistical Summary Document section here: https://jp.msubmit.net/cgi-bin/main.plex?form_type=display_requirements#statistics

-Papers must comply with the Statistics Policy https://jp.msubmit.net/cgi-bin/main.plex?form_type=display_requirements#statistics

In summary:

-If $n \leq 30$, all data points must be plotted in the figure in a way that reveals their range and distribution. A bar graph with data points overlaid, a box and whisker plot or a violin plot (preferably with data points included) are acceptable formats.

-If $n > 30$, then the entire raw dataset must be made available either as supporting information, or hosted on a not-for-profit repository e.g. FigShare, with access details provided in the manuscript.

- n clearly defined (e.g. x cells from y slices in z animals) in the Methods. Authors should be mindful of pseudoreplication.

-All relevant ' n ' values must be clearly stated in the main text, figures and tables, and the Statistical Summary Document (required upon revision)

-The most appropriate summary statistic (e.g. mean or median and standard deviation) must be used. Standard Error of the Mean (SEM) alone is not permitted.

-Exact p values must be stated. Authors must not use 'greater than' or 'less than'. Exact p values must be stated to three significant figures even when 'no statistical significance' is claimed.

-Statistics Summary Document completed appropriately upon revision

-A Data Availability Statement is required for all papers reporting original data. This must be in the Additional Information section of the manuscript itself. It must have the paragraph heading "Data Availability Statement". All data supporting the results in the paper must be either: in the paper itself; uploaded as Supporting Information for Online Publication; or archived in an appropriate public repository. The statement needs to describe the availability or the absence of shared data. Authors must include in their Statement: a link to the repository they have used, or a statement that it is available as Supporting Information; reference the data in the appropriate sections(s) of their manuscript; and cite the data they have shared in the References section. Whenever possible the scripts and other artefacts used to generate the analyses presented in the paper should also be publicly archived. If sharing data compromises ethical standards or legal requirements then authors are not expected to share it, but must note this in their Statement. For more information, see our Statistics Policy.

Oct. 18, 2022

Re: JP-RP-2022-283915 “**Neural Representation and Modulation of Volitional Motivation in Response to Escalating efforts**”

Dear Editor:

We thank the Editors and Reviewers very much for the constructive comments and suggestions and for the opportunity to revise and improve the manuscript. As suggested, We have performed additional analyses and revised the manuscript in response to the editors and the Reviewer's critical comments and their suggestions. The point-by-point response to the editors and reviewer's comments are provided as following:

Reviewing Editor:

The manuscript designs a novel behavioral paradigm to assess volitional motivation and reports a role of striatopallidal pathway and adenosine A2A receptor in motivation. Although both reviewers think the manuscript is interesting, it need a substantial revision according to the reviewers' comments, especially demonstrating the relationship between the calcium PTE/PTH in the M1 and behavioral action or volitional motivation, explaining the criterion for behavioral procedures, checking the statistics, and extensively revising sentences.

Besides, it would be better if the authors will clarify the following issues:

1. The activity of the M1 is generally considered to be related to behavior execution. The manuscript need to clarify whether the calcium signal change in the PTE and PTH test was related to movement or motivation.

Response: We thank the editor's comments. Indeed, M1 neuronal activity is generally associated with behavior execution, but also with reward anticipation, motivation and motor planning [1-6]. In the previously study [7], we have partially verified the calcium fluorescence signal in M1 for operant volitionally controlled task was not related to overt movement. (1) we have shown the temporal disassociation of the volitional control of M1 neural activity from movements of the right forelimb as monitored with electromyographic (EMG) recordings (below Figure). (2) the mice did not cross the defined threshold during free movement and foraging. (3) the M1 population calcium fluorescence signal in the operant (motor) behavior displayed the different pattern with volitional control of neural activity. Our view of M1 calcium fluorescence signal representation of volitional control is consistent with as the previous findings that arbitrarily selected primary motor cortex (M1) neurons for volitional control have little relationship with native limb movement [9] and that a stable M1 cortical representation for prosthetic function can be stored [8] . Taken together, we believe that M1 calcium fluorescence signals are representation for volitional signal with limited relationship with movement.

The development of Calcium PTE and Calcium PTH based on the operant volitionally controlled task coupled with the concept of representing behavioral motivation by the break-point in response to escalating efforts. Thus, the calcium signal change in M1 for Calcium PTE and Calcium PTH also was not related to the overt movement, but to the volitional signal. In the current study, we progressively increased M1 neural activity with a series of pre-set criterion to progressively escalate volitional efforts and estimated the motivation by maximal efforts via breakpoints in the tasks. Indeed, there were progressively increase in the TCE and PTH as defined by the formula, representing the escalating efforts (and thus the volitional motivation). Therefore, the maximal neural activities (calcium fluorescence signal, either in TCE or PTH) is related to volitional motivation (not movement).

Senior Editor:

In the event that you choose to resubmit a new version of the manuscript, I would ask that you first pay particular attention to the comments that have been provided in relation to the statistical analyses. In particular, it should be demonstrated that all assumptions pertaining to the use of parametric analyses have been satisfied. If this is not the case, non-parametric procedures would be used instead. Given that multiple tests were performed, appropriately stringent tests should be applied to account for potential inflation of the effective alpha level.

Response: We thank the Senior Editor for the comment. We have paid special attention to the statistical analyses. The data in Figure 3B, 3E, 4E were tested and shown to be normally distributed and accordingly we have used the unpaired for data analyses. The data in Figure 4B were tested and found to be not normally distributed, and accordingly we have used the non-parametric Mann-Whitney U test for data analysis. We analyzed the data in Figure 5C, 5D, 5F, 6C, 6D, 6F by RM one-way ANOVA($P < 0.05$), followed by post-hoc comparison with LSD test.

REFEREE COMMENTS

Referee #1:

Title: Neural Representation and Modulation of Volitional Motivation in Response to Escalating efforts

Summary: In this article, the authors present data from mice expressing calcium indicator dyes while recording changes in fluorescence within the primary motor cortex (M1) or the nucleus accumbens. The authors aimed to determine if the amount of effort the mice had to expend to obtain a given reward amount would change as the effort per unit reward increased in a predictable manner, and how modulation of the striatum changes motivational aspects of the subjects as seen via breakpoints in the tasks.

Specific questions and issues:

1. The summary (the first two figures without legends) figure seems to show the rat using its face to press the lever in yellow. Is this in fact what was done? If not, please make the figure more representative of the real situation. In addition, the Volitional motivation figure is also a bit unclear as to what is being represented as compared to the actual experiment.

Response: We thank the reviewer's comments. The mice used the right forelimb to press the lever to receive the reward and we have revised the first Figure accordingly. The first figure was graphical abstract, illustrating how to assess volitional motivation in the current study (the below figure).

2. There are no figure legends for the first two figures shown in the combined .pdf, I'm not sure if these were to be in the supplemental information or somewhere else in the paper?

Response: We thank the reviewer's comments. The first figure in the combined PDF file was graphical abstract and thus no figure legend was included. The second figure was the Figure 5 in the manuscript and have deleted the second figure in the combined PDF file and the figure legend to Figure 5 was included in the manuscript.

events (reward delivery). We have revised “low baseline procedure” and the baseline calcium signal for the event-related behavior in the manuscript.

8. In the above what are the trigger events, as this term is not used elsewhere in the paper?

Response: We thank the reviewer’s comments and have revised “the trigger events” as “reward delivery”.

9. Statistics: It seems from many of the figures that the variance of the two populations is not similar, which is a violation of the assumptions made for using the unpaired t-test. In addition, it is not indicated that a test for normality was conducted. A non-parametric test, such as the Mann-Whitney U-test may be more suited for this data.

Response: We thank the reviewer’s comments on the statistical analyses and have carefully analyzed the data distribution. The data in Figure 3B, 3E and 4E were tested and shown to be normally distributed. Accordingly, we have used parametric analysis (i.e. the unpaired t-test) for these data. The data in Figure 4B were tested and found to be not normally distributed, and accordingly we have used the non-parametric Mann-Whitney U test for data analysis.

10. For the following statement there are several publications showing reward expectation, value, and motivational neural correlates that could be cited in this work. "Consistent with the prediction error signal, we detected the development of prediction signal (i.e., calcium fluorescence signal associated with cue presentation, before the reward) in the repeated FR1→FR5 trials.

Response: We thank the reviewer’s comments and have cited the references b and d (see below) in the revised manuscript: “Consistent with the prediction error signal (a. b. 44-45), we detected the development of prediction signal (i.e., calcium fluorescence signal associated with the cue presentation, before the reward) in the repeated FR1→FR5 trials.” .

- a. 1. Marsh, B. T., Tarigoppula, V. S., Chen, C. & Francis, J. T. Toward an autonomous brain machine interface: integrating sensorimotor reward modulation and reinforcement learning. *J Neurosci* 35, 7374-87 (2015).
- b. 2. An, J., Yadav, T., Hessburg, J. P. & Francis, J. T. Reward Expectation Modulates Local Field Potentials, Spiking Activity and Spike-Field Coherence in the Primary Motor Cortex. *eNeuro* June 6 2019, (2019).
- c. 3. Yao, Z., Hessburg, J. P. & Francis, J. T. Normalization by valence and motivational intensity in the sensorimotor cortices (PMd, M1, and S1). *Sci Rep* 11, 24221 (2021).
- d. 4. Ramkumar, P., Dekleva, B., Cooler, S., Miller, L. & Kording, K. Premotor and Motor Cortices Encode Reward. *PLoS One* 11, e0160851 (2016).
- e. 5. Ramakrishnan, A. et al. Cortical neurons multiplex reward-related signals along with sensory and motor information. *Proc Natl Acad Sci U S A* 114, E4841-E4850 (2017).
- f. Zhao, Y., Hessburg, J. P., Kumar, J. N. A. & Francis, J. T. Paradigm Shift in Sensorimotor Control Research and Brain Machine Interface Control: The Influence of Context on Sensorimotor Representations. *Front Neuroscience* 12, (2018).

11. For this text many of the above refs would be applicable. "indicating that neural activity may represent integrated signals." In addition, I've added a ref where BMI was performed while considering this integrated activity of reward expectation/motivation and movement related neural decoding. Note: Please do not feel that you must use any of the suggested citations, but if not these references then please do include any other pertinent refs that might take their place.

Response: We thank the reviewer’s comments and agree with you. We have cited these refs in the revised manuscript :“The direct control of neural activity in BMIs may be a consequence of the integration of the cortical system, subcortical motivational areas, and neurotransmitter system information, indicating that neural activity may represent integrated signals[a-f, 44, 45, 47-50]”.

are activated by the brain region you are recording from and then you would only need to obtain EMG from those muscles I would think. Also, some BMI research uses animals or humans that can't move. "Nevertheless, this is a question common to all BMI studies that is ultimately unanswerable without recordings from every muscle in the body."

Response: We thank the reviewer's comments on this point. We agree with the reviewer that the recording of specific muscle activity from the corresponding brain regions and of the animals that can't move (after local anesthesia) would partially disassociate the motor activity from volitional control. We have deleted the sentence in the revised manuscript.

17. Author Contributions: Please get rid of all the "or" statements and simply put down what everyone did.

Response: We thank the reviewer's comments and revised the text by deleting "or" statement and write down the specific statement.

18. Do you really mean to say that you, the authors, volitionally controlled the M1 population? "In this study, we volitionally controlled the M1 population neural activity in the operational conditioning by real-time monitoring of calcium fluorescence signal using fiber photometry system" If not please correct this sentence to make it clear who is controlling the M1 population.

Response: We thank the reviewer's comments and the sentence as following: "In this study, mice volitionally controlled the M1 population neural activity in the operational conditioning by real-time monitoring of calcium fluorescence signal using fiber photometry system"

19. Some indication as to how the neurons are activated would be helpful to the reader here, as in what is the mechanism of activation. "Activation of the striatopallidal pathway in A2A-rM3Ds mice was performed by CNO."

Response: We thank the reviewer's comments and have provided the brief description of the chemicogenetic activation of the neurons by A_{2A}R-rM3Ds as following: "The rM3Ds was selectively and stably expressed in striatopallidal neurons in A_{2A}-rM3Ds mice and activation of the striatopallidal pathway in A_{2A}-rM3Ds mice was achieved by systemic injection of CNO which specifically activate rM3Ds in the striatopallidal neurons."

20. There is no indication as to where the GRABDA sensors were obtained from. In some sections it is written as above and in others it is GRABDA, please be consistent and use one or the other of these.

Response: We thank the reviewer's comments and have provided the detailed description for the Method section to clearly state that rAAV-hsyn-DA4.4-WPRE-hGH was obtained from BrainVTA (catalog# PT-1340; Wuhan, China). We revised and used "GRAB_{DA}" consistently throughout the manuscript.

21. I'm not sure I fully follow the logic behind the below two statements. Please explicitly state what you have in mind as to how these statements make sense as I seem to be missing something.

a. "Moreover, this result also indicated that the effect of KW6002 on M1 neural activity did not affect the volitional control of neural activity, because M1 signals were lower in KW6002 group compared to the control group(Fig. 4C, 4F).However, the reduction of calcium fluorescence signal after the reward was more pronounced in the KW6002 group compared to the vehicle group (Fig.4C,4F, n=6), which maybe indicated KW6002 enhanced the sensitivity for the reward."

Response: We thank the reviewer's comments on this point. First, the A_{2A}Rs are expressed at the high level in the striatopallidal neurons and the low to moderate levels in cortical neurons and both the cortical and striatal A_{2A}Rs can exert control of cognition such as working memory as we recently demonstrated[12]. Thus, the direct action of A_{2A}Rs in M1 neurons, or the indirect action at the striatal neurons (with circuit feedback onto the M1 neurons) can regulate volitional control. As shown in Figure 4C and 4F, when we

the trials of the PTE (Figure A) and PRT (Figure B). These results indicate that there were significant changes in the mean “height” in the trials of both the PET (Fig. 5F, RM one-way ANOVA, $P=0.025$, $F(2.960, 14.80) = 4.182$) and PRT (RM one-way ANOVA, $P=0.046$, $F(2.048, 10.24) = 4.184$). We have used the LSD as well as Bonferroni post-hoc comparison (with correction for multiple tests) for post-hoc analysis. The analysis indicated that there was significant decrease between the height of the first trial and the height of the last trial when LSD testing ($p<0.01$) was employed, but the effect was not present when Bonferroni test was employed. Thus, there was apparent decrease in the mean “height” in the trials of the PTE (Figure A) and PRT.

Minor concerns:

1. The real data of location and expression of GCamp6f in M1 and GRABDA sensors in NAc should be shown. The injection site and expression area of the drugs the NAc could affect the behavioral actions sensitively.

Response: We thank the reviewer’s comments on this and have now included the real data showing the location and expression of Gcamp6f in M1 and GRAB_{DA} sensors in NAc in A, B, (Figure 5A, 6A in the revised manuscript).

2. Authors said "To verify dopamine dynamics for reward value, we delayed the reward delivery for 10s, then phasic dopamine dynamics also delayed for about 10s (data not shown)". Please show the data which is important in the study.

Response: We thank the reviewer’s comments on this point. As suggested, we have now included this data set (see the figure below) as the supplemental figure 1. As you can see, “the delayed reward delivery by 10 second was associated with the delayed phasic dopamine dynamics by about 10s. These findings strongly support that the dopamine neurons fire specifically in response to the reward delivery.” We have included this in the revised Result and Discussion.

Xiangpeng Ren, Yue Feng, Piu Chan, Jiang-Fan Chen **The Corticostriatal Adenosine A2A Receptor Controls Maintenance and Retrieval of Spatial Working Memory.** *Biol Psychiatry* 2018, **83**:530-541.

Dear Dr Zhang,

Re: JP-RP-2022-283915X "Neural Representation and Modulation of Volitional Motivation in Response to Escalating efforts" by Liping Zhang, Chengwei Liu, Xiaopeng Zhou, Hui Zhou, Shengtao Luo, Qin Wang, Zhimo Yao, and Jiang-Fan Chen

Thank you for submitting your manuscript to The Journal of Physiology. It has been assessed by a Reviewing Editor and by 2 expert Referees and I am pleased to tell you that it is considered to be acceptable for publication following satisfactory revision.

The reports are copied at the end of this email. Please address all of the points and incorporate all requested revisions, or explain in your Response to Referees why a change has not been made.

NEW POLICY: In order to improve the transparency of its peer review process, The Journal of Physiology publishes online as supporting information the peer review history of all articles accepted for publication. Readers will have access to decision letters, including all Editors' comments and referee reports, for each version of the manuscript and any author responses to peer review comments. Referees can decide whether or not they wish to be named on the peer review history document.

Authors are asked to use The Journal's premium BioRender (<https://biorender.com/>) account to create/redraw their Abstract Figures. Information on how to access The Journal's premium BioRender account is here:

<https://physoc.onlinelibrary.wiley.com/journal/14697793/biorender-access> and authors are expected to use this service. This will enable Authors to download high-resolution versions of their figures. The link provided should only be used for the purposes of this submission. Authors will be charged for figures created on this premium BioRender account if they are not related to this manuscript submission.

I hope you will find the comments helpful and have no difficulty returning your revisions within 4 weeks.

Your revised manuscript should be submitted online using the links in Author Tasks: Link Not Available.

Any image files uploaded with the previous version are retained on the system. Please ensure you replace or remove all files that have been revised.

REVISION CHECKLIST:

- Article file, including any tables and figure legends, must be in an editable format (eg Word)
- Abstract figure file (see above)
- Statistical Summary Document
- Upload each figure as a separate high quality file
- Upload a full Response to Referees, including a response to any Senior and Reviewing Editor Comments;
- Upload a copy of the manuscript with the changes highlighted.

- A potential 'Cover Art' file for consideration as the Issue's cover image;
- Appropriate Supporting Information (Video, audio or data set https://jp.msubmit.net/cgi-bin/main.plex?form_type=display_requirements#supp).

To create your 'Response to Referees' copy all the reports, including any comments from the Senior and Reviewing Editors, into a Word, or similar, file and respond to each point in colour or CAPITALS and upload this when you submit your revision.

I look forward to receiving your revised submission.

If you have any queries please reply to this email and staff will be happy to assist.

Yours sincerely,

12. Baseline window concerns: "fluorescence change ($\Delta F/F$) by calculating $(F-F_0)/F_0$, where F_0 is the baseline fluorescence signal averaged over a 1-2s-long control time window, which was typically set 1-2 s preceding the trigger events." Again, not clear what the trigger event is in this sentence, please make this clear. Secondly, as there may be changes in the baseline with learning, or changes due to the increased effort with time, the ratio between baseline and post-trigger could be changing due to either movement of the baseline's height, the post-trigger height, or both. It would be helpful to see non-baseline corrected, more raw representations of the data.

Response: We thank the reviewer's comments and specified "the trigger events" as "reward delivery". As suggested, we have included a proportion of the raw calcium signal for Calcium PTH. We agreed that the baseline may change with learning, effort and motivation. However, the neurons associated with learning, effort, motivation may activate the specific neurons in the different time. These activated neurons did not have the noticeable effect on the population of neural activity when analyzed random activity. Thus, the event-specific analysis of the calcium signal (e.g. in relationship with the reward) is required to shown specific calcium signal patterns

13. Changes in variance:

14. This sentence is hard to follow please edit it as I'm not sure what you are saying. "The volitional control of neurons directly reinforce the neural activity and efforts of volitional control can be escalated by the changed criteria to continuously increase neural activity or by continuously increase holding time for neural activity."

Response: We thank the reviewer's comments and have revised the sentence as following: "The volitional control of neurons directly reinforces the neural activity by biofeedback. The escalated effort for volitional control can be specifically increased by predefined criteria (schedule) to progressively increase the required holding time for neural activity above the defined threshold".

15. Please explain how the below is not contradictory as it is stated that the threshold for the volitional neural task comes from the behavioral task, but that the neural activity between the two tasks is opposite. Perhaps I'm missing something that you can help me see. "mice were conditioned to increase calcium fluorescence signal in M1 neurons above the defined threshold value"... "The defined threshold was based on averaging M1 neural activities over 6 days of instrumental conditioning (pressure lever)."..."Lastly, the M1 population calcium fluorescence signal in the behavior learning is opposite to volitional control of neural activity learning."

Response: We thank the reviewer's comments and apologized for the misrepresentation here. Actually, we want to express that the M1 population calcium fluorescence signal in the behavior operant motor learning displayed the different patterns with volitional control of neural activity learning (, but not "opposite" patterns as we initially described). We also revised the last sentence as following: "Lastly, the M1 population calcium fluorescence signal in the behavior learning is opposite to volitional control of neural activity learning." As "Lastly, the M1 population calcium fluorescence signal in the operant behavior displayed ~~the~~ different patterns with-compared to the volitional control of neural activity." These different patterns may be interpreted that volitional control of neural activity and operant behavior in M1 may involve different neuronal populations.

16. The below statement seems more pessimistic than it may need to be as you can determine what muscles are activated by the brain region you are recording from and then you would only need to obtain EMG from those muscles I would think. Also, some BMI research uses animals or humans that can't move. "Nevertheless, this is a question common to all BMI studies that is ultimately unanswerable without recordings from every muscle in the body."

Response: We thank the reviewer's comments on this point. We agree with the reviewer that the recording of specific muscle activity from the corresponding brain regions and of the animals that can't move (after local anesthesia) would partially disassociate the motor activity from volitional control. We have deleted the sentence in the revised manuscript.→

17. Author Contributions: Please get rid of all the "or" statements and simply put down what everyone did.

Response: We thank the reviewer's comments and revised the text by deleting "or" statement and write down the specific statement.

18. Do you really mean to say that you, the authors, volitionally controlled the M1 population? "In this study, we volitionally controlled the M1 population neural activity in the operational conditioning by real-time monitoring of calcium fluorescence signal using fiber photometry system" If not please correct this sentence to make it clear who is controlling the M1 population.

Response: We thank the reviewer's comments and the sentence as following: "In this study, mice volitionally controlled the M1 population neural activity in the operational conditioning by real-time monitoring of calcium fluorescence signal using fiber photometry system"

19. Some indication as to how the neurons are activated would be helpful to the reader here, as in what is the mechanism of activation. "Activation of the striatopallidal pathway in A_{2A}-rM3Ds mice was performed by CNO."

Response: We thank the reviewer's comments and have provided the brief description of the chemicogenetic activation of the neurons by A_{2A}R-rM3Ds as following: "The rM3Ds was selectively and stably expressed in striatopallidal neurons in A_{2A}-rM3Ds mice and activation of the striatopallidal pathway in A_{2A}-rM3Ds mice was achieved by systemic injection of CNO which specifically activate rM3Ds in the striatopallidal neurons."

20. There is no indication as to where the GRABDA sensors were obtained from. In some sections it is written as above and in others it is GRABDA, please be consistent and use one or the other of these.

Response: We thank the reviewer's comments and have provided the detailed description for the Method section to clearly state that rAAV-hsyn-DA4.4-WPRE-hGH was obtained from BrainVTA (catalog# PT-1340; Wuhan, China). We revised and used "GRAB_{DA}" consistently throughout the manuscript.

21. I'm not sure I fully follow the logic behind the below two statements. Please explicitly state what you have in mind as to how these statements make sense as I seem to be missing something.

a. "Moreover, this result also indicated that the effect of KW6002 on M1 neural activity did not affect the volitional control of neural activity, because M1 signals were lower in KW6002 group compared to the control group (Fig. 4C, 4F). However, the reduction of calcium fluorescence signal after the reward was more pronounced in the KW6002 group compared to the vehicle group (Fig. 4C, 4F, n=6), which maybe indicated KW6002 enhanced the sensitivity for the reward."

Response: We thank the reviewer's comments on this point. First, the A_{2A}Rs are expressed at ~~the a~~ high level in the striatopallidal neurons and ~~the~~ low to moderate levels in cortical neurons, ~~and b~~ Both the cortical and striatal A_{2A}Rs can exert control of cognition, such as working memory, as we recently demonstrated [12]. Thus, the direct action of A_{2A}Rs in M1 neurons, or the indirect action at the striatal neurons (with circuit

Response: We thank the reviewer's comments on this. As suggested, we have analyzed the mean "height" in the trials of the PTE (Figure A) and PRT (Figure B). These results indicate that there were significant changes in the mean "height" in the trials of both the PET (Fig. 5F, RM one-way ANOVA, $P=0.025$, $F(2.960, 14.80) = 4.182$) and PRT (RM one-way ANOVA, $P=0.046$, $F(2.048, 10.24) = 4.184$). We have used the LSD as well as Bonforroni post-hoc comparison (with correction for multiple tests) for post-hoc analysis. The analysis indicated that there was significant decrease between the height of the first trial and the height of the last trial when LSD testing ($p<0.01$) was employed, but the effect was not presence when Bonforronin test was employed. Thus, there was apparent decrease in the mean "height" in the trials of the PTE (Figure A) and PRT.

Minor concerns:

1. The real data of location and expression of GCamp6f in M1 and GRABDA sensors in NAc should be shown. The injection site and expression area of the drugs the NAc could affect the behavioral actions sensitively.

Response: We thank the reviewer's comments on this and have now included the real data showing the location and expression of Gcamp6f in M1 and GRAB_{DA} sensors in NAc in A, B, (Figure 5A, 6A in the revised manuscript).

2. Authors said "To verify dopamine dynamics for reward value, we delayed the reward delivery for 10s, then phasic dopamine dynamics also delayed for about 10s (data no shown)". Please show the data which is important in the study.

Response: We thank the reviewer's comments on this point. As suggested, we have now included this data set (see the figure below) as the supplemental figure 1. As you can see, "the delayed reward delivery by 10 second was associated with the delayed phasic dopamine dynamics by about 10s. These findings strongly support that the dopamine neurons fire specifically in response to the reward delivery." We have included this in the revised Result and Discussion.

We thank the Editors and Reviewers again for their constructive comments and suggestions and the opportunity to revise and improve the manuscript. We hope these new analyses and revision have fully addressed the reviewer and editor’s concerns and the manuscript is now considered to be acceptable for publication in “*Journal of Physiology*”.

Sincerely yours,
Liping Zhang, PhD
Jiang-Fan Chen, MD PhD

References:

1. Marsh BT, Tarigoppula VS, Chen C, Francis JT: **Toward an autonomous brain machine interface: integrating sensorimotor reward modulation and reinforcement learning.** *J Neurosci* 2015, **35**:7374-7387.
2. Ramkumar P, Dekleva B, Cooler S, Miller L, Kording K: **Premotor and Motor Cortices Encode Reward.** *PLoS One* 2016, **11**:e0160851.
3. Ramakrishnan A, Byun YW, Rand K, Pedersen CE, Lebedev MA, Nicolelis MAL: **Cortical neurons multiplex reward-related signals along with sensory and motor information.** *Proc Natl Acad Sci U S A* 2017, **114**:E4841-E4850.
4. Zhao Y, Hessburg JP, Asok Kumar JN, Francis JT: **Paradigm Shift in Sensorimotor Control Research and Brain Machine Interface Control: The Influence of Context on Sensorimotor Representations.** *Front Neurosci* 2018, **12**:579.
5. An J, Yadav T, Hessburg JP, Francis JT: **Reward Expectation Modulates Local Field Potentials, Spiking Activity and Spike-Field Coherence in the Primary Motor Cortex.** *eNeuro* 2019, **6**.
6. Yao Z, Hessburg JP, Francis JT: **Normalization by valence and motivational intensity in the sensorimotor cortices (PMd, M1, and S1).** *Sci Rep* 2021, **11**:24221.
7. Zhang L, Zhou Y, Liu C, Zheng W, Yao Z, Wang Q, Jin Y, Zhang S, Chen W, Chen JF: **Adenosine A2A receptor blockade improves neuroprosthetic learning by volitional control of population calcium signal in M1 cortical neurons.** *Neuropharmacology* 2020:108250.
8. Karunesh Ganguly JMC: **Emergence of a Stable Cortical Map for Neuroprosthetic Control.** *PLoS Biol* 2009.
9. Law AJ, Rivlis G, Schieber MH: **Rapid acquisition of novel interface control by small ensembles of arbitrarily selected primary motor cortex neurons.** *J Neurophysiol* 2014, **112**:1528-1548.
10. Schiffmann SN, Fisone G, Moresco R, Cunha RA, Ferre S: **Adenosine A2A receptors and basal ganglia physiology.** *Prog Neurobiol* 2007, **83**:277-292.
11. Farrell MS, Pei Y, Wan Y, Yadav PN, Daigle TL, Urban DJ, Lee HM, Sciaky N, Simmons A, Nonneman RJ, et al: **A Galphas DREADD mouse for selective modulation of cAMP production in striatopallidal neurons.** *Neuropsychopharmacology* 2013, **38**:854-862.
12. Zhihui Li XC, Tao Wang, Ying Gao, Fei Li, Long Chen, Jin Xue, Yan He, Yan Li, Wei Guo, Wu Zheng, Liping Zhang, Fenfen Ye,

Xiangpeng Ren, Yue Feng, Piu Chan, Jiang-Fan Chen **The Corticostriatal Adenosine A2A Receptor Controls Maintenance and Retrieval of Spatial Working Memory.** *Biol Psychiatry* 2018, **83**:530-541.

Neural Representation and Modulation of Volitional Motivation in Response to Escalating efforts

Liping Zhang^{1,3*}, Chengwei Liu^{1,3}, Xiaopeng Zhou¹, Hui Zhou¹, Shengtao Luo¹, Qin Wang¹, Zhimo Yao¹, Jiang-Fan Chen^{1, 2,*}

1. The Molecular Neuropharmacology Laboratory and the Eye-Brain Research Center, The State Key Laboratory of Ophthalmology, Optometry and Vision Science, School of Ophthalmology & Optometry and Eye Hospital, Wenzhou Medical University, Wenzhou, 325027, China

2. Oujiang Laboratory (Zhejiang Laboratory for Regenerative Medicine, Vision and Brain Health), School of Ophthalmology & Optometry and Eye Hospital, Wenzhou Medical University, Wenzhou, 325027, China

3. These authors contributed equally to this work

* Corresponding author. E-mail address: lip_zhang@eye.ac.cn (L.P. Zhang); Chenjf555@eye.ac.cn (J.-F. Chen).

Key points:

- Volitional motivation is quantitatively evaluated by the M1 neural activity in response to progressively escalating volitional efforts.
- Striatopallidal pathway and adenosine A_{2A} receptor modulate volitional motivation in response to escalating efforts.
- Dopamine dynamics encode prediction signal for reward in response to repeated escalating efforts during motor and volitional conditioning.
- The mice learn to modulate neural activity to compensate for repeated escalating efforts in volitional control.

Abstract

Task-dependent volitional control of the selected neural activity in the cortex is critical to neuroprosthetic learning to achieve reliable and robust control of the external device. **The volitional control of neural activity is driven by a motivational factor (volitional motivation), which directly reinforces the target neurons via real-time biofeedback.** But in the absence of motor behavior, how to evaluate the volitional motivation? Here, we defined the criterion ($\Delta F/F$) of calcium fluorescence signal in volitionally controlled neural task, then escalated the efforts by progressively increasing the number of times for reaching the criterion or holding time after reaching the criterion. We devised the calcium-based progressive threshold-crossing events (termed “Calcium PTE”) and calcium-based progressive threshold-crossing holding-time (termed “Calcium PTH”) for quantitative assessment of volitional motivation in respond to progressively escalating efforts. Furthermore, we used this novel neural representation of the volitional motivation to explore the neural circuit and neuromodulator bases for volitional motivation. Like behavioral motivation, chemogenetic activation and pharmacological blockade of the striatopallidal pathway decreased and increased, respectively, the breakpoints of the “Calcium PTE” and “Calcium PTH” responding to escalating efforts. Furthermore, volitional and behavioral motivation shared similar dopamine dynamics in the nucleus accumbens in response to trial-by-trial escalating efforts. In general, the development of neural representation of volitional motivation may open new avenue for smooth and effective control of BMI task.

Key words: motivation; BMIs; volitional control; efforts; dopamine; NAc; A_{2A} receptor

Introduction

The operation of brain–computer interfaces (BCIs) and brain–machine interfaces (BMIs) usually depends on the degree of volitional control of neural activity[1].The volitional drive on cortical neurons can be demonstrated directly by operant training subjects to control the neural activity with biofeedback[2-7]. The volitional control single or multiple neuron using biofeedback bypasses the normal biological pathways mediating volitional movements[5]. Since there is no direct relationship between volitional control of neurons and their physiological functions, we can set up different criteria to reinforce neural activity with biofeedback. Just as in animal behavior training, animals are rewarded by setting criterion to reinforce their behaviors. The volitional control of neural activity provides a defined link between neural activity and the criteria set by the experimenter allowing a detailed study of the neural adaptive responses for the changed criteria [8].

Motivation, defined as the energizing of behavior in pursuit of a goal, requires the subject to weigh the costs of an action against its potential benefits [9-11]. Motivation is represented by the rewards of maximal efforts against the costs of an action for its potential benefits [9]. In animal models, this is mainly evaluated by animal’s behavioral response to progressively escalating efforts with the breakpoints representing the size of the motivation, i.e., when animal stops (motor) responding to the efforts (behavioral motivation). Volitional control of neural activity are also driven by motivational factor (volitional motivation), which is critical for improving the volitional modulation of neural activity and neuroprosthetic learning [12, 13]. How to evaluate volitional motivation in the absence of motor response? **The volitional control of neurons directly reinforces the neural activity by biofeedback. The escalated effort for volitional control can be specifically increased by predefined criteria (schedule) to progressively increase the required holding time for neural activity above the defined threshold.** Finally, volitional motivation evaluated by neural plasticity in response to progressively escalating efforts with the breakpoints (maximal plasticity of neurons) representing the size of the volitional motivation.

The imaging of neural activity using calcium indicators (Gcamp6f) has been widely used to observe the neural activity by fluorescence intensity of calcium indicator [14]. In this study, mice volitionally controlled the M1 population neural activity in the operational conditioning by real-time monitoring of calcium fluorescence signal using fiber photometry system. We first set a criterion of calcium fluorescence signal (defined threshold) in the volitionally controlled neural activity task. We then progressively increased the efforts by increasing defined threshold-crossing event (TCE) or holding time after defined threshold-crossing. Furthermore, we developed the representation and quantitative analysis of volitional motivation by coupling volitionally controlled neural task with the scheme of a progressive-ratio task

(PRT) [15] and a progressive hold-down (PHD) task [16]. Specifically, we devised the calcium-based progressive threshold-crossing events (termed Calcium PTE) and calcium-based progressive threshold-crossing holding-time (termed Calcium PTH) for quantitative assessment of volitional motivation responding to progressively escalating efforts. Using this novel representation of volitional motivation, we demonstrated that volitional motivation was similarly modulated by chemogenetic and pharmacological manipulation of the striatopallidal pathway and shared similar dopamine dynamics in nucleus accumbens (NAc) in response to escalating efforts as behavioral motivation. Totally, our findings established the first neural representation of volitional motivation and provided novel insights into circuit and neuromodulator control of volitional motivation that may help overcome bottlenecks in smooth and effective control of BMI tasks.

Methods and Materials

Animals

Animals were handled in accordance with the national and institutional guidelines. All experimental protocols in Methods and Materials were approved by the Institutional Ethics Committee for Animal Use in Research and Education at Wenzhou Medical University, China. Adult (8-10 weeks old) male C57B6/J mice were purchased from SPF (Beijing) Biotechnology Co., Ltd., and A_{2A} -rM3Ds mice were obtained from the Jackson's labs (stock No: 017863) as described previously [17]. All mice were maintained with a 12/12 photoperiod (light on at 8 AM). After surgery, the mice were individually housed with a reversed photoperiod (light on at 8 AM) for at least 14 days before conducting any further experiments. After completing Calcium PTE, the mice rested for half a month and then trained on Calcium PTH. The rM3Ds was selectively and stably expressed in striatopallidal neurons in A_{2A} -rM3Ds mice and activation of the striatopallidal pathway in A_{2A} -rM3Ds mice was achieved by systemic injection of CNO which specifically activate rM3Ds in the striatopallidal neurons[17]. The blockade of A_{2A} Rs by KW6002 and monitoring dopamine dynamics in NAc were performed with male C57B6/J mice.

Surgery, virus injection, and optic fiber implantation

Mice were anesthetized with pentobarbital (i.p. 60 mg/kg) and mounted to a stereotaxic apparatus. A homeothermic pad was placed below each mouse to maintain the body temperature at approximately 36°C. Ophthalmic gel was applied to the eyes to prevent dryness. Each animal was unilaterally injected with 200 nl of rAAV-hsyn-DA4.4-WPRE-hGH (catalogy# PT-1340; BrainVTA, Wuhan, China) into NAc (AP: 1.0 mm, ML: 1.2 mm, DV: -3.9 mm) or/and injected with 300 nl of AAV9-Syn-GCaMP6f-WPRE-SV40 into the left M1 cortex (AP: 1.50 mm, ML: 1.54 mm, DV: -1 mm) using a Nanojet II injector (Drummod Scientific, USA) at a rate of 60 nl/min. The mice were then

implanted with an optical fiber (230 μ m O.D., 0.37 NA; Shanghai fiblaser) within a ceramic ferrule at the same virus injection sites of the NAc and M1. The ceramic ferrule was supported with a skull-penetrating M1 or/and NAc screw and dental acrylic resin.

The volitionally controlled neural task

We employed an operant volitionally controlled neural task with closed-loop feedback system by volitional conditioning of population neurons in the M1 cortex by real-time monitoring of calcium fluorescence signal using fiber photometry system (the low baseline procedure)[18]. In the low baseline procedure, the baseline was defined as the lowest F0 value within 1 min time window and recalculated for every minute using the lowest F0 value[18]. Briefly, mice were transfected with AAV9-syn-GCaMP6f-WPRE-SV40 to express the genetically encoded Ca²⁺ indicator GCaMP6f in M1 neurons and implanted with optical fibers into the same area. Then the mice were conditioned to increase calcium fluorescence signal in M1 neurons above the defined threshold value within a specific time interval (30s) to acquire a sucrose drop reward (Fig.1A). The defined threshold was referenced averaging M1 neural activities over 1 days of instrumental conditioning (pressure lever). This operant volitionally controlled neural task is the basis for all the training in the following task.

In previously study, we have tried to eliminate the overt movement in operant volitionally controlled neural task. For example, we have examined the temporal disassociation of the volitional control of M1 neural activity from movements of the right forelimb as monitored with electromyographic (EMG) recordings [18]. Furthermore, the mice did not cross the defined threshold during free movement and foraging. Lastly, the M1 population calcium fluorescence signal in the operant behavior displayed the different patterns with compared to volitional/volitionally control/controlled of neural activity.

Comment [FIT1]: A clear statement of your definition of operant behavior and volitional should be made clear as one could say the BMI control is also operant.

Analysis of volitional motivation by Calcium PTE and Calcium PTH

The development of the representation and quantitative analysis of motivation involved three main steps: (1) Establishing an operant volitionally controlled neural task; (2) Formation of stable mapping of M1 activity responding to increasing efforts by fixed ratio schedule; (3) Assessing motivation by calcium PTE and calcium PTH. The timeline of the training and testing procedures is illustrated in Fig.1B and Fig.2A. After completing Calcium PTE, 6 male C57B6/J mice rested for half a month and then trained on Calcium PTH.

Fixed-ratio 1(FR1) and Fixed-ratio 5(FR5): Mice were conditioned to exceed the defined threshold (calcium fluorescence signal, $\Delta F/F$) for one time (FR1) or five times (FR5) to earn a drop of sucrose (50 rewards/session), and they earned 50 rewards in 30 min. The red light indicated the beginning

laser beam from a 488-nm laser (OBIS 488LS; Coherent) was reflected by a dichroic mirror (MD498; Thorlabs)[19].

KW6002 or CNO treatment

The specific adenosine A_{2A}R antagonist KW6002 (5 mg/kg, Sundia, United States) for male C57B6/J mice was suspended in vehicle (15% dimethyl sulfoxide (DMSO, Sigma), 15% ethoxylated castor oil (Sigma) and 75% saline) and was administered by intraperitoneal injection. KW6002 and vehicle group had 6 male C57B6/J mice in each group Clozapine N-oxide (CNO, Sigma) for A_{2A}-rM3Ds mice was dissolved in DMSO and then administered by intraperitoneal injection (1 mg/kg). CNO and vehicle group had 6 male A_{2A}-rM3Ds mice in each group.

Immunohistochemistry and image

Mice were deeply anesthetized with an overdose of chloral hydrate (500 mg/kg, 303 i.p.). Transcardiac perfusion was conducted with saline, followed by 4% paraformaldehyde. Brains were removed and post-fixed in 4% paraformaldehyde for 4-6 hours at 4°C, and then allowed to equilibrate using gradient sucrose solution (10%, 20%, 30%). Brain slice (30 µm) were sectioned on a freezing microtome (Leica CM 307 1850). Primary antibodies used: A_{2A}R (frontier, 1:500), mCherry (Clontech, 1:500), D1 (Clontech, 1:500), goat anti-rabbit AlexaFluor-594 (1:250), goat anti-rat AlexaFluor-555 (1:250). The neurons in the mouse brain expressed the Gcamp6f in M1 or/and GRAB_{DA} sensors in NAc was post-fixed, equilibrated, sectioned. Then the brain slice was imaged by a fluorescence microscope.

Data analysis

Statistical analyses were performed using Graphpad Prism 5.01. Data are expressed as mean ± SD. Unpaired two-tailed Student's t-tests, Mann-Whitney U-test were used to compare 2-group data, as appropriate. The mean "height" of the dopamine fluorescence signals were analyzed by with one-way ANOVA and followed by post-hoc comparison with LSD test. A p-value of <0.05 was considered statistically significant. *P < 0.05, **P < 0.01, ***P < 0.001, ****P < 0.0001.

Comment [FJT3]: You never defined what LSD is.

Results

Establishment of Calcium PTE and Calcium PTH to assess volitional motivation

Mice were transfected with AAV9-syn-GCaMP6f-WPRE-SV40 to express the genetically encoded Ca²⁺ indicator GCaMP6f in M1 neurons and the calcium fluorescence signal was monitored by fiber photometry system[18]. Mice were trained to perform the volitionally controlled neural task to reach of at least the correct percentage of 85-100% correct for obtaining the reward (Fig.1A). To quantitatively assess volitional motivation

at the neural level, we adopted the volitionally controlled neural task to establish calcium PTE and calcium PTH methods. These methods combined the behavioral concept of motivation (PRT and PHD) represented volitional motivation by neural activity in response to progressively escalating efforts with the breakpoints representing the size of the motivation (Fig.1-2). We set a criterion ($\Delta F/F$, defined threshold) for calcium fluorescence signal in volitionally controlled neural task, then escalated efforts by progressively increasing the number of times for defined threshold-crossing event (TCE) or holding time after defined threshold-crossing. For the calcium PTE analysis, six mice received one day of instrumental conditioning and then 10 days of training of the volitionally controlled neural task, followed by three days of fixed-ratio 1 (FR1) training (one TCE / drop of sucrose; 50 trials/day), followed by five days of fixed-ratio 5 (FR5) training (five TCEs / drop of sucrose; 50 trials/day), and finally calcium PTE was assessed on the last day (Fig.1B, upper panel). In Calcium PTE, TCEs were progressively increased to escalate volitional efforts in the sequential trials (Fig.1B, lower panel). The breakpoint was defined as the maximal TCEs at which the subject stops responding to progressive escalation of efforts (progressively increase TCEs). We analyzed calcium fluorescence signal locking into the reward delivery (± 5 s) for trial 1, 3, 5, 7, 9, 11, 13, indicating these signal have the difference in response to the escalating efforts (trial by trial) in calcium PTE test (Fig.1C, n=6). Moreover, the breakpoints of the six mice ranged from 13 to 19 (number of sessions) and 62 to 219 (total TCEs), indicating individual variation in volitional motivation (Fig.1D, n=6).

For calcium PTH analysis, six mice received one day of instrumental conditioning and then training over ten days for the volitionally controlled neural task. This was followed by 200 ms holding time above the defined threshold(criterion) to earn a drop of sucrose for three days, and then 240 ms holding time above the defined threshold to earn a drop of sucrose for five days, and finally a day of Calcium PTH test with progressive increasing holding time from 105 to 339 ms (Fig.2A, upper panel). In calcium PTH, holding time after crossing defined threshold was progressively increased to escalate efforts in the sequential trials (Fig.2A, lower panel). The breakpoint was defined as the maximal holding time at which the subject stopped responding to progressive escalation of efforts. Similar to Calcium PTE, there was the difference of the neural activity for the escalating efforts during Calcium PTH test (Fig.2B, n=6). However, the difference between the trial was relatively small for Calcium PTH. Furthermore, Calcium PTH analysis revealed the breakpoint distribution ranged from 218 to 307 ms of holding time above the threshold from 15 to 23 trials (Fig.2C, n=6). Taken together, we concluded that Calcium PTE and Calcium PTH analyses provided a quantitative assessment of volitional motivation at the level of M1 neural activity.

Striatopallidal pathway and adenosine A_{2A} receptor modulate volitional motivation

We further used the Calcium PTE and Calcium PTH to evaluate the neural circuit modulation of volitional motivation by chemogenetic activation or pharmacological blockade of the striatopallidal pathway. The striatopallidal pathway has been confirmed to exert an inhibitory effect on behavioral motivation [20-22]. Towards this, we first employed a transgenic approach with the genetically mutant acetylcholine receptor hM3Dq, which is unresponsive to endogenous acetylcholine, but can be activated by the exogenous ligand clozapine N-oxide (CNO). In this model, the transgenic hM3Dq receptors are preferentially expressed in the striatopallidal neurons under the control of the adenosine A_{2A} receptors (A_{2A}Rs) gene promoter which promotes 20-fold higher expression in striatopallidal neurons compared to other brain regions [17]. As shown in Fig. 3A-C, the transgenic hM3Dq receptors were preferentially expressed in the striatopallidal neurons and striatopallidal projections (Figure 3A). The transgenic hM3Dq receptors (red) co-localized with A_{2A} receptor (green) in the striatonigral neurons (Fig. 3B), but not dopamine D1 receptor(D1R, green) (Fig. 3C).

After successfully establishing the stable mapping of calcium fluorescence signal responding to the escalating efforts, the mice were tested for Calcium PTE (Fig.3D-F) and Calcium PTH breakpoints (Fig.3G-I) after intraperitoneal injection of saline or CNO 30 min before the test (1 mg/kg). Compared to the vehicle group, the breakpoint distribution of Calcium PTE (Fig.3D) and Calcium PTH (Fig.3G) in the CNO-treated group was lower. Moreover, similar to prior reports [20, 21], chemogenetic activation of the striatopallidal pathway reduced the breakpoint for Calcium PTE (Fig.3E; unpaired t-test, P =0.026, t=2.609, df=10) and Calcium PTH (Fig.3H; unpaired t-test, P =0.047, t=2.257, df=10). We also analyzed calcium fluorescence signal for 5 seconds before and after the reward delivery during Calcium PTE (Fig.3F,n=6) and Calcium PTH (Fig.3I,n=6) between CNO and vehicle groups. We observed calcium fluorescence signal was not different between CNO and vehicle groups. Consistent with previous studies [17], activation of the striatopallidal pathway inhibited motor function but had no influence the calcium fluorescence signal in the M1 cortical neurons. These results suggested that the calcium PTE and calcium PTH were sensitive to manipulation of the neural circuit that was known to control behavioral motivation and that activation of the striatopallidal pathway similarly suppressed volitional motivation as behavioral motivation.

Lastly, we determined the effect of striatal A_{2A}Rs on volitional motivation by intraperitoneal injection (5 mg/kg) of the specific A_{2A}R antagonist KW6002 30 min before Calcium PTE or Calcium PTH test (Fig.4A-F). The breakpoint distribution of Calcium PTE (Fig.4A) and Calcium

PTH (Fig.4D) in the KW6002-treated group was higher than the vehicle-treated group. The breakpoint for the maximal TCEs increased compared to the vehicle-treated group by Calcium PTE (Fig.4B; Mann-Whitney U-test, $P = 0.009$). Similarly, the breakpoint for Calcium PTH in the KW6002 group was higher than the vehicle group (Fig.4E; unpaired t-test, $P = 0.004$, $t = 3.699$, $df = 10$). Moreover, this result also indicated that the effect of KW6002 on M1 neural activity did not affect the volitional control of neural activity, because M1 signals were lower and the breakpoint was higher in KW6002 group compared to the control group (Fig. 4C, 4F). Collectively, these data revealed that the striatopallidal pathway and $A_{2A}R$ activity similarly modulate volitional motivation.

Escalating efforts produces diminishing dopamine signal in NAc during volitional and behavioral motivation

The dopamine projection from VTA to NAc is critical for reward motivation and reward-driven learning [23]. To determine the effect of dopamine in these two different motivation assessment methods, we separately monitored the dopamine dynamics in the NAc using GRAB_{DA} sensors [24] for PRT (behavior motivation test) and calcium PTE (volitional motivation test). As illustrated in Fig.1B, the mice performed three days of FR1 training, followed by five days of FR5 training, and finally one day of calcium PTE after the learning volitionally controlled neural task. Fig.5A indicated that the loci for expression GCamp6f in M1 and GRAB_{DA} sensors in NAc. We analyzed the dopamine fluorescence signal ($\Delta F/F$) before (10 s) and after (5 s) the delivery of the reward during FR5 training and calcium PTE testing. However, we detected two dopamine signal peaks in the NAc during volitional control of neural activity: the prediction signal for the future reward (the signal detected within 5 seconds prior to the reward delivery, indicated by black box) and reward value (the signal detected within 5 seconds after the reward delivery, indicated by purple box) (Fig.5B). To verify dopamine dynamics for reward value, we have programmed the time for the reward delivery with a delay by of 10s. Interestingly, the delayed reward delivery by 10 second was associated with the delayed phasic dopamine dynamics by about 10s (supplemental figure 1). These findings also demonstrated that the dopamine neurons to rewards consist of an early prediction component and a subsequent reward component. Then we analyzed the “Height” of the dopamine fluorescence signal before or and after reward delivery for FR5, indicating there were significant changes in the prediction component (RM one-way ANOVA, $P = 0.043$, $F(2.333, 11.67) = 4.007$) and in the reward component (RM one-way ANOVA, $P = 0.035$, $F(2.263, 11.32) = 4.422$). Furthermore, Compared with FR5-1 (first day of FR5 training) the reward prediction signal (height) of dopamine fluorescence signal FR5-5 (day 5 of FR5 training) was increased (Fig.5C; $P = 0.029$) and

the reward value decreased in FR5-5 (Fig.5D; $P=0.009$), indicating the mice increased their prediction of reward, but decreased their sensitivity to reward after five days of learning. During calcium PTE test, the required TCEs were progressively increased on each trial according to the formula. **However, the dopamine dynamics ~~was~~were analyzed for the last 13 trials of Calcium PTE test, indicating the dopamine dynamics for the prediction signal progressively disappeared (Fig.5E, Fig. 5F, RM one-way ANOVA, $P=0.025$, $F(2.960, 14.80) = 4.182$). Similarly, the mean reward predictions signal for total trials in calcium PTE test also disappeared (Fig.5I). When we analyzed the correlation between the reward prediction signals and the escalating volitional efforts for each trial, we found that the reward prediction signal was negatively correlated with the escalating volitional efforts (Fig.5G, $r^2 = 0.06$, $P = 0.027$). However, there was no correlation between the reward value and escalating volitional efforts (Fig. 5H, $r^2 = 0.01$, $P = 0.34$). Totally, escalating efforts were negatively correlated with dopamine dynamics for reward prediction in NAc but not with the reward value in volitional control of neural activity.**

Lastly, we also analyzed dopamine dynamics in the NAc in response to escalating efforts (with PRT) during motor skills (Fig.6). Fig.6A indicated that the loci for expression GRAB_{DA} sensors in NAc. There were also two dopamine signal peaks in the NAc during motor skills: the prediction signal for the future reward (the black box) and reward value (the purple box) (Fig.6B). **Then we analyzed the “Height” of the dopamine fluorescence signal before or ~~or~~and after reward delivery for FR5, indicating there were significant changes in the prediction component (RM one-way ANOVA, $P=0.044$, $F(2.115, 14.80) = 3.839$) and in the reward component (RM one-way ANOVA, $P=0.027$, $F(2.797, 19.58) = 3.874$). Like volitional control of neural activity, the reward prediction signal and reward value of dopamine fluorescence signal significantly changed between FR5-1 and FR5-5 (Fig.6C, $P=0.049$; Fig.6D, $P=0.01$). The reward prediction signals of dopamine largely disappeared during trial-by-trial PRT testing (Fig.6E, Fig.6F, RM one-way ANOVA, $P=0.046$, $F(2.048, 10.24) = 4.184$). The mean reward predictions signal for total trials also disappeared in calcium PTH test (Fig.6I). Similarly, correlation analyses revealed that dopamine dynamics for the reward prediction was negatively correlated with the escalating behavioral efforts during PRT testing (Fig.6G, $r^2 = 0.06$, $P=0.018$), but there was no correlation between the reward value and escalating behavioral efforts (Fig.6H, $r^2 = 0.01$, $P = 0.56$). **These results also indicated that escalating efforts were negatively correlated with dopamine dynamics for reward prediction in NAc but not with the reward value in motor skills.****

Discussion

The development of the first neural representation and quantitative

assessment of volitional motivation

In this study, by leveraging the causal link between neuron activity and criteria set by the experimenter, we have adopted the concept of behavioral motivation in PRT and PHD to develop Calcium PTE and Calcium PTH tests which allow us to directly link the calcium signal (neural activity) to the escalating effort-related motivation (i.e., a subject is willing to expend to earn a reward) during volitional conditioning of neural activity. This quantitative analysis of volitional motivation by Calcium PTE and Calcium PTH permit determination of individual variations in volitional motivation at the neural level. The validity of these calcium-based PTE and PTH analyses for quantification of the volitional motivation is supported by the chemogenetic finding that the activation of the striatopallidal pathway inhibited the motivation in the neuroprosthetic control, in agreement with the previous study on behavioral motivation [25, 26].

The development of the first neural representation and quantitative method for volitional motivation enables opportunities to address the specific contribution of the neural circuit and neuromodulator for BMI improvement. From the perspective of human subjects, the assessment and training of cognitive impairments in advanced stages of paralysis represent a challenge as the standard assessment of motivation typically involves a motor response. However, some or all motor abilities are lost in stroke patients and in other cases of severe motor loss[27]. Therefore, quantitative analysis of motivation at the neuron level in disabled patients may lead to a new therapeutic approach to enhance motivation during neurorehabilitation.

Dopamine dynamics in the NAc reflect cue-triggered “wanting” not escalating efforts

Motivation and reinforcement learning has been classically associated with dopamine neurons in the VTA that predominantly project to the NAc [28]. The critical role of the dopamine reward system in motivational control of behaviors is supported by the finding that disrupting dopamine transmission by pharmacological and neurotoxic approaches regulates response vigor [9, 29] and work output [30]. Animals with impaired dopamine transmission reallocate their instrumental behavior away from food-reinforced tasks with high response requirements, and instead select less effortful food-seeking behaviors [31, 32]. The instrumental output and effort-related choice impaired by dopamine D₂ antagonism were reversed by A_{2A}R blockade or genetic deletion[33-37]. Indeed, dopamine dynamics in the NAc encode the reward prediction error [38-40] [41], efforts and delay-related costs [42] and modulate rewards through delays conferred by the escalating costs [43]. Consistent with the prediction error signal[44, 45], we detected the development of prediction signal (i.e., calcium fluorescence signal

dynamics underlies neuroprosthetic learning. Moreover, reaching proficient control with cohesive neural firing patterns [57-60] similarly requires reinforcement learning with a lot of repetitive training to produce stable representation mapping [61, 62]. Our study further confirms that both behavioral and volitional conditionings are driven by motivational factors with similar modulation at the neural circuit and the neurotransmitter levels. Indeed, we found that chemogenetic activation of the striatopallidal neurons similarly suppress volitional motivation (as evident by reduced breakpoint of calcium PTE) and behavioral motivation [17, 20-22]. Collectively, these above-mentioned studies together with ours suggest that the similarities between volitionally controlled neural activity and control of motor behaviors far outweigh their differences.

Conclusions

We firstly developed the novel methods for detecting volitional motivation by the representation of the M1 population neural activity in respond to progressively escalating efforts. Meanwhile, we further verified volitional control of population neural activity shared brain structures and learning mechanisms including motivational control with sensorimotor learning.

Data availability statement

Data will be made available upon reasonable request to the corresponding author.

Funding

This work was supported by Zhejiang Provincial Natural Science Foundation Grant Nos. LY22C090005(to LPZ), LQ19C090005 (to QW), LQ18C090002 (to ZMY).

Acknowledgments

The authors thank Shaomin Zhang and Yi Zhang for commenting on earlier versions of the manuscript.

Compliance with Ethics Requirements

All Institutional and National Guidelines for the care and use of mice were followed.

Declaration of Competing Interest

The authors declare that they have no known competing financial interests or personal relationships that could have appeared to influence the work reported in this paper.

Author contributions

Liping Zhang designed the experiments. Jiang-Fan Chen and Liping Zhang conceptualized the project. Chengwei Liu performed most of the experiments. Xiaopeng Zhou, Hui Zhou and Shengtao Luo provided experimental facilities, administrative assistance, the surgery implanting and Immunohistochemistry. Liping Zhang, Qin Wang and Zhimo Yao analyzed the data and acquired the funding. Jiang-Fan Chen and Liping Zhang wrote and revised the paper. All authors have read and approved the final version of this manuscript. All persons designated as authors qualify for authorship, and all those who qualify for authorship are listed.

Figures

Fig.1. Development of calcium PTE for detecting volitional motivation.

A) Close-loop volitional control system. The calcium fluorescence signal ($\Delta F/F$) of M1 neurons was monitored in real-time by a fiber photometry system. Calcium fluorescence signals ($\Delta F/F$) exceeding the defined threshold value triggered the operant box to deliver a drop of sucrose solution reward. B) Scheme of the training procedure for calcium PTE. The upper panel indicated the scheme of the training procedure for calcium PTE. The lower panel indicated the number of TCEs of the sequential trial for calcium PTE test. C) The calcium fluorescence signal change in M1 neurons before/after the reward delivery (± 5 s) for escalating efforts (trial 1, 3, 5, 7, 9, 11, 13) in Calcium PTE testing (n = 6). D) The breakpoint (the maximal TCEs) distribution of six mice by calcium PTE test (n = 6).

Fig. 2. Development of calcium PTH for detecting volitional motivation.

A) Scheme of the training procedure for calcium PTH. The upper panel indicated the scheme of the training procedure for calcium PTH. The lower panel indicated the holding time of the sequential trial for calcium PTH test. B) The calcium fluorescence signal change in M1 neurons before/after the reward delivery (± 5 s) for escalating efforts (trial 1,3, 5, 7, 9, 11, 13,15) in calcium PTH testing($n=6$). C) The breakpoint (the maximal holding time) distribution of six mice by calcium PTH test($n=6$).

Fig. 3. Striatopallidal pathway regulate the volitional motivation

A) Sagittal whole-brain expression pattern of A_{2A}-rM3Ds mice (Str: Striatum; LGP: lateral globus pallidus). B) A_{2A}-rM3Ds (red) was co-localized with A_{2A} receptor (green) in the striatonigral neurons. Scale bar: 50 μm C) A_{2A}-rM3Ds (red) was not co-localized with D1 receptor (green) in the striatonigral neurons. Scale bar: 50 μm. D) The breakpoint distribution of calcium PTE test in individual mice for CNO (blue) and vehicle-treated groups (red) (n=12: CNO group=6 and vehicle group=6). E) The chemogenetic activation of the striatopallidal pathway impaired the breakpoint in calcium PTE task (E; unpaired t-test, P = 0.026, t = 2.609, df = 10). F) The mean calcium fluorescence signal change in M1 neurons before (+5s) and after (+5s) the reward delivery (0) for calcium PTE in CNO (purple) and vehicle-treated groups (red). G-I) Calcium PTH under the same condition as A-C (H; unpaired t-test, P = 0.047, t = 2.257, df = 10).

Fig. 4. Adenosine $A_{2A}Rs$ regulate the volitional motivation

A) The blockade of adenosine $A_{2A}Rs$ enhanced motivation in volitional control ($n=12$: KW6002 group=6 and vehicle group=6). The breakpoint distribution of calcium PTE testing in individual mice for the KW6002(blue) and vehicle-treated groups (red)(A). B) Blockade of adenosine $A_{2A}Rs$ improved the breakpoint in calcium PTE (B; Mann-Whitney U-test, $P=0.009$). C) The mean calcium fluorescence signal change in M1 neurons before (+5s) and after (+5s) the reward delivery (0) for calcium PTE in KW6002(purple) and vehicle-treated groups (red). D-F) Calcium PTH under the same conditions as A-C (E; unpair t-test, $P=0.004$, $t=3.699$, $df=10$).

Fig. 5. Analysis of the dopamine dynamics in the NAc for calcium PTE.

A) The illustration of GCaMP6f expression loci in M1(left) and GRAB_{DA} sensors expression loci in NAc (right). (n = 6). The expression of GCaMP6f in M1 only employed for the volitionally controlled neural task. B) The mean dopamine dynamics changes ($\Delta F/F$) before (10 s) and after (5 s) the reward presentation for FR5-1 and FR5-5 training. C) The mean “height” before (-5) the reward presentation is higher in FR5-5 training (P = 0.029). D) The mean “height” after the reward (+5) presentation is lower in FR5-5 (P = 0.009). E) The mean dopamine dynamic changes before (10 s) and after (5 s) the reward presentation in calcium PTE test for last 13 trials (aligned to the last trial). F) The mean “height” for per trial before (5 s) the reward delivery in calcium PTE test (RM one-way ANOVA, P=0.025, F (2.960, 14.80) = 4.182). G-I) Correlation analysis of the dopamine dynamics for the reward prediction (F, P = 0.027 , $r^2 = 0.06$) and reward value (G, P = 0.34 , $r^2 = 0.01$) with the volitional efforts in calcium PTE. H) The mean dopamine dynamic changes before (10 s) and after (5 s) the reward delivery in calcium PTE test. “0” represents the reward delivery. “height” represents the highest peak of dopamine dynamic of ± 5 s of reward delivery. The black box indicated the prediction signal, the purple box indicated the reward value signal.

Fig. 6. Analysis of dopamine dynamics in the NAc for PRT.

A) The illustration of GRAB_{DA} sensors expression and the fluorescence signal observation loci in NAc (n=8). B) The mean dopamine dynamics changes ($\Delta F/F$) before (10 s) and after (5 s) the reward presentation for FR5-1 and FR5-5 training. C) The mean “height” before (-5) the reward presentation is higher in FR5-5 training ($P=0.049$). D) The mean “height” after the reward (+5) presentation is lower in FR5-5 (pair t-test, $P=0.01$). E) The mean dopamine dynamic changes before (10 s) and after (5 s) the reward presentation in PRT testing for last 9 trials (aligned to the last trial). F) The mean “height” for per trial before (5 s) the reward delivery in PRT test (RM one-way ANOVA, $P=0.046$, $F(2.048, 10.24) = 4.184$). F-G) Correlation analysis of the dopamine dynamics for the reward prediction (F, $P=0.018$, $r^2=0.06$) and reward value (G, $P=0.56$, $r^2=0.01$) with the behavioral efforts in PRT. H) The mean dopamine dynamic changes before (10 s) and after (5 s) the reward delivery in PRT test. “0” represents the reward delivery. “height” represents the highest peak of dopamine dynamic of ± 5 s of reward delivery. The black box indicated the prediction signal, the purple box indicated the reward value signal.

Supplemental fig. 1 Analysis of dopamine dynamics before (2 s) and after (20 s) pressure lever in NAc. Reward delivery was delayed for 10s after pressure lever (Mouse 1, 2, 3).

- neurons in the dorsal raphe nucleus encode reward signals. *Nat Commun* 2016, 7:10503.
20. Eduardo F Gallo, Jozsef Meszaros, Jeremy D Sherman, Muhammad O Chohan, Eric Teboul, Claire S Choi, Holly Moore, Jonathan A Javitch, Kellendonk C: **Accumbens dopamine D2 receptors increase motivation by decreasing inhibitory transmission to the ventral pallidum.** *Nat Commun* 2018, 9:1086.
 21. Soares-Cunha C, Coimbra B, Domingues AV, Vasconcelos N, Sousa N, Rodrigues AJ: **Nucleus Accumbens Microcircuit Underlying D2-MSN-Driven Increase in Motivation. LID - ENEURO.0386-18.2018 [pii] LID - 10.1523/ENEURO.0386-18.2018 [doi].** *eNeuro* 2018, 5:0386-0318.
 22. Gallo EF, Meszaros J, Sherman JD, Chohan MOA-Ohoo, Teboul E, Choi CS, Moore H, Javitch JAA-Ohoo, Kellendonk C: **Accumbens dopamine D2 receptors increase motivation by decreasing inhibitory transmission to the ventral pallidum.** *Pharmacol Rev* 2018, 70:747-762.
 23. Mohebi A, Pettibone JR, Hamid AA, Wong JT, Vinson LT, Patriarchi T, Tian L, Kennedy RT, Berke JD: **Dissociable dopamine dynamics for learning and motivation.** *Nature* 2019, 570:65-70.
 24. Sun F, Zeng J, Jing M, Zhou J, Feng J, Owen SF, Luo Y, Li F, Wang H, Yamaguchi T, et al: **A Genetically Encoded Fluorescent Sensor Enables Rapid and Specific Detection of Dopamine in Flies, Fish, and Mice.** *Cell* 2018, 174:481-496 e419.
 25. Gallo EF, Meszaros J, Sherman JD, Chohan MO, Teboul E, Choi CS, Moore H, Javitch JA, Kellendonk C: **Accumbens dopamine D2 receptors increase motivation by decreasing inhibitory transmission to the ventral pallidum.** *Nat Commun* 2018, 9:1086.
 26. Ruder L, Schina R, Kanodia H, Valencia-Garcia S, Pivetta C, Arber S: **A functional map for diverse forelimb actions within brainstem circuitry.** *Nature* 2021.
 27. Carelli L, Solca F, Faini A, Meriggi P, Sangalli D, Cipresso P, Riva G, Ticozzi N, Ciammola A, Silani V, Poletti B: **Brain-Computer Interface for Clinical Purposes: Cognitive Assessment and Rehabilitation.** *BioMed Research International* 2017, 2017:1-11.
 28. Volkow ND, Wise RA, Baler R: **The dopamine motive system: implications for drug and food addiction.** *Nat Rev Neurosci* 2017, 18:741-752.
 29. Hamid AA, Pettibone JR, Mabrouk OS, Hetrick VL, Schmidt R, Vander Weele CM, Kennedy RT, Aragona BJ, Berke JD: **Mesolimbic dopamine signals the value of work.** *Nat Neurosci* 2016, 19:117-126.
 30. Salamone JD, Correa M, Ferrigno S, Yang JH, Rotolo RA, Presby RE: **The Psychopharmacology of Effort-Related Decision Making: Dopamine, Adenosine, and Insights into the Neurochemistry of Motivation.** *Pharmacol Rev* 2018, 70:747-762.
 31. Nunes EJ, Randall PA, Podurriel S, Correa M, Salamone JD: **Nucleus accumbens neurotransmission and effort-related choice behavior in food motivation: effects of drugs acting on dopamine, adenosine, and muscarinic acetylcholine receptors.** *Neurosci Biobehav Rev* 2013, 37:2015-2025.
 32. Salamone JD, Wisniecki A, Carlson BB, M. C: **Nucleus accumbens dopamine depletions make animals highly sensitive to high fixed ratio requirements but do not impair primary food reinforcement.** *Neuroscience* 2001, 105:863-870.
 33. Collins LE, Sager TN, Sams AG, Pennarola A, Port RG, Shahriari M, Salamone JD: **The novel adenosine A2A antagonist Lu AA47070 reverses the motor and motivational effects**

- produced by dopamine D2 receptor blockade. *Pharmacol Biochem Behav* 2012, **100**:498-505.
34. Farrar AM, Pereira M, Velasco F, Hockemeyer J, Muller CE, Salamone JD: **Adenosine A(2A) receptor antagonism reverses the effects of dopamine receptor antagonism on instrumental output and effort-related choice in the rat: implications for studies of psychomotor slowing.** *Psychopharmacology (Berl)* 2007, **191**:579-586.
 35. Mott AM, Nunes EJ, Collins LE, Port RG, Sink KS, Hockemeyer J, Muller CE, Salamone JD: **The adenosine A2A antagonist MSX-3 reverses the effects of the dopamine antagonist haloperidol on effort-related decision making in a T-maze cost/benefit procedure.** *Psychopharmacology (Berl)* 2009, **204**:103-112.
 36. Pardo M, Lopez-Cruz L, Valverde O, Ledent C, Baqi Y, Muller CE, Salamone JD, Correa M: **Adenosine A2A receptor antagonism and genetic deletion attenuate the effects of dopamine D2 antagonism on effort-based decision making in mice.** *Neuropharmacology* 2012, **62**:2068-2077.
 37. Salamone JD, Farrar AM, Font L, Patel V, Schlar DE, Nunes EJ, Collins LE, Sager TN: **Differential actions of adenosine A1 and A2A antagonists on the effort-related effects of dopamine D2 antagonism.** *Behav Brain Res* 2009, **201**:216-222.
 38. Schultz W: **Dopamine reward prediction-error signalling: a two-component response.** *Nat Rev Neurosci* 2016, **17**:183-195.
 39. Schultz W: **Reward functions of the basal ganglia.** *J Neural Transm (Vienna)* 2016, **123**:679-693.
 40. Schultz W: **Dopamine Reward Prediction Error Coding** *Dialogues Clin Neurosci* 2016, **18**:23-32.
 41. du Hoffmann J, Nicola SM: **Dopamine invigorates reward seeking by promoting cue-evoked excitation in the nucleus accumbens.** *J Neurosci* 2014, **34**:14349-14364.
 42. Day JJ, Jones JL, Wightman RM, Carelli RM: **Phasic nucleus accumbens dopamine release encodes effort- and delay-related costs.** *Biol Psychiatry* 2010, **68**:306-309.
 43. Wanat MJ, Kuhn CM, Phillips PE: **Delays conferred by escalating costs modulate dopamine release to rewards but not their predictors.** *J Neurosci* 2010, **30**:12020-12027.
 44. An J, Yadav T, Hessburg JP, Francis JT: **Reward Expectation Modulates Local Field Potentials, Spiking Activity and Spike-Field Coherence in the Primary Motor Cortex.** *eNeuro* 2019, **6**.
 45. Yao Z, Hessburg JP, Francis JT: **Normalization by valence and motivational intensity in the sensorimotor cortices (PMD, M1, and S1).** *Sci Rep* 2021, **11**:24221.
 46. Wyvell CL, Berridge KC: **Intra-accumbens amphetamine increases the conditioned incentive salience of sucrose reward: enhancement of reward "wanting" without enhanced "liking" or response reinforcement.** *J Neurosci* 2000, **20**:8122-8130.
 47. Marsh BT, Tarigoppula VS, Chen C, Francis JT: **Toward an autonomous brain machine interface: integrating sensorimotor reward modulation and reinforcement learning.** *J Neurosci* 2015, **35**:7374-7387.
 48. Ramkumar P, Dekleva B, Cooler S, Miller L, Kording K: **Premotor and Motor Cortices Encode Reward.** *PLoS One* 2016, **11**:e0160851.
 49. Ramakrishnan A, Byun YW, Rand K, Pedersen CE, Lebedev MA, Nicolelis MAL: **Cortical neurons multiplex reward-related signals along with sensory and motor information.** *Proc Natl Acad Sci U S A* 2017, **114**:E4841-E4850.

50. Zhao Y, Hessburg JP, Asok Kumar JN, Francis JT: **Paradigm Shift in Sensorimotor Control Research and Brain Machine Interface Control: The Influence of Context on Sensorimotor Representations.** *Front Neurosci* 2018, **12**:579.
51. Dangi S OA, Moorman HG, Carmena JM: **Design and analysis of closed-loop decoder adaptation algorithms for brain-machine interfaces.** *Neural Comput* 2013, **25**:1693-1731.
52. Chapin JK, Moxon KA, Markowitz RS, Nicolelis MA: **Real-time control of a robot arm using simultaneously recorded neurons in the motor cortex.** *Nat Neurosci* 1999, **2**:664-670.
53. Hira R, Ohkubo F, Ozawa K, Isomura Y, Kitamura K, Kano M, Kasai H, Matsuzaki M: **Spatiotemporal dynamics of functional clusters of neurons in the mouse motor cortex during a voluntary movement.** *J Neurosci* 2013, **33**:1377-1390.
54. Lopez-Huerta VG, Denton JA, Nakano Y, Jaidar O, Garcia-Munoz M, Arbutnott GW: **Striatal bilateral control of skilled forelimb movement.** *Cell Rep* 2021, **34**:108651.
55. Li Y, Pan X, He Y, Ruan Y, Huang L, Zhou Y, Hou Z, He C, Wang Z, Zhang X, Chen JF: **Pharmacological Blockade of Adenosine A2A but Not A1 Receptors Enhances Goal-Directed Valuation in Satiety-Based Instrumental Behavior.** *Front Pharmacol* 2018, **9**:393.
56. Hwang EJ, Bailey PM, Andersen RA: **Volitional control of neural activity relies on the natural motor repertoire.** *Curr Biol* 2013, **23**:353-361.
57. Koralek AC, Costa RM, Carmena JM: **Temporally precise cell-specific coherence develops in corticostriatal networks during learning.** *Neuron* 2013, **79**:865-872.
58. Koralek AC, Jin X, Long JD, 2nd, Costa RM, Carmena JM: **Corticostriatal plasticity is necessary for learning intentional neuroprosthetic skills.** *Nature* 2012, **483**:331-335.
59. Marchesotti S, Martuzzi R, Schurger A, Blefari ML, Del Millan JR, Bleuler H, Blanke O: **Cortical and subcortical mechanisms of brain-machine interfaces.** *Hum Brain Mapp* 2017, **38**:2971-2989.
60. Neely RM, Koralek AC, Athalye VR, Costa RM, JM. C: **Volitional Modulation of Primary Visual Cortex Activity Requires the Basal Ganglia.** *Neuron* 2018, **97**:1356-1368.
61. Athalye VR SF, Carmena JM, Costa RM.: **Evidence for a neural law of effect.** *Science* 2018, **359**:1024-1029.
62. Pohlmeier EA, Mahmoudi B, Geng S, Prins NW, Sanchez JC: **Using reinforcement learning to provide stable brain-machine interface control despite neural input reorganization.** *PLoS One* 2014, **9**:e87253.

Oct. 18, 2022

Re: JP-RP-2022-283915 “**Neural Representation and Modulation of Volitional Motivation in Response to Escalating efforts**”

Dear Editor:

We thank the Editors and Reviewers very much for the constructive comments and suggestions and for the opportunity to revise and improve the manuscript. As suggested, we have performed additional analyses and revised the manuscript in response to the editors and the Reviewer's critical comments and their suggestions. The point-by-point response to the editors and reviewer's comments are provided as following:

Reviewing Editor:

The manuscript designs a novel behavioral paradigm to assess volitional motivation and reports a role of striatopallidal pathway and adenosine A2A receptor in motivation. Although both reviewers think the manuscript is interesting, it need a substantial revision according to the reviewers' comments, especially demonstrating the relationship between the calcium PTE/PTH in the M1 and behavioral action or volitional motivation, explaining the criterion for behavioral procedures, checking the statistics, and extensively revising sentences.

Besides, it would be better if the authors will clarify the following issues:

1. The activity of the M1 is generally considered to be related to behavior execution. The manuscript need to clarify whether the calcium signal change in the PTE and PTH test was related to movement or motivation.

Response: We thank the editor’s comments. Indeed, M1 neuronal activity is generally associated with behavior execution, but also with reward anticipation, motivation and motor planning [1-6]. In the previously study [7], we have partially verified the calcium fluorescence signal in M1 for operant volitionally controlled task was not related to overt movement. (1) we have shown the temporal disassociation of the volitional control of M1 neural activity from movements of the right forelimb as monitored with electromyographic (EMG) recordings (below Figure). (2) the mice did not cross the defined threshold during free movement and foraging. (3) the M1 population calcium fluorescence signal in the operant (motor) behavior displayed the different pattern with volitional control of neural activity. Our view of M1 calcium fluorescence signal representation of volitional control is consistent with as the previous findings that arbitrarily selected primary motor cortex (M1) neurons for volitional control have little relationship with native limb movement [9] and that a stable M1 cortical representation for prosthetic function can be stored [8] . Taken together, we believe that M1 calcium fluorescence signals are representation for volitional signal with limited relationship with movement.

The development of Calcium PTE and Calcium PTH based on the operant volitionally controlled task coupled with the concept of representing behavioral motivation by the break-point in response to escalating efforts. Thus, the calcium signal change in M1 for Calcium PTE and Calcium PTH also was not related to the overt movement, but to the volitional signal. In the current study, we progressively increased M1 neural activity with a series of pre-set criterion to progressively escalate volitional efforts and estimated the motivation by maximal efforts via breakpoints in the tasks. Indeed, there were progressively increase in the TCE and PTH as defined by the formula, representing the escalating efforts (and thus the volitional motivation). Therefore, the maximal neural activities (calcium fluorescence signal, either in TCE or PTH) is related to volitional motivation (not movement).

Figure, The population calcium signal was dissociated with the EMG signal (Left) as the distribution of correlation coefficients between EMG activity and M1 population calcium signal changes for all trials in all session across the neuroprosthetic learning was not significantly different from zero (right) [7].

2. Whether A₂AR is specifically expressed in the striatopallidal neurons, which determines whether the chemogenetic and pharmacological manipulations are specific.

Response: We thank the editor's comments. A_{2A} receptors are predominantly expressed in the striatopallidal neurons and are highly relevant to the function of the indirect pathway of the striatum [10]. Consistent with the previous study with adora2A-rM3Ds mice [11], we confirmed that rM3Ds was specifically expressed in the striatopallidal pathway (i.e. striatal neurons (Fig. A, Str: Striatum) and striatopallidal projections) (Fig. A, LGP: lateral globus pallidus) and were colocalized with A_{2A}Rs in the striatopallidal neurons (Figure B, rM3Ds: red, A_{2A}R: green), but not with dopamine D1 receptors in the striatonigral neurons (Fig. C, rM3Ds: red, D1R: green). These results confirm that chemogenetic (dM3Ds) and pharmacological (KW6002) manipulations are specific to the striatopallidal pathway.

3. Figure 2B does not clearly reflect that the holding time of the calcium signal above the threshold gradually increases with the progressively increase efforts. The holding time above the calcium fluorescence threshold across trials should be counted in the PTH test.

Response: We thank the editor's comments on this point. By adapting the break-point for representing "volitional motivation" in response to the escalating effort, we defined every trial's holding time using the formula in Calcium PTH ($\text{holding time} = 0.1 * 1.05^{(t-1)}$, $t = \text{trial number}$). In this schedule, the holding time for per trial was achieved by animal only once, and as such the holding time above the calcium fluorescence threshold across trials can not be accounted. Moreover, the increase of holding time is mainly reflected in the width of calcium fluorescence signal. We also tried to analyze the width of calcium fluorescence signal for per trial in Calcium PTH. However, once the set holding time by the formula is reached in the experimental process, the reward will be given and the trial will be terminated. In addition, each trial is relatively an independent experiment (holding time is different for per trial) and no stable calcium fluorescence signal will be generated. Thus, the width of calcium signal of each trial is not invariably consistent with the set holding time. However, the neural activity did not represent volitional motivation and the maximal neural activities (calcium fluorescence signal, either in TCE or holding time) is related to volitional motivation in the current study.

Senior Editor:

In the event that you choose to resubmit a new version of the manuscript, I would ask that you first pay particular attention to the comments that have been provided in relation to the statistical analyses. In particular, it should be demonstrated that all assumptions pertaining to the use of parametric analyses have been satisfied. If this is not the case, non-parametric procedures would be used instead. Given that multiple tests were performed, appropriately stringent tests should be applied to account for potential inflation of the effective alpha level.

Response: We thank the Senior Editor for the comment. We have paid special attention to the statistical analyses. The data in Figure 3B, 3E, 4E were tested and shown to be normally distributed and accordingly we have used the unpaired for data analyses. The data in Figure 4B were tested and found to be not normally distributed, and accordingly we have used the non-parametric Mann-Whitney U test for data analysis. We analyzed the data in Figure 5C, 5D, 5F, 6C, 6D, 6F by RM one-way ANOVA($P < 0.05$), followed by post-hoc comparison with LSD test.

REFEREE COMMENTS

Referee #1:

Title: Neural Representation and Modulation of Volitional Motivation in Response to Escalating efforts

Summary: In this article, the authors present data from mice expressing calcium indicator dyes while recording changes in fluorescence within the primary motor cortex (M1) or the nucleus accumbens. The authors aimed to determine if the amount of effort the mice had to expend to obtain a given reward amount would change as the effort per unit reward increased in a predictable manner, and how modulation of the striatum changes motivational aspects of the subjects as seen via breakpoints in the tasks.

Specific questions and issues:

1. The summary (the first two figures without legends) figure seems to show the rat using its face to press the lever in yellow. Is this in fact what was done? If not, please make the figure more representative of the real situation. In addition, the Volitional motivation figure is also a bit unclear as to what is being represented as compared to the actual experiment.

Response: We thank the reviewer's comments. The mice used the right forelimb to press the lever to receive the reward and we have revised the first Figure accordingly. The first figure was graphical abstract, illustrating how to assess volitional motivation in the current study (the below figure).

2. There are no figure legends for the first two figures shown in the combined .pdf, I'm not sure if these were to be in the supplemental information or somewhere else in the paper?

Response: We thank the reviewer's comments. The first figure in the combined PDF file was graphical abstract and thus no figure legend was included. The second figure was the Figure 5 in the manuscript and have deleted the second figure in the combined PDF file and the figure legend to Figure 5 was included in the manuscript.

3. The below sentence is rather difficult to follow. There are many grammatical errors in the text making some ideas rather hard to follow. Perhaps using grammar checking software could help with this, or a native American English speaker.

Response: We thank the reviewer's comments and have revised the grammatical errors in the text.

a. "The first quantitative assessment of volitional motivation by progressively representation of the M1 neural activity"

Response: We thank the reviewer's comments and have revised this sentences as "Volitional motivation was quantitatively evaluated by the M1 neural activity in response to progressively escalating volitional efforts."

4. I'm not fully sure I follow the below sentence, please revise.

a. "The volitional control of neural activity directly reinforces the target neurons using real-time biofeedback and is driven by motivational factor (volitional motivation)."

Response: We thank the reviewer's comments and have revised the sentence as following: "The volitional control of neural activity is driven by a motivational factor (volitional motivation) which directly reinforces the target neurons via real-time biofeedback ."

5. Please summarize the previous studies relevant information on this point here. "Mice underwent volitional neural learning for 10 days as described previously [16]".

Response: We thank the reviewer's comments and have summarized the relevant information from the previous study as following: "Mice were transfected with AAV9-syn-GCaMP6f-WPRE-SV40 to express the genetically encoded Ca^{2+} indicator GCaMP6f in the M1 cortex and the calcium fluorescence signal was monitored by fiber photometry system [18]. Mice were trained to perform the volitionally controlled neural task to reach the correct percentage of 85-100% for obtaining the reward (Fig.1A)."

6. Again, you need to at least give the reader the information needed to judge and understand your current work, so, please summarize the pertinent information here as well. If the information in the following sentence is that description, please make this clear such as saying we briefly summarize this information below etc. "After smoothing the data with a moving average filter (20 ms span), the calcium fluorescence signal and dopamine fluorescence signal analysis for the event-related behavior is described in previous research [16]."

Response: We thank the reviewer's comments and have summarize the previous studies relevant information as "As in the our previous study, we performed data analysis in MatLab platerform (Math Works, Natick, USA) with custom-written programs [18]. After smoothing the data with a moving average filter (20 ms span), we analyzed the event-related calcium fluorescence signal and dopamine fluorescence signal in relationship with the reward (with the reward as time "0" point)"

7. Is this baseline the same as the aforementioned "low baseline procedure"? "We derived the values of fluorescence change ($\Delta F/F$) by calculating $(F-F_0)/F_0$, where F_0 is the baseline fluorescence signal averaged over a 1-2s-long control time window, which was typically set 1-2 s preceding the trigger events." If they are not the same perhaps make this clearer.

Response: We thank the reviewer's comments and apologize for any confusion between these two different expressions of the baseline. (a) In the "low baseline procedure" (on-line analysis), the baseline was defined as the lowest F_0 value within 1 min time window and recalculated for every minute using the lowest F_0 value during the volitionally controlled neural task. Therefore, the baseline adjustment in the low baseline procedure is an online real-time adjustment throughout the training process. (b) In the baseline calcium signal for the event-related behavior (off line-analysis), the calcium fluorescence signal for the event-related behavior is offline analysis, where the baseline was typically set 1-2 s preceding the trigger

events (reward delivery). We have revised “low baseline procedure” and the baseline calcium signal for the event-related behavior in the manuscript.

8. In the above what are the trigger events, as this term is not used elsewhere in the paper?

Response: We thank the reviewer’s comments and have revised “the trigger events” as “reward delivery”.

9. Statistics: It seems from many of the figures that the variance of the two populations is not similar, which is a violation of the assumptions made for using the unpaired t-test. In addition, it is not indicated that a test for normality was conducted. A non-parametric test, such as the Mann-Whitney U-test may be more suited for this data.

Response: We thank the reviewer’s comments on the statistical analyses and have carefully analyzed the data distribution. The data in Figure 3B, 3E and 4E were tested and shown to be normally distributed. Accordingly, we have used parametric analysis (i.e. the unpaired t-test) for these data. The data in Figure 4B were tested and found to be not normally distributed, and accordingly we have used the non-parametric Mann-Whitney U test for data analysis.

10. For the following statement there are several publications showing reward expectation, value, and motivational neural correlates that could be cited in this work. "Consistent with the prediction error signal, we detected the development of prediction signal (i.e., calcium fluorescence signal associated with cue presentation, before the reward) in the repeated FR1→FR5 trials.

Response: We thank the reviewer’s comments and have cited the references b and d (see below) in the revised manuscript: “Consistent with the prediction error signal (a. b. 44-45), we detected the development of prediction signal (i.e., calcium fluorescence signal associated with the cue presentation, before the reward) in the repeated FR1→FR5 trials.” .

a. 1. Marsh, B. T., Tarigoppula, V. S., Chen, C. & Francis, J. T. Toward an autonomous brain machine interface: integrating sensorimotor reward modulation and reinforcement learning. *J Neurosci* 35, 7374-87 (2015).

b. 2. An, J., Yadav, T., Hessburg, J. P. & Francis, J. T. Reward Expectation Modulates Local Field Potentials, Spiking Activity and Spike-Field Coherence in the Primary Motor Cortex. *eNeuro* June 6 2019, (2019).

c. 3. Yao, Z., Hessburg, J. P. & Francis, J. T. Normalization by valence and motivational intensity in the sensorimotor cortices (PMd, M1, and S1). *Sci Rep* 11, 24221 (2021).

d. 4. Ramkumar, P., Dekleva, B., Cooler, S., Miller, L. & Kording, K. Premotor and Motor Cortices Encode Reward. *PLoS One* 11, e0160851 (2016).

e. 5. Ramakrishnan, A. et al. Cortical neurons multiplex reward-related signals along with sensory and motor information. *Proc Natl Acad Sci U S A* 114, E4841-E4850 (2017).

f. Zhao, Y., Hessburg, J. P., Kumar, J. N. A. & Francis, J. T. Paradigm Shift in Sensorimotor Control Research and Brain Machine Interface Control: The Influence of Context on Sensorimotor Representations. *Front Neuroscience* 12, (2018).

11. For this text many of the above refs would be applicable. "indicating that neural activity may represent integrated signals." In addition, I've added a ref where BMI was performed while considering this integrated activity of reward expectation/motivation and movement related neural decoding. Note: Please do not feel that you must use any of the suggested citations, but if not these references then please do include any other pertinent refs that might take their place.

Response: We thank the reviewer’s comments and agree with you. We have cited these refs in the revised manuscript :“The direct control of neural activity in BMIs may be a consequence of the integration of the cortical system, subcortical motivational areas, and neurotransmitter system information, indicating that neural activity may represent integrated signals[a-f, 44, 45, 47-50]”.

12. Baseline window concerns: "fluorescence change ($\Delta F/F$) by calculating $(F-F_0)/F_0$, where F_0 is the baseline fluorescence signal averaged over a 1-2s-long control time window, which was typically set 1-2 s preceding the trigger events." Again, not clear what the trigger event is in this sentence, please make this clear. Secondly, as there may be changes in the baseline with learning, or changes due to the increased effort with time, the ratio between baseline and post-trigger could be changing due to either movement of the baseline's height, the post-trigger height, or both. It would be helpful to see non-baseline corrected, more raw representations of the data.

Response: We thank the reviewer's comments and specified "the trigger events" as "reward delivery". As suggested, we have included a proportion of the raw calcium signal for Calcium PTH. We agreed that the baseline may change with learning, effort and motivation. However, the neurons associated with learning, effort, motivation may activate the specific neurons in the different time. These activated neurons did not have the noticeable effect on the population of neural activity when analyzed random activity. Thus, the event-specific analysis of the calcium signal (e.g. in relationship with the reward) is required to shown specific calcium signal patterns

13. Changes in variance:

14. This sentence is hard to follow please edit it as I'm not sure what you are saying. "The volitional control of neurons directly reinforce the neural activity and efforts of volitional control can be escalated by the changed criteria to continuously increase neural activity or by continuously increase holding time for neural activity."

Response: We thank the reviewer's comments and have revised the sentence as following: "The volitional control of neurons directly reinforces the neural activity by biofeedback. The escalated effort for volitional control can be specifically increased by predefined criteria (schedule) to progressively increase the required holding time for neural activity above the defined threshold".

15. Please explain how the below is not contradictory as it is stated that the threshold for the volitional neural task comes from the behavioral task, but that the neural activity between the two tasks is opposite. Perhaps I'm missing something that you can help me see. "mice were conditioned to increase calcium fluorescence signal in M1 neurons above the defined threshold value"... "The defined threshold was based on averaging M1 neural activities over 6 days of instrumental conditioning (pressure lever)."..."Lastly, the M1 population calcium fluorescence signal in the behavior learning is opposite to volitional control of neural activity learning."

Response: We thank the reviewer's comments and apologized for the misrepresentation here. Actually, we want to express that the M1 population calcium fluorescence signal in the behavior operant motor learning displayed the different patterns with volitional control of neural activity learning (, but not "opposite" patterns as we initially described). We also revised the last sentence as following: "Lastly, the M1 population calcium fluorescence signal in the behavior learning is opposite to volitional control of neural activity learning." As "Lastly, the M1 population calcium fluorescence signal in one-lever operant behavior (press the lever for one time to get a reward in a trial) displayed different patterns compare to volitional control of neural activity.." These different patterns may be interpreted that volitional control of neural activity and operant behavior in M1 may involve different neuronal populations.

16. The below statement seems more pessimistic than it may need to be as you can determine what muscles are activated by the brain region you are recording from and then you would only need to obtain EMG from those muscles I would think. Also, some BMI research uses animals or humans that can't move. "Nevertheless, this is a question common to all BMI studies that is ultimately unanswerable without recordings from every muscle in the body."

Response: We thank the reviewer's comments on this point. We agree with the reviewer that the recording of specific muscle activity from the corresponding brain regions and of the animals that can't move (after local anesthesia) would partially disassociate the motor activity from volitional control. We have deleted the sentence in the revised manuscript. .

17. Author Contributions: Please get rid of all the "or" statements and simply put down what everyone did.

Response: We thank the reviewer's comments and revised the text by deleting "or" statement and write down the specific statement.

18. Do you really mean to say that you, the authors, volitionally controlled the M1 population? "In this study, we volitionally controlled the M1 population neural activity in the operational conditioning by real-time monitoring of calcium fluorescence signal using fiber photometry system" If not please correct this sentence to make it clear who is controlling the M1 population.

Response: We thank the reviewer's comments and the sentence as following: "In this study, mice volitionally controlled the M1 population neural activity in the operational conditioning by real-time monitoring of calcium fluorescence signal using fiber photometry system"

19. Some indication as to how the neurons are activated would be helpful to the reader here, as in what is the mechanism of activation. "Activation of the striatopallidal pathway in A2A-rM3Ds mice was performed by CNO."

Response: We thank the reviewer's comments and have provided the brief description of the chemicogenetic activation of the neurons by A_{2A}R-rM3Ds as following: "The rM3Ds was selectively and stably expressed in striatopallidal neurons in A_{2A}-rM3Ds mice and activation of the striatopallidal pathway in A_{2A}-rM3Ds mice was achieved by systemic injection of CNO which specifically activate rM3Ds in the striatopallidal neurons."

20. There is no indication as to where the GRABDA sensors were obtained from. In some sections it is written as above and in others it is GRABDA , please be consistent and use one or the other of these.

Response: We thank the reviewer's comments and have provided the detailed description for the Method section to clearly state that rAAV-hsyn-DA4.4-WPRE-hGH was obtained from BrainVTA (catalog# PT-1340; Wuhan, China). We revised and used "GRAB_{DA}" consistently throughout the manuscript.

21. I'm not sure I fully follow the logic behind the below two statements. Please explicitly state what you have in mind as to how these statements make sense as I seem to be missing something.

a. "Moreover, this result also indicated that the effect of KW6002 on M1 neural activity did not affect the volitional control of neural activity, because M1 signals were lower in KW6002 group compared to the control group(Fig. 4C, 4F).However, the reduction of calcium fluorescence signal after the reward was more pronounced in the KW6002 group compared to the vehicle group (Fig.4C,4F, n=6), which maybe indicated KW6002 enhanced the sensitivity for the reward."

Response: We thank the reviewer's comments on this point. First, the A_{2A}Rs are expressed at a high level in the striatopallidal neurons and low to moderate levels in cortical neurons. Both the cortical and striatal A_{2A}Rs can exert control of cognition such as working memory as we recently demonstrated[12]. Thus, the direct action of A_{2A}Rs in M1 neurons, or the indirect action at the striatal neurons (with circuit feedback onto

the M1 neurons) can regulate volitional control. As shown in Figure 4C and 4F, when we analyzed the calcium fluorescence signal for successful volitional control trials, KW6002 did not influence the volitional control of M1 neural activity. Thus, we reasoned that KW6002 acted indirectly at the striatal A_{2A}Rs with feedback onto the M1 neurons to regulate volitional control. This notion is consistent with our preliminary analysis indicating that focal genetic knockdown of A_{2A}Rs in DMS also enhanced volitional control of neuroprosthetic learning (unpublished data). However, we have deleted the sentence "However, the reduction of calcium fluorescence signal after the reward was more pronounced in the KW6002 group compared to the vehicle group (Fig.4C,4F, n=6), which maybe indicated KW6002 enhanced the sensitivity for the reward." in the revised manuscript.

22. In the below statements you state (data not shown) and then mention a ref [23], but you never state what the actual outcome was. Please state explicitly what ref 23 and the (data not shown) indicate.

a. "To verify dopamine dynamics for reward value, we delayed the reward delivery for 10s, then phasic dopamine dynamics also delayed for about 10s (data not shown). These findings are consistent with the phasic dopamine dynamics in the NAc during motor skill tasks [23]."

Response: We thank the reviewer's comments on this point. We have provided the data in the revised manuscript as "supplemental Figure 1" and revised as "To verify dopamine dynamics for reward value, we have programmed the time for the reward delivery with delay by 10s. Interestingly, the delayed reward delivery by 10 second was associated with the delayed phasic dopamine dynamics by about 10s (supplemental figure 1)". These findings strongly support that the dopamine neurons fire specifically in response to the reward value."

23. indicated that the dopamine dynamics for reward predictions in the NAc was inversely correlated with motivation, but not with the reward value in motivation test.

Response: We thank the reviewer's comments and revised as "These results also indicated that escalating efforts were also negatively correlated with dopamine dynamics for reward prediction in NAc but not with the reward value in motor skills."

24. Don't animals in general choose the path of least effort when obtaining food. I'm assuming some information is missing in the following statement. ..."but they select the path to food reinforcement that requires less effort [30, 31],"

Response: We thank the reviewer's comments on this point. Indeed, animals in general choose the path of least effort when the reward was same for the both paths. However, the animal faced the choice here between making more effort to obtain more food or making less effort to obtain less food. We have clarified the statement as following: "Animals with impaired dopamine transmission can reallocate their instrumental behavior away from food-reinforced tasks with high response requirements, and instead select less effortful food-seeking behaviors [30, 31]. The instrumental output and effort-related choice impaired by dopamine D₂ antagonism were reversed by A_{2A}R blockade or genetic deletion [32-36]."

25. As the previous sentence uses ref 9 perhaps you could use a more specific ref for this definition of

motivation "Motivation is represented by the rewards of maximal efforts against the costs of an action for its potential benefits [9].

Response: We thank the reviewer's comments and have used a more specific ref [9-11] in the revised manuscript.

(see below)

Figures:

Fig.1 It seems panel C shows -2 seconds to +5 seconds, not +-5 as stated in the legend. "C) The calcium fluorescence signal change before/after the reward delivery (± 5 s) for escalating efforts (trial by trial) in Calcium PTE testing (n = 6)." You may also want to include M1 in the title to make this more clear.

Response: We thank the reviewer's comments and revised as "C) The calcium fluorescence signal change in M1 neurons before/after the reward delivery (+5s) for escalating efforts (trial by trial) in Calcium PTE testing (n = 6)."

Fig.2 Panel A has PTE rather than PTH in the flow chart. This figure shows the +-5s, perhaps you should change Fig.1 to match Fig.2s format.

Response: We have revised the figure as suggested in the manuscript.

Fig.4. C) should read (-5) and after (+5) rather than what is currently written, which is both are +5.

Response: We have revised figure as suggested in the manuscript.

Fig. 5. Is F significant and if so perhaps use the same convention of *, **, *** etc.

Response: We thank the reviewer's comments. We have analyzed the 5 days FR5(Fig,5C, 5D) for volitional control using RM one-way ANOVA and used LSD for post-hoc analysis. These results indicate that there were significant changes in prediction component (RM one-way ANOVA, $P=0.043$, $F(2.333, 11.67) = 4.007$) and in reward component (RM one-way ANOVA, $P=0.035$, $F(2.263, 11.32) = 4.422$). There is significant change between FR5-1 and FR5-5 when LSD test was used(Fig. 5C, $P=0.029$; Fig. 5D, $P=0.009$). We have revised Figure 5C, 5D by adding *, **, *** when there was statistical significance accordingly.

Fig. 6. Same as Fig.5.

Response: We thank the reviewer's comments. We have analyzed the 5 days FR5(Fig,5C, 5D) for motor skills using RM one-way ANOVA and used LSD for post-hoc analysis. These results indicate that there were significant changes in prediction component (RM one-way ANOVA, $P=0.044$, $F(2.115, 14.80) = 3.839$) and in reward component (RM one-way ANOVA, $P=0.027$, $F(2.797, 19.58) = 3.874$). There is significant change between FR5-1 and FR5-5 when LSD test was used(Fig. 6C, $P=0.049$; Fig. 6D, $P=0.01$). We have revised Figure 6C, 6D by adding *, **, *** when there was statistical significance accordingly.

Referee #2:

General comments: The manuscript "Neural Representation and Modulation of Volitional Motivation in Response to Escalating efforts" by Zhang et al. reported a criterion of calcium fluorescence signal in the progressive ratio task and progressive hold-down task, which may consider as a quantitative assessment of progressively escalating efforts and relate to striatopallidal pathway. It is an interesting design. The authors clearly described the procedure and detected calcium signals in the M1 and dopamine dynamics in the NAc during the behaviors. But the details of the experiments are less compelling. Below are some comments that the authors may need to consider to improve from the current version:

Major concerns:

1. In general, the authors tried to use the calcium PTE and PTH in the M1 to detect volitional motivation. However, it is still hard to identify if the calcium PTE and PTH in the M1 are specifically coding for behavioral action or for volitional motivation. It would be better to know what is the frequency of calcium based threshold-crossing events (TCE) with 30s interval :1) during the pavlovian training stage (auditory cue pairs with the reward); 2) during the instrumental training stage (press the lever); 3) during the palvovian-instrumental transfer (PIT). These patterns of calcium-TCE would give us a clue that the coding pattern of the calcium signals in the goal directed actions. To my understanding, the volitional motivation should be more related to goal directed action rather than habitual action.

Response: We thank the reviewer's comments. As we outlined in the detailed response to the Senior Editor (see above), we have partially verified in the previously study that the calcium fluorescence signal in M1 for volitionally controlled neural task was not related to overt movement, including the disassociation of M1 calcium signal and EMG activity of the forelimb during volitional conditioning; lack of the crossing-threshold M1 calcium activity during free movement and foraging; and different the M1 population calcium fluorescence signal during operant (motor) and volitional conditioning. Therefore, the calcium fluorescence signal in M1 mainly represents volitional signal for operant volitionally controlled neural task. Furthermore, the development of volitionally controlled neural task involves the instrumental and volitional conditioning procedures. The sound cue was presented as the beginning of the trial and was present throughout the trial. After the cue presentation, animals can only obtain the reward after pressing lever. Therefore, the procedure did not involve Pavlovian conditioning and Pavlovian-to-instrumental transfer procedure. Importantly, we progressively increased M1 neural activity with a series of pre-set criterion to escalate volitional efforts and estimated the motivation by maximal efforts via the breakpoint in the tasks.

2. Figures 1 and 2: Authors should explain why they choose 1, 2, 4, 6, 9, 12, 15...of TCEs as the sequential trial for calcium PTE test. Is it the only effective or optimal procedure for detecting the increased efforts? Similar in the calcium PTH analysis, why did the authors choose a start from 105ms? They should provide the general or average holding time in a single action, or any criterion for these procedures, since the procedure itself could affect calcium signals during different trials.

Response: We thank the reviewer's comments. We adapted the formula ($TCE = 5 * e^{(0.2 * t) - 5}$, $t =$ trial number) for TCE for per trial in analog to the representation of behavioral motivation by the break-point to escalating efforts in the PRT test. Furthermore, we found in the previous study that the average of holding time for the volitional control by the preset threshold was ~ 100 ms. Thus, we choosed the increasing holding time with the starting holding time at 105ms for volitional control.

3. Figures 3 and 4: Could authors explain why CNO manipulation inhibited motor function but not affected the calcium signal in the M1? Does it mean the volitional motivation is different to behavioral motivation or behavioral action?

Response: We thank the reviewer's comments on this point. Consistent with the previous study, CNO-mediated activation of the striatopallidal pathway inhibited motor activity, confirming the inhibitory effect of this pathway on motor activity. However, CNO injection did not affect M1 activity and yet did suppress volitional motivation as evident by the reduced break-point in the PET and PHD test. This suggest that the operant and volitional condition may involve different neural mechanisms (such as involving different neural populations of the parallel cortex-basal ganglia-cortex loop).

4. Figures 5 and 6: Due to the correlation analysis of dopamine dynamics for the reward prediction (Figure 5F and 6F) was quite low (r square = 0.06). It is better to provide the mean "Height" in the trials of PTE and PRT tests as well. Also it would be easy to compare the height of the first trial vs. the height of the last trial from each mouse in the PTE and PRT tests to confirm the conclusion.

Response: We thank the reviewer's comments on this. As suggested, we have analyzed the mean "height" in the trials of the PTE (Figure A) and PRT (Figure B). These results indicate that there were significant changes in the mean "height" in the trials of both the PET (Fig. 5F, RM one-way ANOVA, $P=0.025$, $F(2.960, 14.80) = 4.182$) and PRT (RM one-way ANOVA, $P=0.046$, $F(2.048, 10.24) = 4.184$). We have used the LSD as well as Bonferroni post-hoc comparison (with correction for multiple tests) for post-hoc analysis. The analysis indicated that there was significant decrease between the height of the first trial and the height of the last trial when LSD testing ($p<0.01$) was employed, but the effect was not present when Bonferroni test was employed. Thus, there was apparent decrease in the mean "height" in the trials of the PTE (Figure A) and PRT.

Minor concerns:

1. The real data of location and expression of GCamp6f in M1 and GRABDA sensors in NAc should be shown. The injection site and expression area of the drugs the NAc could affect the behavioral actions sensitively.

Response: We thank the reviewer's comments on this point and have now included the real data showing the location and expression of Gcamp6f in M1 and GRAB_{DA} sensors in NAc in A, B, (Figure 5A, 6A in the revised manuscript).

2. Authors said "To verify dopamine dynamics for reward value, we delayed the reward delivery for 10s, then phasic dopamine dynamics also delayed for about 10s (data not shown)". Please show the data which is important in the study.

Response: We thank the reviewer's comments on this point. As suggested, we have now included this data set (see the figure below) as the supplemental figure 1. As you can see, "the delayed reward delivery by 10 second was associated with the delayed phasic dopamine dynamics by about 10s. These findings strongly support that the dopamine neurons fire specifically in response to the reward delivery." We have included this in the revised Result and Discussion.

We thank the Editors and Reviewers again for their constructive comments and suggestions and the opportunity to revise and improve the manuscript. We hope these new analyses and revision have fully addressed the reviewer and editor's concerns and the manuscript is now considered to be acceptable for publication in "*Journal of Physiology*".

Sincerely yours,
Liping Zhang, PhD
Jiang-Fan Chen, MD PhD

References:

1. Marsh BT, Tarigoppula VS, Chen C, Francis JT: **Toward an autonomous brain machine interface: integrating sensorimotor reward modulation and reinforcement learning.** *J Neurosci* 2015, **35**:7374-7387.
2. Ramkumar P, Dekleva B, Cooler S, Miller L, Kording K: **Premotor and Motor Cortices Encode Reward.** *PLoS One* 2016, **11**:e0160851.
3. Ramakrishnan A, Byun YW, Rand K, Pedersen CE, Lebedev MA, Nicolelis MAL: **Cortical neurons multiplex reward-related signals along with sensory and motor information.** *Proc Natl Acad Sci U S A* 2017, **114**:E4841-E4850.
4. Zhao Y, Hessburg JP, Asok Kumar JN, Francis JT: **Paradigm Shift in Sensorimotor Control Research and Brain Machine Interface Control: The Influence of Context on Sensorimotor Representations.** *Front Neurosci* 2018, **12**:579.
5. An J, Yadav T, Hessburg JP, Francis JT: **Reward Expectation Modulates Local Field Potentials, Spiking Activity and Spike-Field Coherence in the Primary Motor Cortex.** *eNeuro* 2019, **6**.
6. Yao Z, Hessburg JP, Francis JT: **Normalization by valence and motivational intensity in the sensorimotor cortices (PMd, M1, and S1).** *Sci Rep* 2021, **11**:24221.
7. Zhang L, Zhou Y, Liu C, Zheng W, Yao Z, Wang Q, Jin Y, Zhang S, Chen W, Chen JF: **Adenosine A2A receptor blockade improves neuroprosthetic learning by volitional control of population calcium signal in M1 cortical neurons.** *Neuropharmacology* 2020:108250.
8. Karunesh Ganguly JMC: **Emergence of a Stable Cortical Map for Neuroprosthetic Control.** *PLoS Biol* 2009.
9. Law AJ, Rivlis G, Schieber MH: **Rapid acquisition of novel interface control by small ensembles of arbitrarily selected primary motor cortex neurons.** *J Neurophysiol* 2014, **112**:1528-1548.
10. Schiffmann SN, Fisone G, Moresco R, Cunha RA, Ferre S: **Adenosine A2A receptors and basal ganglia physiology.** *Prog Neurobiol* 2007, **83**:277-292.
11. Farrell MS, Pei Y, Wan Y, Yadav PN, Daigle TL, Urban DJ, Lee HM, Sciaky N, Simmons A, Nonneman RJ, et al: **A Galphas DREADD mouse for selective modulation of cAMP production in striatopallidal neurons.** *Neuropsychopharmacology* 2013, **38**:854-862.
12. Zhihui Li XC, Tao Wang, Ying Gao, Fei Li, Long Chen, Jin Xue, Yan He, Yan Li, Wei Guo, Wu Zheng, Liping Zhang, Fenfen Ye,

Xiangpeng Ren, Yue Feng, Piu Chan, Jiang-Fan Chen **The Corticostriatal Adenosine A2A Receptor Controls Maintenance and Retrieval of Spatial Working Memory.** *Biol Psychiatry* 2018, **83**:530-541.

Nov. 21, 2022

Re: JP-RP-2022-283915XR1 “**Neural Representation and Modulation of Volitional Motivation in Response to Escalating efforts**”

Dear Editor:

We thank the Editor and Reviewers very much and have further revised the manuscript in response to the Reviewer's the minor issues. The point-by-point response to reviewer's comments are provided as following:

Reviewing Editor:

The authors have made a substantial and extensive revision. Please further address the minor issues raised by two reviewers.

Response: We thank the editor's comments and have further addressed the minor issues of the reviewers.

REFEREE COMMENTS

Referee #1 (please see attachments):

Thank you for all of the modifications you have made, as they have made a difference in the readability of the paper. I've included a couple more in the attached files that should be taken into consideration.

Response: We thank the reviewer's comments and have made suggested revisions as follows:

1. This distribution does not seem normal and the median may be a better statistic to ask questions about differences from zero.

Response: We thank the reviewer's comments on this point. The EMG distribution in the rebuttal letter has been published in the previous study (Zhang et al. 2020). We will carefully evaluate the EMG distribution data and adopt this suggestion in future experiments.

2. Was this 20 ms bin moved forward by 1 ms or some other number? This level of information should be given.

Response: We thank the reviewer's comments on this point. In the experiment, a moving average filter (20 ms span) was moved at 10 ms per step. We have revised the text as following: “After smoothing the data with a moving average filter (20 ms span with 10 ms moving step), we analyzed the event-related calcium fluorescence signal and dopamine fluorescence signal in relationship with the reward (with the reward as time “0” point)”.

3. It seems from many of the figures that the variance of the two populations is not similar, which is a violation of the assumptions made for using the unpaired t-test.

Response: We thank the reviewer's comments and re-analyzed these data. Although

the variance of the two populations is not similar in the figure, only one data set does not conform to the normal distribution, which we have used non-parameter Mann-Whitney U test (Figure 4B). The other data sets (Figure 3E, 3H and 4E) were conformed to the normal distribution, so these data sets were analyzed by the unpaired t-test.

4. A clear statement of your definition of operant behavior and volitional should be made clear as one could say the BMI control is also operant.

Response: We thank the reviewer's comments on operant behavior. To avoid this confusion, we have used "instrumental" behavior to distinguish volitional (BMI) control. We have revised the manuscript as following: "Lastly, the M1 population calcium fluorescence signal in one-lever instrumental behavior (i.e. by pressing the lever for one time to get a reward in a trial) displayed different patterns compare to volitional control of neural activity."

5. You never defined what LSD is.

Response: We thank the reviewer's comments and defined the LSD as following "The mean "height" of the dopamine fluorescence signals analyzed by one-way ANOVA and followed by post-hoc Fisher's Least Significant Difference (LSD) test."

Finally, we also have revised the manuscript to correct grammatical error according to the reviewer's comments in the text.

Referee #2:

The authors have carefully addressed issues raised by reviewers. However, there remain some minor concerns.

1. Figures 5 and 6: How did the authors calculate the "Height" of fiber photometry signals? Does "height" mean the highest peak value minus baseline? If so, what is used as the baseline? The authors should describe how they analyze fiber photometry data and calculate the "height" in more detail in Method. In figure 6E, it seems the highest peak in trial 9 is higher than that in trial 7. However, the "height" of trial 9 is lower than that of trial 7 in figure 6F. Why? Lastly, is it better to use AUC (area under curve) instead of the highest peak as AUC better describes the change of calcium signals in the selected time window when there is more than one peak? This is rather important because it directly leads to the conclusion that there is negative correlation between the escalating efforts and NAc dopamine signal.

Response: We thank the reviewer's comments on this point. The baseline was defined as the average dopamine fluorescence signal within -5~-6 seconds (immediately prior to the peak time window) before the reward delivery. The "height" represents the highest peak of dopamine dynamics of 0~-5s of reward delivery. We reanalyzed the dopamine fluorescence signal of trial 7 and 9 in 6 mice respectively.

The "height" of trial 7 was indeed higher than that of trial 9 (the below figure). Therefore, the inconsistency between Figure 6E and Figure 6F was due to slightly different alignment of individual calcium signal tracer in Figure 6E. Therefore, we have carefully rechecked all the alignment to ensure its accuracy. We have made corresponding minor changes in the manuscript.

In our experience, the AUC analysis is more suitable for the dopamine fluorescence signals recorded during the continuous slow release of dopamine. Nonetheless, most of the dopamine fluorescence signals in our current study have a major peak (the above picture). Furthermore, we have also performed the AUC analysis for each individual the dopamine fluorescence signals of 9 trials (the below figure), which revealed an overall similar reduction trend, but without no clear pattern ($p=0.0868$ by One-way ANOVA).

2. Page 10: "this result also indicated that the effect of KW6002 on M1 neural activity did not affect the volitional control of neural activity ..." This description is confusing. The activation/inhibition of NAc result in lower/higher breakpoint with little change in M1 activity pattern, which infers that M1 is not in charge of volition control but is controlled directly/indirectly by volition and functions as the final output. KW6002

affected the volitional control because it increased the breakpoints in PTE and PTH tests.

Response: We thank the reviewer's comments. We have revised this sentence as "this result also indicated that KW6002 acted indirectly at the striatal A_{2A}Rs with feedback onto the M1 neurons to regulate volitional control."

3. Page 7: "Primary antibodies used: A2AR (frontier, 1:500), mCherry (Clontech, 1:500), D1 (Clontech, 1:500), goat anti-rabbit AlexaFluor-594(1:250), goat anti-rat AlexaFluor-555(1:250)." Goat anti-rabbit AlexaFluor-594 and goat anti-rat AlexaFluor-555 are secondary antibodies.

Response: We apologized for this error and revised the Method as following: "For immunohistochemistry analysis, we have used the following primary antibodies A2AR (frontier, 1:500), mCherry (Clontech, 1:500), D1 (Clontech, 1:500), together with secondary antibodies goat anti-rabbit AlexaFluor-594(1:250), goat anti-rat AlexaFluor-555(1:250)".

4. Page 3: "Finally, volitional motivation evaluated by neural plasticity in response to progressively escalating efforts with the breakpoints (maximal plasticity of neurons) representing the size of the volitional motivation." Grammatical error.

Response: We thank the reviewer's comments and have revised as "Finally, volitional motivation was evaluated by the response of neuroplasticity to escalating effort, with the breakpoint (maximum plasticity of neurons) representing the size of the volitional motivation".

We hope these new analyses and revision have fully addressed all these additional issues raised by the reviewer and the manuscript is now considered to be acceptable for publication in "*Journal of Physiology*".

Sincerely yours,
Liping Zhang, PhD
Jiang-Fan Chen, MD PhD

Dear Dr Zhang,

Re: JP-RP-2022-283915XR1 "Neural Representation and Modulation of Volitional Motivation in Response to Escalating efforts" by Liping Zhang, Chengwei Liu, Xiaopeng Zhou, Hui Zhou, Shengtao Luo, Qin Wang, Zhimo Yao, and Jiang-Fan Chen

Thank you for submitting your manuscript to The Journal of Physiology. It has been assessed by a Reviewing Editor and by 2 expert referees and we are pleased to tell you that it is acceptable for publication following minor revision.

REVISION CHECKLIST:

We look forward to receiving your revised submission.

Yours sincerely,

Richard Carson
Senior Editor
The Journal of Physiology

REQUIRED ITEMS:

- The contact information provided for the person responsible for 'Research Governance' at your institution is an author on this paper. Please provide an alternative contact who is not an author on this paper or confirm that the author whose email was provided has sole responsibility for research governance. This is the person who is responsible for regulations, principles and standards of good practice in research carried out at the institution, for instance the ethical treatment of animals, the keeping of proper experimental records or the reporting of results.
- You must start the Methods section with a paragraph headed Ethical Approval. A detailed explanation of journal policy and regulations on animal experimentation is given in Principles and standards for reporting animal experiments in The Journal of Physiology and Experimental Physiology by David Grundy J Physiol, 593: 2547-2549. doi:10.1113/JP270818). A checklist outlining these requirements and detailing the information that must be provided in the paper can be found at: <https://physoc.onlinelibrary.wiley.com/hub/animal-experiments>. Authors should confirm in their Methods section that their experiments were carried out according to the guidelines laid down by their institution's animal welfare committee, and conform to the principles and regulations as described in the Editorial by Grundy (2015). The Methods section must contain details of the anaesthetic regime: anaesthetic used, dose and route of administration and method of killing the experimental animals.
- The Reference List must be in Journal format. The references should be alphabetical, not numbered.
- The Journal of Physiology funds authors of provisionally accepted papers to use the premium BioRender site to create high resolution schematic figures. Follow this link and enter your details and the manuscript number to create and download figures. Upload these as the figure files for your revised submission. If you choose not to take up this offer we require figures to be of similar quality and resolution. If you are opting out of this service to authors, state this in the Comments section on the Detailed Information page of the submission form. The link provided should only be used for the purposes of this submission. Authors will be charged for figures created on this premium BioRender account if they are not related to this manuscript submission.
- Papers must comply with the Statistics Policy: https://jp.msubmit.net/cgi-bin/main.plex?form_type=display_requirements#statistics.

In summary:

- If $n \leq 30$, all data points must be plotted in the figure in a way that reveals their range and distribution. A bar graph with data points overlaid, a box and whisker plot or a violin plot (preferably with data points included) are acceptable formats.
- If $n > 30$, then the entire raw dataset must be made available either as supporting information, or hosted on a not-for-profit repository e.g. FigShare, with access details provided in the manuscript.
- 'n' clearly defined (e.g. x cells from y slices in z animals) in the Methods. Authors should be mindful of pseudoreplication.
- All relevant 'n' values must be clearly stated in the main text, figures and tables, and the Statistical Summary Document (required upon revision).
- The most appropriate summary statistic (e.g. mean or median and standard deviation) must be used. Standard Error of the Mean (SEM) alone is not permitted.
- Exact p values must be stated. Authors must not use 'greater than' or 'less than'. Exact p values must be stated to three significant figures even when 'no statistical significance' is claimed.

- Please include an Abstract Figure legend text within the main article file.

- Please include a full title page as part of your article (Word) file (containing title, authors, affiliations, corresponding author name and contact details, keywords, and running title).

EDITOR COMMENTS

Reviewing Editor:

Ethical information, including surgery, anaesthetic monitoring, euthanasia, and terminal procedures, needs to be described in the manuscript.

REFEREE COMMENTS

Referee #1:

My previous concerns have been addressed adequately.

Referee #2:

The authors have addressed my comments satisfactorily. The revised manuscript is ready to be published.

END OF COMMENTS

2nd Confidential Review

22-Nov-2022

Dec. 5, 2022

Re: JP-RP-2022-283915XR2 **“Neural Representation and Modulation of Volitional Motivation in Response to Escalating efforts”**

Dear Editor:

We thank the Editor and Reviewers very much and have started the Methods section with a paragraph headed Ethical Approval including surgery, anaesthetic monitoring, euthanasia, and terminal procedures.

We hope these new revision have fully addressed all these additional issues raised by the editor and the manuscript is now considered to be acceptable for publication in *“Journal of Physiology”*.

Sincerely yours,
Liping Zhang, PhD
Jiang-Fan Chen, MD PhD

Oct. 18, 2022

Re: JP-RP-2022-283915 “**Neural Representation and Modulation of Volitional Motivation in Response to Escalating efforts**”

Dear Editor:

We thank the Editors and Reviewers very much for the constructive comments and suggestions and for the opportunity to revise and improve the manuscript. As suggested, we have performed additional analyses and revised the manuscript in response to the editors and the Reviewer's critical comments and their suggestions. The point-by-point response to the editors and reviewer's comments are provided as following:

Reviewing Editor:

The manuscript designs a novel behavioral paradigm to assess volitional motivation and reports a role of striatopallidal pathway and adenosine A2A receptor in motivation. Although both reviewers think the manuscript is interesting, it need a substantial revision according to the reviewers' comments, especially demonstrating the relationship between the calcium PTE/PTH in the M1 and behavioral action or volitional motivation, explaining the criterion for behavioral procedures, checking the statistics, and extensively revising sentences.

Besides, it would be better if the authors will clarify the following issues:

1. The activity of the M1 is generally considered to be related to behavior execution. The manuscript need to clarify whether the calcium signal change in the PTE and PTH test was related to movement or motivation.

Response: We thank the editor's comments. Indeed, M1 neuronal activity is generally associated with behavior execution, but also with reward anticipation, motivation and motor planning [1-6]. In the previously study [7], we have partially verified the calcium fluorescence signal in M1 for operant volitionally controlled task was not related to overt movement. (1) we have shown the temporal disassociation of the volitional control of M1 neural activity from movements of the right forelimb as monitored with electromyographic (EMG) recordings (below Figure). (2) the mice did not cross the defined threshold during free movement and foraging. (3) the M1 population calcium fluorescence signal in the operant (motor) behavior displayed the different pattern with volitional control of neural activity. Our view of M1 calcium fluorescence signal representation of volitional control is consistent with as the previous findings that arbitrarily selected primary motor cortex (M1) neurons for volitional control have little relationship with native limb movement [9] and that a stable M1 cortical representation for prosthetic function can be stored [8] . Taken together, we believe that M1 calcium fluorescence signals are representation for volitional signal with limited relationship with movement.

The development of Calcium PTE and Calcium PTH based on the operant volitionally controlled task coupled with the concept of representing behavioral motivation by the break-point in response to escalating efforts. Thus, the calcium signal change in M1 for Calcium PTE and Calcium PTH also was not related to the overt movement, but to the volitional signal. In the current study, we progressively increased M1 neural activity with a series of pre-set criterion to progressively escalate volitional efforts and estimated the motivation by maximal efforts via breakpoints in the tasks. Indeed, there were progressively increase in the TCE and PTH as defined by the formula, representing the escalating efforts (and thus the volitional motivation). Therefore, the maximal neural activities (calcium fluorescence signal, either in TCE or PTH) is related to volitional motivation (not movement).

Figure, The population calcium signal was dissociated with the EMG signal (Left) as the distribution of correlation coefficients between EMG activity and M1 population calcium signal changes for all trials in all session across the neuroprosthetic learning was not significantly different from zero (right) [7].

2. Whether A2AR is specifically expressed in the striatopallidal neurons, which determines whether the chemogenetic and pharmacological manipulations are specific.

Response: We thank the editor’s comments. A_{2A} receptors are predominantly expressed in the striatopallidal neurons and are highly relevant to the function of the indirect pathway of the striatum [10]. Consistent with the previous study with adora2A-rM3Ds mice [11], we confirmed that rM3Ds was specifically expressed in the striatopallidal pathway (i.e. striatal neurons (Fig. A, Str: Striatum) and striatopallidal projections) (Fig. A, LGP: lateral globus pallidus) and were colocalized with A_{2A}Rs in the striatopallidal neurons (Figure B, rM3Ds: red, A_{2A}R: green), but not with dopamine D1 receptors in the striatonigral neurons (Fig. C, rM3Ds: red, D1R: green). These results confirm that chemogenetic (dM3Ds) and pharmacological (KW6002) manipulations are specific to the striatopallidal pathway.

3. Figure 2B does not clearly reflect that the holding time of the calcium signal above the threshold gradually increases with the progressively increase efforts. The holding time above the calcium fluorescence threshold across trials should be counted in the PTH test.

Response: We thank the editor’s comments on this point. By adapting the break-point for representing “volitional motivation” in response to the escalating effort, we defined every trial's holding time using the formula in Calcium PTH (holding time = $0.1 * 1.05^{(t-1)}$, t = trial number). In this schedule, the holding time for per trial was achieved by animal only once, and as such the holding time above the calcium fluorescence threshold across trials can not be accounted. Moreover, the increase of holding time is mainly reflected in the width of calcium fluorescence signal. We also tried to analyze the width of calcium fluorescence signal for per trial in Calcium PTH. However, once the set holding time by the formula is reached in the experimental process, the reward will be given and the trial will be terminated. In addition, each trial is relatively an independent experiment(holding time is different for per trial) and no stable calcium fluorescence signal will be generated. Thus, the width of calcium signal of each trial is not invariably consistent with the set holding time. However, the neural activity did not represent volitional motivation and the maximal neural activities (calcium fluorescence signal, either in TCE or holding time) is related to volitional motivation in the current study.

Senior Editor:

In the event that you choose to resubmit a new version of the manuscript, I would ask that you first pay particular attention to the comments that have been provided in relation to the statistical analyses. In particular, it should be demonstrated that all assumptions pertaining to the use of parametric analyses have been satisfied. If this is not the case, non-parametric procedures would be used instead. Given that multiple tests were performed, appropriately stringent tests should be applied to account for potential inflation of the effective alpha level.

Response: We thank the Senior Editor for the comment. We have paid special attention to the statistical analyses. The data in Figure 3B, 3E, 4E were tested and shown to be normally distributed and accordingly we have used the unpaired for data analyses. The data in Figure 4B were tested and found to be not normally distributed, and accordingly we have used the non-parametric Mann-Whitney U test for data analysis. We analyzed the data in Figure 5C, 5D, 5F, 6C, 6D, 6F by RM one-way ANOVA($P < 0.05$), followed by post-hoc comparison with LSD test.

REFEREE COMMENTS

Referee #1:

Title: Neural Representation and Modulation of Volitional Motivation in Response to Escalating efforts

Summary: In this article, the authors present data from mice expressing calcium indicator dyes while recording changes in fluorescence within the primary motor cortex (M1) or the nucleus accumbens. The authors aimed to determine if the amount of effort the mice had to expend to obtain a given reward amount would change as the effort per unit reward increased in a predictable manner, and how modulation of the striatum changes motivational aspects of the subjects as seen via breakpoints in the tasks.

Specific questions and issues:

1. The summary (the first two figures without legends) figure seems to show the rat using its face to press the lever in yellow. Is this in fact what was done? If not, please make the figure more representative of the real situation. In addition, the Volitional motivation figure is also a bit unclear as to what is being represented as compared to the actual experiment.

Response: We thank the reviewer's comments. The mice used the right forelimb to press the lever to receive the reward and we have revised the first Figure accordingly. The first figure was graphical abstract, illustrating how to assess volitional motivation in the current study (the below figure).

2. There are no figure legends for the first two figures shown in the combined .pdf, I'm not sure if these were to be in the supplemental information or somewhere else in the paper?

Response: We thank the reviewer's comments. The first figure in the combined PDF file was graphical abstract and thus no figure legend was included. The second figure was the Figure 5 in the manuscript and have deleted the second figure in the combined PDF file and the figure legend to Figure 5 was included in the manuscript.

3. The below sentence is rather difficult to follow. There are many grammatical errors in the text making some ideas rather hard to follow. Perhaps using grammar checking software could help with this, or a native American English speaker.

Response: We thank the reviewer's comments and have revised the grammatical errors in the text.

a. "The first quantitative assessment of volitional motivation by progressively representation of the M1 neural activity"

Response: We thank the reviewer's comments and have revised this sentences as "Volitional motivation was quantitatively evaluated by the M1 neural activity in response to progressively escalating volitional efforts."

4. I'm not fully sure I follow the below sentence, please revise.

a. "The volitional control of neural activity directly reinforces the target neurons using real-time biofeedback and is driven by motivational factor (volitional motivation)."

Response: We thank the reviewer's comments and have revised the sentence as following: "The volitional control of neural activity is driven by a motivational factor (volitional motivation) which directly reinforces the target neurons via real-time biofeedback ."

5. Please summarize the previous studies relevant information on this point here. "Mice underwent volitional neural learning for 10 days as described previously [16]".

Response: We thank the reviewer's comments and have summarized the relevant information from the previous study as following: "Mice were transfected with AAV9-syn-GCaMP6f-WPRE-SV40 to express the genetically encoded Ca^{2+} indicator GCaMP6f in the M1 cortex and the calcium fluorescence signal was monitored by fiber photometry system [18]. Mice were trained to perform the volitionally controlled neural task to reach the correct percentage of 85-100% for obtaining the reward (Fig.1A)."

6. Again, you need to at least give the reader the information needed to judge and understand your current work, so, please summarize the pertinent information here as well. If the information in the following sentence is that description, please make this clear such as saying we briefly summarize this information below etc. "After smoothing the data with a moving average filter (20 ms span), the calcium fluorescence signal and dopamine fluorescence signal analysis for the event-related behavior is described in previous research [16]."

Response: We thank the reviewer's comments and have summarize the previous studies relevant information as "As in the our previous study, we performed data analysis in MatLab platerform (Math Works, Natick, USA) with custom-written programs [18]. After smoothing the data with a moving average filter (20 ms span), we analyzed the event-related calcium fluorescence signal and dopamine fluorescence signal in relationship with the reward (with the reward as time "0" point)"

7. Is this baseline the same as the aforementioned "low baseline procedure"? "We derived the values of fluorescence change ($\Delta F/F$) by calculating $(F-F_0)/F_0$, where F_0 is the baseline fluorescence signal averaged over a 1-2s-long control time window, which was typically set 1-2 s preceding the trigger events." If they are not the same perhaps make this clearer.

Response: We thank the reviewer's comments and apologize for any confusion between these two different expressions of the baseline. (a) In the "low baseline procedure" (on-line analysis), the baseline was defined as the lowest F_0 value within 1 min time window and recalculated for every minute using the lowest F_0 value during the volitionally controlled neural task. Therefore, the baseline adjustment in the low baseline procedure is an online real-time adjustment throughout the training process. (b) In the baseline calcium signal for the event-related behavior (off line-analysis), the calcium fluorescence signal for the event-related behavior is offline analysis, where the baseline was typically set 1-2 s preceding the trigger

events (reward delivery). We have revised “low baseline procedure” and the baseline calcium signal for the event-related behavior in the manuscript.

8. In the above what are the trigger events, as this term is not used elsewhere in the paper?

Response: We thank the reviewer’s comments and have revised “the trigger events” as “reward delivery”.

9. Statistics: It seems from many of the figures that the variance of the two populations is not similar, which is a violation of the assumptions made for using the unpaired t-test. In addition, it is not indicated that a test for normality was conducted. A non-parametric test, such as the Mann-Whitney U-test may be more suited for this data.

Response: We thank the reviewer’s comments on the statistical analyses and have carefully analyzed the data distribution. The data in Figure 3B, 3E and 4E were tested and shown to be normally distributed. Accordingly, we have used parametric analysis (i.e. the unpaired t-test) for these data. The data in Figure 4B were tested and found to be not normally distributed, and accordingly we have used the non-parametric Mann-Whitney U test for data analysis.

10. For the following statement there are several publications showing reward expectation, value, and motivational neural correlates that could be cited in this work. "Consistent with the prediction error signal, we detected the development of prediction signal (i.e., calcium fluorescence signal associated with cue presentation, before the reward) in the repeated FR1→FR5 trials.

Response: We thank the reviewer’s comments and have cited the references b and d (see below) in the revised manuscript: “Consistent with the prediction error signal (a. b. 44-45), we detected the development of prediction signal (i.e., calcium fluorescence signal associated with the cue presentation, before the reward) in the repeated FR1→FR5 trials.” .

a. 1. Marsh, B. T., Tarigoppula, V. S., Chen, C. & Francis, J. T. Toward an autonomous brain machine interface: integrating sensorimotor reward modulation and reinforcement learning. *J Neurosci* 35, 7374-87 (2015).

b. 2. An, J., Yadav, T., Hessburg, J. P. & Francis, J. T. Reward Expectation Modulates Local Field Potentials, Spiking Activity and Spike-Field Coherence in the Primary Motor Cortex. *eNeuro* June 6 2019, (2019).

c. 3. Yao, Z., Hessburg, J. P. & Francis, J. T. Normalization by valence and motivational intensity in the sensorimotor cortices (PMd, M1, and S1). *Sci Rep* 11, 24221 (2021).

d. 4. Ramkumar, P., Dekleva, B., Cooler, S., Miller, L. & Kording, K. Premotor and Motor Cortices Encode Reward. *PLoS One* 11, e0160851 (2016).

e. 5. Ramakrishnan, A. et al. Cortical neurons multiplex reward-related signals along with sensory and motor information. *Proc Natl Acad Sci U S A* 114, E4841-E4850 (2017).

f. Zhao, Y., Hessburg, J. P., Kumar, J. N. A. & Francis, J. T. Paradigm Shift in Sensorimotor Control Research and Brain Machine Interface Control: The Influence of Context on Sensorimotor Representations. *Front Neuroscience* 12, (2018).

11. For this text many of the above refs would be applicable. "indicating that neural activity may represent integrated signals." In addition, I've added a ref where BMI was performed while considering this integrated activity of reward expectation/motivation and movement related neural decoding. Note: Please do not feel that you must use any of the suggested citations, but if not these references then please do include any other pertinent refs that might take their place.

Response: We thank the reviewer’s comments and agree with you. We have cited these refs in the revised manuscript :“The direct control of neural activity in BMIs may be a consequence of the integration of the cortical system, subcortical motivational areas, and neurotransmitter system information, indicating that neural activity may represent integrated signals[a-f, 44, 45, 47-50]”.

12. Baseline window concerns: "fluorescence change ($\Delta F/F$) by calculating $(F-F_0)/F_0$, where F_0 is the baseline fluorescence signal averaged over a 1-2s-long control time window, which was typically set 1-2 s preceding the trigger events." Again, not clear what the trigger event is in this sentence, please make this clear. Secondly, as there may be changes in the baseline with learning, or changes due to the increased effort with time, the ratio between baseline and post-trigger could be changing due to either movement of the baseline's height, the post-trigger height, or both. It would be helpful to see non-baseline corrected, more raw representations of the data.

Response: We thank the reviewer's comments and specified "the trigger events" as "reward delivery". As suggested, we have included a proportion of the raw calcium signal for Calcium PTH. We agreed that the baseline may change with learning, effort and motivation. However, the neurons associated with learning, effort, motivation may activate the specific neurons in the different time. These activated neurons did not have the noticeable effect on the population of neural activity when analyzed random activity. Thus, the event-specific analysis of the calcium signal (e.g. in relationship with the reward) is required to shown specific calcium signal patterns

13. Changes in variance:

14. This sentence is hard to follow please edit it as I'm not sure what you are saying. "The volitional control of neurons directly reinforce the neural activity and efforts of volitional control can be escalated by the changed criteria to continuously increase neural activity or by continuously increase holding time for neural activity."

Response: We thank the reviewer's comments and have revised the sentence as following: "The volitional control of neurons directly reinforces the neural activity by biofeedback. The escalated effort for volitional control can be specifically increased by predefined criteria (schedule) to progressively increase the required holding time for neural activity above the defined threshold".

15. Please explain how the below is not contradictory as it is stated that the threshold for the volitional neural task comes from the behavioral task, but that the neural activity between the two tasks is opposite. Perhaps I'm missing something that you can help me see. "mice were conditioned to increase calcium fluorescence signal in M1 neurons above the defined threshold value"... "The defined threshold was based on averaging M1 neural activities over 6 days of instrumental conditioning (pressure lever)."..."Lastly, the M1 population calcium fluorescence signal in the behavior learning is opposite to volitional control of neural activity learning."

Response: We thank the reviewer's comments and apologized for the misrepresentation here. Actually, we want to express that the M1 population calcium fluorescence signal in the behavior operant motor learning displayed the different patterns with volitional control of neural activity learning (, but not "opposite" patterns as we initially described). We also revised the last sentence as following: "Lastly, the M1 population calcium fluorescence signal in the behavior learning is opposite to volitional control of neural activity learning." As "Lastly, the M1 population calcium fluorescence signal in one-lever operant behavior (press the lever for one time to get a reward in a trial) displayed different patterns compare to volitional control of neural activity.." These different patterns may be interpreted that volitional control of neural activity and operant behavior in M1 may involve different neuronal populations.

16. The below statement seems more pessimistic than it may need to be as you can determine what muscles are activated by the brain region you are recording from and then you would only need to obtain EMG from those muscles I would think. Also, some BMI research uses animals or humans that can't move. "Nevertheless, this is a question common to all BMI studies that is ultimately unanswerable without recordings from every muscle in the body."

Response: We thank the reviewer's comments on this point. We agree with the reviewer that the recording of specific muscle activity from the corresponding brain regions and of the animals that can't move (after local anesthesia) would partially disassociate the motor activity from volitional control. We have deleted the sentence in the revised manuscript. .

17. Author Contributions: Please get rid of all the "or" statements and simply put down what everyone did.

Response: We thank the reviewer's comments and revised the text by deleting "or" statement and write down the specific statement.

18. Do you really mean to say that you, the authors, volitionally controlled the M1 population? "In this study, we volitionally controlled the M1 population neural activity in the operational conditioning by real-time monitoring of calcium fluorescence signal using fiber photometry system" If not please correct this sentence to make it clear who is controlling the M1 population.

Response: We thank the reviewer's comments and the sentence as following: "In this study, mice volitionally controlled the M1 population neural activity in the operational conditioning by real-time monitoring of calcium fluorescence signal using fiber photometry system"

19. Some indication as to how the neurons are activated would be helpful to the reader here, as in what is the mechanism of activation. "Activation of the striatopallidal pathway in A2A-rM3Ds mice was performed by CNO."

Response: We thank the reviewer's comments and have provided the brief description of the chemogenetic activation of the neurons by A_{2A}R-rM3Ds as following: "The rM3Ds was selectively and stably expressed in striatopallidal neurons in A_{2A}-rM3Ds mice and activation of the striatopallidal pathway in A_{2A}-rM3Ds mice was achieved by systemic injection of CNO which specifically activate rM3Ds in the striatopallidal neurons."

20. There is no indication as to where the GRABDA sensors were obtained from. In some sections it is written as above and in others it is GRABDA , please be consistent and use one or the other of these.

Response: We thank the reviewer's comments and have provided the detailed description for the Method section to clearly state that rAAV-hsyn-DA4.4-WPRE-hGH was obtained from BrainVTA (catalog# PT-1340; Wuhan, China). We revised and used "GRAB_{DA}" consistently throughout the manuscript.

21. I'm not sure I fully follow the logic behind the below two statements. Please explicitly state what you have in mind as to how these statements make sense as I seem to be missing something.

a. "Moreover, this result also indicated that the effect of KW6002 on M1 neural activity did not affect the volitional control of neural activity, because M1 signals were lower in KW6002 group compared to the control group(Fig. 4C, 4F).However, the reduction of calcium fluorescence signal after the reward was more pronounced in the KW6002 group compared to the vehicle group (Fig.4C,4F, n=6), which maybe indicated KW6002 enhanced the sensitivity for the reward."

Response: We thank the reviewer's comments on this point. First, the A_{2A}Rs are expressed at a high level in the striatopallidal neurons and low to moderate levels in cortical neurons. Both the cortical and striatal A_{2A}Rs can exert control of cognition such as working memory as we recently demonstrated[12]. Thus, the direct action of A_{2A}Rs in M1 neurons, or the indirect action at the striatal neurons (with circuit feedback onto

the M1 neurons) can regulate volitional control. As shown in Figure 4C and 4F, when we analyzed the calcium fluorescence signal for successful volitional control trials, KW6002 did not influence the volitional control of M1 neural activity. Thus, we reasoned that KW6002 acted indirectly at the striatal A_{2A}Rs with feedback onto the M1 neurons to regulate volitional control. This notion is consistent with our preliminary analysis indicating that focal genetic knockdown of A_{2A}Rs in DMS also enhanced volitional control of neuroprosthetic learning (unpublished data). However, we have deleted the sentence "However, the reduction of calcium fluorescence signal after the reward was more pronounced in the KW6002 group compared to the vehicle group (Fig.4C,4F, n=6), which maybe indicated KW6002 enhanced the sensitivity for the reward." in the revised manuscript.

22. In the below statements you state (data not shown) and then mention a ref [23], but you never state what the actual outcome was. Please state explicitly what ref 23 and the (data not shown) indicate.

a. "To verify dopamine dynamics for reward value, we delayed the reward delivery for 10s, then phasic dopamine dynamics also delayed for about 10s (data not shown). These findings are consistent with the phasic dopamine dynamics in the NAc during motor skill tasks [23]."

Response: We thank the reviewer's comments on this point. We have provided the data in the revised manuscript as "supplemental Figure 1" and revised as "To verify dopamine dynamics for reward value, we have programmed the time for the reward delivery with delay by 10s. Interestingly, the delayed reward delivery by 10 second was associated with the delayed phasic dopamine dynamics by about 10s (supplemental figure 1)". These findings strongly support that the dopamine neurons fire specifically in response to the reward value."

23. indicated that the dopamine dynamics for reward predictions in the NAc was inversely correlated with motivation, but not with the reward value in motivation test.

Response: We thank the reviewer's comments and revised as "These results also indicated that escalating efforts were also negatively correlated with dopamine dynamics for reward prediction in NAc but not with the reward value in motor skills."

24. Don't animals in general choose the path of least effort when obtaining food. I'm assuming some information is missing in the following statement. ..."but they select the path to food reinforcement that requires less effort [30, 31]."

Response: We thank the reviewer's comments on this point. Indeed, animals in general choose the path of least effort when the reward was same for the both paths. However, the animal faced the choice here between making more effort to obtain more food or making less effort to obtain less food. We have clarified the statement as following: "Animals with impaired dopamine transmission can reallocate their instrumental behavior away from food-reinforced tasks with high response requirements, and instead select less effortful food-seeking behaviors [30, 31]. The instrumental output and effort-related choice impaired by dopamine D2 antagonism were reversed by A_{2A}R blockade or genetic deletion [32-36]."

25. As the previous sentence uses ref 9 perhaps you could use a more specific ref for this definition of

motivation "Motivation is represented by the rewards of maximal efforts against the costs of an action for its potential benefits [9].

Response: We thank the reviewer's comments and have used a more specific ref [9-11] in the revised manuscript.

(see below)

Figures:

Fig.1 It seems panel C shows -2 seconds to +5 seconds, not +-5 as stated in the legend. "C) The calcium fluorescence signal change before/after the reward delivery (± 5 s) for escalating efforts (trial by trial) in Calcium PTE testing (n = 6)." You may also want to include M1 in the title to make this more clear.

Response: We thank the reviewer's comments and revised as "C) The calcium fluorescence signal change in M1 neurons before/after the reward delivery (+5s) for escalating efforts (trial by trial) in Calcium PTE testing (n = 6)."

Fig.2 Panel A has PTE rather than PTH in the flow chart. This figure shows the +-5s, perhaps you should change Fig.1 to match Fig.2s format.

Response: We have revised the figure as suggested in the manuscript.

Fig.4. C) should read (-5) and after (+5) rather than what is currently written, which is both are +5.

Response: We have revised figure as suggested in the manuscript.

Fig. 5. Is F significant and if so perhaps use the same convention of *, **, *** etc.

Response: We thank the reviewer's comments. We have analyzed the 5 days FR5(Fig,5C, 5D) for volitional control using RM one-way ANOVA and used LSD for post-hoc analysis. These results indicate that there were significant changes in prediction component (RM one-way ANOVA, $P=0.043$, $F(2.333, 11.67) = 4.007$) and in reward component (RM one-way ANOVA, $P=0.035$, $F(2.263, 11.32) = 4.422$). There is significant change between FR5-1 and FR5-5 when LSD test was used(Fig. 5C, $P=0.029$; Fig. 5D, $P=0.009$). We have revised Figure 5C, 5D by adding *, **, *** when there was statistical significance accordingly.

Fig. 6. Same as Fig.5.

Response: We thank the reviewer's comments. We have analyzed the 5 days FR5(Fig,5C, 5D) for motor skills using RM one-way ANOVA and used LSD for post-hoc analysis. These results indicate that there were significant changes in prediction component (RM one-way ANOVA, $P=0.044$, $F(2.115, 14.80) = 3.839$) and in reward component (RM one-way ANOVA, $P=0.027$, $F(2.797, 19.58) = 3.874$). There is significant change between FR5-1 and FR5-5 when LSD test was used(Fig. 6C, $P=0.049$; Fig. 6D, $P=0.01$). We have revised Figure 6C, 6D by adding *, **, *** when there was statistical significance accordingly.

Referee #2:

General comments: The manuscript "Neural Representation and Modulation of Volitional Motivation in Response to Escalating efforts" by Zhang et al. reported a criterion of calcium fluorescence signal in the progressive ratio task and progressive hold-down task, which may consider as a quantitative assessment of progressively escalating efforts and relate to striatopallidal pathway. It is an interesting design. The authors clearly described the procedure and detected calcium signals in the M1 and dopamine dynamics in the NAc during the behaviors. But the details of the experiments are less compelling. Below are some comments that the authors may need to consider to improve from the current version:

Major concerns:

1. In general, the authors tried to use the calcium PTE and PTH in the M1 to detect volitional motivation. However, it is still hard to identify if the calcium PTE and PTH in the M1 are specifically coding for behavioral action or for volitional motivation. It would be better to know what is the frequency of calcium based threshold-crossing events (TCE) with 30s interval :1) during the pavlovian training stage (auditory cue pairs with the reward); 2) during the instrumental training stage (press the lever); 3) during the palvovian-instrumental transfer (PIT). These patterns of calcium-TCE would give us a clue that the coding pattern of the calcium signals in the goal directed actions. To my understanding, the volitional motivation should be more related to goal directed action rather than habitual action.

Response: We thank the reviewer's comments. As we outlined in the detailed response to the Senior Editor (see above), we have partially verified in the previously study that the calcium fluorescence signal in M1 for volitionally controlled neural task was not related to overt movement, including the disassociation of M1 calcium signal and EMG activity of the forelimb during volitional conditioning; lack of the crossing-threshold M1 calcium activity during free movement and foraging; and different the M1 population calcium fluorescence signal during operant (motor) and volitional conditioning. Therefore, the calcium fluorescence signal in M1 mainly represents volitional signal for operant volitionally controlled neural task. Furthermore, the development of volitionally controlled neural task involves the instrumental and volitional conditioning procedures. The sound cue was presented as the beginning of the trial and was present throughout the trial. After the cue presentation, animals can only obtain the reward after pressing lever. Therefore, the procedure did not involve Pavlovian conditioning and Pavlovian-to-instrumental transfer procedure. Importantly, we progressively increased M1 neural activity with a series of pre-set criterion to escalate volitional efforts and estimated the motivation by maximal efforts via the breakpoint in the tasks.

2. Figures 1 and 2: Authors should explain why they choose 1, 2, 4, 6, 9, 12, 15...of TCEs as the sequential trial for calcium PTE test. Is it the only effective or optimal procedure for detecting the increased efforts? Similar in the calcium PTH analysis, why did the authors choose a start from 105ms? They should provide the general or average holding time in a single action, or any criterion for these procedures, since the procedure itself could affect calcium signals during different trials.

Response: We thank the reviewer's comments. We adapted the formula ($TCE = 5 * e^{(0.2 * t) - 5}$, $t =$ trial number) for TCE for per trial in analog to the representation of behavioral motivation by the break-point to escalating efforts in the PRT test. Furthermore, we found in the previous study that the average of holding time for the volitional control by the preset threshold was ~ 100 ms. Thus, we choosed the increasing holding time with the starting holding time at 105ms for volitional control.

3. Figures 3 and 4: Could authors explain why CNO manipulation inhibited motor function but not affected the calcium signal in the M1? Does it mean the volitional motivation is different to behavioral motivation or behavioral action?

Response: We thank the reviewer's comments on this point. Consistent with the previous study, CNO-mediated activation of the striatopallidal pathway inhibited motor activity, confirming the inhibitory effect of this pathway on motor activity. However, CNO injection did not affect M1 activity and yet did suppress volitional motivation as evident by the reduced break-point in the PET and PHD test. This suggest that the operant and volitional condition may involve different neural mechanisms (such as involving different neural populations of the parallel cortex-basal ganglia-cortex loop).

4. Figures 5 and 6: Due to the correlation analysis of dopamine dynamics for the reward prediction (Figure 5F and 6F) was quite low (r square = 0.06). It is better to provide the mean "Height" in the trials of PTE and PRT tests as well. Also it would be easy to compare the height of the first trial vs. the height of the last trial from each mouse in the PTE and PRT tests to confirm the conclusion.

Response: We thank the reviewer's comments on this. As suggested, we have analyzed the mean "height" in the trials of the PTE (Figure A) and PRT (Figure B). These results indicate that there were significant changes in the mean "height" in the trials of both the PET (Fig. 5F, RM one-way ANOVA, $P=0.025$, $F(2.960, 14.80) = 4.182$) and PRT (RM one-way ANOVA, $P=0.046$, $F(2.048, 10.24) = 4.184$). We have used the LSD as well as Bonferroni post-hoc comparison (with correction for multiple tests) for post-hoc analysis. The analysis indicated that there was significant decrease between the height of the first trial and the height of the last trial when LSD testing ($p<0.01$) was employed, but the effect was not present when Bonferroni test was employed. Thus, there was apparent decrease in the mean "height" in the trials of the PTE (Figure A) and PRT.

Minor concerns:

1. The real data of location and expression of GCamp6f in M1 and GRABDA sensors in NAc should be shown. The injection site and expression area of the drugs the NAc could affect the behavioral actions sensitively.

Response: We thank the reviewer's comments on this point and have now included the real data showing the location and expression of Gcamp6f in M1 and GRAB_{DA} sensors in NAc in A, B, (Figure 5A, 6A in the revised manuscript).

2. Authors said "To verify dopamine dynamics for reward value, we delayed the reward delivery for 10s, then phasic dopamine dynamics also delayed for about 10s (data not shown)". Please show the data which is important in the study.

Response: We thank the reviewer's comments on this point. As suggested, we have now included this data set (see the figure below) as the supplemental figure 1. As you can see, "the delayed reward delivery by 10 second was associated with the delayed phasic dopamine dynamics by about 10s. These findings strongly support that the dopamine neurons fire specifically in response to the reward delivery." We have included this in the revised Result and Discussion.

We thank the Editors and Reviewers again for their constructive comments and suggestions and the opportunity to revise and improve the manuscript. We hope these new analyses and revision have fully addressed the reviewer and editor's concerns and the manuscript is now considered to be acceptable for publication in "*Journal of Physiology*".

Sincerely yours,
Liping Zhang, PhD
Jiang-Fan Chen, MD PhD

References:

1. Marsh BT, Tarigoppula VS, Chen C, Francis JT: **Toward an autonomous brain machine interface: integrating sensorimotor reward modulation and reinforcement learning.** *J Neurosci* 2015, **35**:7374-7387.
2. Ramkumar P, Dekleva B, Cooler S, Miller L, Kording K: **Premotor and Motor Cortices Encode Reward.** *PLoS One* 2016, **11**:e0160851.
3. Ramakrishnan A, Byun YW, Rand K, Pedersen CE, Lebedev MA, Nicolelis MAL: **Cortical neurons multiplex reward-related signals along with sensory and motor information.** *Proc Natl Acad Sci U S A* 2017, **114**:E4841-E4850.
4. Zhao Y, Hessburg JP, Asok Kumar JN, Francis JT: **Paradigm Shift in Sensorimotor Control Research and Brain Machine Interface Control: The Influence of Context on Sensorimotor Representations.** *Front Neurosci* 2018, **12**:579.
5. An J, Yadav T, Hessburg JP, Francis JT: **Reward Expectation Modulates Local Field Potentials, Spiking Activity and Spike-Field Coherence in the Primary Motor Cortex.** *eNeuro* 2019, **6**.
6. Yao Z, Hessburg JP, Francis JT: **Normalization by valence and motivational intensity in the sensorimotor cortices (PMd, M1, and S1).** *Sci Rep* 2021, **11**:24221.
7. Zhang L, Zhou Y, Liu C, Zheng W, Yao Z, Wang Q, Jin Y, Zhang S, Chen W, Chen JF: **Adenosine A2A receptor blockade improves neuroprosthetic learning by volitional control of population calcium signal in M1 cortical neurons.** *Neuropharmacology* 2020:108250.
8. Karunesh Ganguly JMC: **Emergence of a Stable Cortical Map for Neuroprosthetic Control.** *PLoS Biol* 2009.
9. Law AJ, Rivlis G, Schieber MH: **Rapid acquisition of novel interface control by small ensembles of arbitrarily selected primary motor cortex neurons.** *J Neurophysiol* 2014, **112**:1528-1548.
10. Schiffmann SN, Fisone G, Moresco R, Cunha RA, Ferre S: **Adenosine A2A receptors and basal ganglia physiology.** *Prog Neurobiol* 2007, **83**:277-292.
11. Farrell MS, Pei Y, Wan Y, Yadav PN, Daigle TL, Urban DJ, Lee HM, Sciaky N, Simmons A, Nonneman RJ, et al: **A Galphas DREADD mouse for selective modulation of cAMP production in striatopallidal neurons.** *Neuropsychopharmacology* 2013, **38**:854-862.
12. Zhihui Li XC, Tao Wang, Ying Gao, Fei Li, Long Chen, Jin Xue, Yan He, Yan Li, Wei Guo, Wu Zheng, Liping Zhang, Fenfen Ye, Xiangpeng Ren, Yue Feng, Piu Chan, Jiang-Fan Chen **The Corticostriatal Adenosine A2A Receptor Controls Maintenance and Retrieval of Spatial Working Memory.** *Biol Psychiatry* 2018, **83**:530-541.

Nov. 21, 2022

Re: JP-RP-2022-283915XR1 “**Neural Representation and Modulation of Volitional Motivation in Response to Escalating efforts**”

Dear Editor:

We thank the Editor and Reviewers very much and have further revised the manuscript in response to the Reviewer's the minor issues. The point-by-point response to reviewer's comments are provided as following:

Reviewing Editor:

The authors have made a substantial and extensive revision. Please further address the minor issues raised by two reviewers.

Response: We thank the editor's comments and have further addressed the minor issues of the reviewers.

REFEREE COMMENTS

Referee #1 (please see attachments):

Thank you for all of the modifications you have made, as they have made a difference in the readability of the paper. I've included a couple more in the attached files that should be taken into consideration.

Response: We thank the reviewer's comments and have made suggested revisions as follows:

1. This distribution does not seem normal and the median may be a better statistic to ask questions about differences from zero.

Response: We thank the reviewer's comments on this point. The EMG distribution in the rebuttal letter has been published in the previous study (Zhang et al. 2020). We will carefully evaluate the EMG distribution data and adopt this suggestion in future experiments.

2. Was this 20 ms bin moved forward by 1 ms or some other number? This level of information should be given.

Response: We thank the reviewer's comments on this point. In the experiment, a moving average filter (20 ms span) was moved at 10 ms per step. We have revised the text as following: “After smoothing the data with a moving average filter (20 ms span with 10 ms moving step), we analyzed the event-related calcium fluorescence signal and dopamine fluorescence signal in relationship with the reward (with the reward as time “0” point)”.

3. It seems from many of the figures that the variance of the two populations is not similar, which is a violation of the assumptions made for using the unpaired t-test.

Response: We thank the reviewer's comments and re-analyzed these data. Although the variance of the two populations is not similar in the figure, only one data set does not conform to the normal distribution, which we have used non-parameter Mann-Whitney U test (Figure 4B). The other data sets (Figure 3E, 3H and 4E) were conformed to the normal distribution, so these data sets were analyzed by the unpaired t-test.

4. A clear statement of your definition of operant behavior and volitional should be made clear as one could say the BMI control is also operant.

Response: We thank the reviewer's comments on operant behavior. To avoid this confusion, we have used “instrumental” behavior to distinguish volitional (BMI) control. We have revised the manuscript as following: “Lastly, the M1 population calcium fluorescence signal in one-lever instrumental behavior (i.e. by pressing the lever for one time to get a reward in a trial) displayed different patterns compare to

volitional control of neural activity.”

5. You never defined what LSD is.

Response: We thank the reviewer’s comments and defined the LSD as following “The mean “height” of the dopamine fluorescence signals analyzed by one-way ANOVA and followed by post-hoc Fisher's Least Significant Difference (LSD) test.”

Finally, we also have revised the manuscript to correct grammatical errors according to the reviewer’s comments in the text.

Referee #2:

The authors have carefully addressed issues raised by reviewers. However, there remain some minor concerns.

1. Figures 5 and 6: How did the authors calculate the "Height" of fiber photometry signals? Does "height" mean the highest peak value minus baseline? If so, what is used as the baseline? The authors should describe how they analyze fiber photometry data and calculate the "height" in more detail in Method. In figure 6E, it seems the highest peak in trial 9 is higher than that in trial 7. However, the "height" of trial 9 is lower than that of trial 7 in figure 6F. Why? Lastly, is it better to use AUC (area under curve) instead of the highest peak as AUC better describes the change of calcium signals in the selected time window when there is more than one peak? This is rather important because it directly leads to the conclusion that there is negative correlation between the escalating efforts and NAc dopamine signal.

Response: We thank the reviewer’s comments on this point. The baseline was defined as the average dopamine fluorescence signal within -5~-6 seconds (immediately prior to the peak time window) before the reward delivery. The “height” represents the highest peak of dopamine dynamics of 0~-5s of reward delivery. We reanalyzed the dopamine fluorescence signal of trial 7 and 9 in 6 mice respectively. The "height" of trial 7 was indeed higher than that of trial 9 (the below figure). Therefore, the inconsistency between Figure 6E and Figure 6F was due to slightly different alignment of individual calcium signal tracer in Figure 6E. Therefore, we have carefully rechecked all the alignment to ensure its accuracy. We have made corresponding minor changes in the manuscript.

In our experience, the AUC analysis is more suitable for the dopamine fluorescence signals recorded during the continuous slow release of dopamine. Nonetheless, most of the dopamine fluorescence signals in our current study have a major peak (the above picture). Furthermore, we have also performed the AUC analysis

for each individual the dopamine fluorescence signals of 9 trials (the below figure), which revealed an overall similar reduction trend, but without no clear pattern ($p=0.0868$ by One-way ANOVA).

2. Page 10: "this result also indicated that the effect of KW6002 on M1 neural activity did not affect the volitional control of neural activity ..." This description is confusing. The activation/inhibition of NAc result in lower/higher breakpoint with little change in M1 activity pattern, which infers that M1 is not in charge of volition control but is controlled directly/indirectly by volition and functions as the final output. KW6002 affected the volitional control because it increased the breakpoints in PTE and PTH tests.

Response: We thank the reviewer's comments. We have revised this sentence as "this result also indicated that KW6002 acted indirectly at the striatal $A_{2A}R$ s with feedback onto the M1 neurons to regulate volitional control."

3. Page 7: "Primary antibodies used: A2AR (frontier, 1:500), mCherry (Clontech, 1:500), D1 (Clontech, 1:500), goat anti-rabbit AlexaFluor-594(1:250), goat anti-rat AlexaFluor-555(1:250)." Goat anti-rabbit AlexaFluor-594 and goat anti-rat AlexaFluor-555 are secondary antibodies.

Response: We apologized for this error and revised the Method as following: "For immunohistochemistry analysis, we have used the following primary antibodies A2AR (frontier, 1:500), mCherry (Clontech, 1:500), D1 (Clontech, 1:500), together with secondary antibodies goat anti-rabbit AlexaFluor-594(1:250), goat anti-rat AlexaFluor-555(1:250)".

4. Page 3: "Finally, volitional motivation evaluated by neural plasticity in response to progressively escalating efforts with the breakpoints (maximal plasticity of neurons) representing the size of the volitional motivation." Grammatical error.

Response: We thank the reviewer's comments and have revised as "Finally, volitional motivation was evaluated by the response of neuroplasticity to escalating effort, with the breakpoint (maximum plasticity of neurons) representing the size of the volitional motivation".

We hope these new analyses and revision have fully addressed all these additional issues raised by the reviewer and the manuscript is now considered to be acceptable for publication in "*Journal of Physiology*".

Sincerely yours,
Liping Zhang, PhD
Jiang-Fan Chen, MD PhD

Dear Dr Zhang,

Re: JP-RP-2022-283915XR2 "Neural Representation and Modulation of Volitional Motivation in Response to Escalating efforts" by Liping Zhang, Chengwei Liu, Xiaopeng Zhou, Hui Zhou, Shengtao Luo, Qin Wang, Zhimo Yao, and Jiang-Fan Chen

We are pleased to tell you that your paper has been accepted for publication in The Journal of Physiology.

Authors should note that it is too late at this point to offer corrections prior to proofing. The accepted version will be published online, ahead of the copy edited and typeset version being made available. Major corrections at proof stage, such as changes to figures, will be referred to the Editors for approval before they can be incorporated. Only minor changes, such as to style and consistency, should be made at proof stage. Changes that need to be made after proof stage will usually require a formal correction notice.

Yours sincerely,

Richard Carson
Senior Editor
The Journal of Physiology

P.S. - You can help your research get the attention it deserves! Check out Wiley's free Promotion Guide for best-practice recommendations for promoting your work at www.wileyauthors.com/eoo/guide. You can learn more about Wiley Editing Services which offers professional video, design, and writing services to create shareable video abstracts, infographics, conference posters, lay summaries, and research news stories for your research at www.wileyauthors.com/eoo/promotion.

IMPORTANT NOTICE ABOUT OPEN ACCESS: To assist authors whose funding agencies mandate public access to published research findings sooner than 12 months after publication, The Journal of Physiology allows authors to pay an Open Access (OA) fee to have their papers made freely available immediately on publication.

You can check if your funder or institution has a Wiley Open Access Account here: <https://authorservices.wiley.com/author-resources/Journal-Authors/licensing-and-open-access/open-access/author-compliance-tool.html>.

EDITOR COMMENTS

Reviewing Editor:

The authors have addressed all issues.